# FedAST: Federated Asynchronous Simultaneous Training

Baris Askin[1]      Pranay Sharma[1]      Carlee Joe-Wong[1]      Gauri Joshi[1]

[1]Carnegie Mellon University, Pittsburgh, Pennsylvania, USA

## Abstract

Federated Learning (FL) enables edge devices or *clients* to collaboratively train machine learning (ML) models without sharing their private data. Much of the existing work in FL focuses on efficiently learning a model for a single task. In this paper, we study *simultaneous training* of multiple FL models using a common set of clients. The few existing simultaneous training methods employ synchronous aggregation of client updates, which can cause significant delays because large models and/or slow clients can bottleneck the aggregation. On the other hand, a naïve asynchronous aggregation is adversely affected by stale client updates. We propose `FedAST`, a buffered asynchronous federated simultaneous training algorithm that overcomes bottlenecks from slow models and adaptively allocates client resources across heterogeneous tasks. We provide theoretical convergence guarantees of `FedAST` for smooth non-convex objective functions. Extensive experiments over multiple real-world datasets demonstrate that our proposed method outperforms existing simultaneous FL approaches, achieving up to $46.0\%$ reduction in time to train multiple tasks to completion.

## 1 INTRODUCTION

Federated Learning (FL) is a distributed learning paradigm where edge devices or *clients* collaboratively train machine learning (ML) models using privately held local data [McMahan et al., 2017, Kairouz et al., 2021]. Clients iteratively update their local models, which are periodically sent to a central server for aggregation. The aggregated model is then sent to the clients to begin the next round of local updates. Since its introduction in [McMahan et al., 2017], various practical and theoretical aspects of FL, in-

cluding client selection [Nishio and Yonetani, 2019, Cho et al., 2022], communication challenges [Ang et al., 2020, Chellapandi et al., 2023], scalability and fast training [Xie et al., 2019, Wang et al., 2020b], have been extensively studied. However, these works almost exclusively assume that the server aims to learn model(s) for a *single task*. Some FL frameworks attempt to learn models personalized to each client [Mansour et al., 2020, Li et al., 2021, Tan et al., 2022], but these models are still intended for the same learning task, e.g., next-word prediction on keyboards.

Many practical applications need devices to perform a wide range of learning tasks, which require training of multiple ML models. For instance, our phones need language models for keyboard next-word prediction as well as image recommendation models to highlight images more likely to be shared [McMahan et al., 2017]. Yao et al. [2023] propose training multiple models in federated smart car networks for different tasks, such as pothole detection and maneuver prediction. Another example can be a chat application requiring speech recognition and response text generator models concurrently, while Le et al. [2022] suggest federated learning of multiple models for air quality index forecasting. Thus, in this paper, we seek to answer the following question:

*How can we efficiently train models for multiple tasks in a federated setting using a shared pool of clients?*

**Simple Solutions that Extend FedAvg.** A naïve approach to training multiple models is *sequential training*, where the models corresponding to different tasks are trained one at a time, each utilizing all the clients. The total training runtime of this approach scales linearly with the number of tasks. An alternative is for all the clients to train all tasks at the same time. However, with this approach each client will have to keep all models in memory, which is infeasible for resource-limited edge clients such as smartphones. To preserve memory, clients will have to queue the training requests and process them sequentially, again resulting in the runtime linearly increasing with the number of tasks. On the other hand, *parallel* or *simultaneous training* (ST) of all

the models with time-varying subsets of clients assigned to each task can strike a better trade-off between accuracy and runtime. Bhuyan et al. [2023]'s approach assigns a disjoint subset of clients to each model in each round, which significantly improves the time taken to reach a target accuracy as compared to sequential training. However, these federated simultaneous training (FST) approaches leave room for significant improvement. There are two particular drawbacks: 1) *straggler delays* due to synchronous aggregation, and 2) the *lack of adaptation* to the training progress of heterogeneous tasks, which we address in this work.

**Synchronous Aggregation and Straggler Delays.** Conventional FL employs synchronous aggregation, where in each round, the server waits to receive updates from all the participating clients before each aggregation. However, when the clients have diverse hardware and communication capabilities, faster clients must remain idle until slow or *straggling* clients finish, causing a large wallclock runtime to complete each communication round. This problem is further exacerbated in FL with multiple simultaneous models [Bhuyan et al., 2023, Zhou et al., 2022], where the aggregation is synchronized across tasks as well. Therefore, the server has to wait for the slowest client across *all* the parallel tasks. Solutions proposed to alleviate the straggler problem in the single-model context include allowing faster clients to run more local steps [Wang et al., 2020b], aggregating only the client updates that arrive before a timeout [Bonawitz et al., 2019], and sub-sampling from the set of available clients [Luo et al., 2022]. Although these approaches perform well when stragglers appear uniformly at random, they do not work well in the simultaneous training setting because some models (e.g., larger ones) are naturally slower to train. When the multiple models have inherently different training times, synchronized global aggregation rounds are bottlenecked by the slowest client assigned to the most computationally intensive model, leading to large idle times.

**Asynchronous Aggregation and Staleness Issues.** Another solution to the straggler problem is asynchronous aggregation at the server, as proposed in AsyncFL [Xie et al., 2019], where the server updates the global model whenever it receives any client update. While asynchronous aggregation has been extensively studied in single-model federated learning [Chen et al., 2020, Wang et al., 2022, Xu et al., 2023, Yu et al., 2023a], it has not been well-explored for simultaneous federated training. Although AsyncFL addresses the straggler issue, it suffers from undesired *staleness* even in the standard FL setting, since the received client updates are often based on outdated models. To alleviate the staleness problem in single-model FL, Nguyen et al. [2022] proposed storing the incoming client updates in a buffer at the server and aggregating when the buffer is full.

**Adaptive Allocation of Clients to Heterogeneous Tasks.** In this work, we employ asynchronous buffered aggregation to overcome the straggler issue while controlling staleness. However, extending single-model FL algorithms [Xie et al., 2019, Nguyen et al., 2022] to the simultaneous training of multiple models is not straightforward — running multiple independent instances of asynchronous FL can be suboptimal. This is because the tasks can have heterogeneous computation complexities and different data heterogeneity that affect both the number of rounds required to achieve a given target accuracy as well as the wall-clock time taken to complete each round. Since a shared set of clients is used to train the models, the training processes are coupled – more resources assigned to one task implies less for the others. Moreover, the optimal resource requirement for each task can change over time according to its data heterogeneity and training progress and may be difficult to predict before training. Therefore, we propose an adaptive algorithm that *dynamically* reallocates clients across tasks depending on their training progress, and also adapts the buffer size used for asynchronous aggregation of updates.

**Our Contributions.** We formalize the FST setting in Section 2 and then make the following main contributions:

- We introduce `FedAST`, a Federated Asynchronous Simultaneous Training algorithm[1] to simultaneously train models for multiple tasks (Section 3). Our work is one of the first to mitigate the straggler problem faced by synchronous FST methods that extend vanilla FedAvg.

- The proposed algorithm addresses the problem of balancing resources across heterogeneous tasks, a unique challenge to the FST framework, using novel dynamic client allocation, and it also dynamically adjusts the buffer size used in asynchronous aggregation to strike the best trade-off between staleness and runtime.

- We provide a theoretical convergence analysis of `FedAST` (Section 4), which improves previous analyses even in the single-model FL setting. It improves upon [Koloskova et al., 2022] by considering multiple local updates and the buffer, and on [Nguyen et al., 2022] by relaxing the restrictive assumptions.

- We experimentally validate `FedAST`'s performance (Section 5) in terms of its wall-clock training time and model accuracy on multiple real-world datasets compared to synchronous and asynchronous FL baselines.

We conclude and discuss future work in Section 6.

**Related Work.** Only a few recent works [Zhou et al., 2022, Bhuyan and Moharir, 2022, Siew et al., 2023, Bhuyan et al., 2023] consider federated simultaneous training of multiple models. In [Zhou et al., 2022], clients are selected

---

[1]Our code is provided at `https://github.com/askinb/FedAST`.

with either Bayesian optimization or reinforcement learning to minimize training time and unfairness in participation. Bhuyan and Moharir [2022] formulate the client assignment FL as a bandit problem leveraging local training losses as scores. Siew et al. [2023] introduce biased client sampling, favoring the clients with higher local losses. These methods lack convergence guarantees. Bhuyan et al. [2023] assign clients uniformly at random or in a round-robin fashion and analyze the convergence assuming convex objective functions and bounded gradients. While these works only consider synchronous aggregation, Chang et al. [2023] propose a fully asynchronous FST algorithm. Their approach entails solving a non-convex optimization problem to optimize client assignment, which requires information about delays and models that may be difficult to obtain in practice. Also, the obtained bound does not converge to a stationary point in the presence of data heterogeneity and suffers from increased staleness when the number of clients increases. Lastly, Liu et al. [2023] propose an extension of their single-model adaptive asynchronous approach to the multi-model setting. However, they do not carefully handle heterogeneous data distributions across clients and the staleness of updates in their theoretical guarantees for a single model, and they lack these guarantees for multiple models. Also, their method under-utilizes client resources, since after sending a model update to the server, the clients are idle until the next training round.

## 2   PROBLEM FORMULATION

**Notations.**   For a positive integer $c$, we define $[c] \triangleq \{1, \ldots, c\}$. $\widetilde{\nabla}$ denotes stochastic gradients. Bold lowercase letters (e.g., $\mathbf{x}$) denote vectors. $|A|$ denotes the cardinality of set $A$. $\|\cdot\|$ denotes the Euclidean norm.

We now formally introduce the federated simultaneous training (FST) setting, where $N$ clients train $M$ models $\mathbf{x}_1, \ldots, \mathbf{x}_M$ corresponding to $M$ independent tasks. For each task $m \in [M]$, our goal is to find the model that solves that following optimization problem:

$$\min_{\mathbf{x} \in \mathbb{R}^{d_m}} \left\{ f_m(\mathbf{x}) := \frac{1}{N} \sum_{i=1}^{N} f_{m,i}(\mathbf{x}) \right\}, \qquad (1)$$

where $f_m$ is the global loss function for task $m$, and $f_{m,i}$ is the local loss for task $m$ at client $i$.

First, we examine a simple extension of FedAvg [McMahan et al., 2017] to simultaneous training of models for $M$ tasks. At the start of each round, the server randomly partitions the available set of clients across the tasks [Bhuyan et al., 2023]. The server sends the current models $\{\mathbf{x}_m^{(t_m)}\}_{m=1}^{M}$ for all the tasks to the corresponding subset of clients. The clients perform local training (Algorithm 1) and return their updates to the server, which *synchronously* aggregates the updates for each task. This naïve simultaneous training extension of

---

**Algorithm 1** $\mathtt{LocalTrain}(m, \tau_m, \mathbf{x}_m^{(t_m)}, \eta_m^c)$ at client $i$

---

1: **Set** $\mathbf{x}_{m,i}^{(t_m,0)} \leftarrow \mathbf{x}_m^{(t_m)}$
2: **for** $k = 1, \ldots, \tau_m$ **do**
3: $\quad \mathbf{x}_{m,i}^{(t_m,k)} \leftarrow \mathbf{x}_{m,i}^{(t,k-1)} - \eta_m^c \widetilde{\nabla} f_{m,i}(\mathbf{x}_{m,i}^{(t,k-1)})$
4: **end for**
5: **Return** $\boldsymbol{\Delta}_m \leftarrow (\mathbf{x}_m^{(t_m)} - \mathbf{x}_{m,i}^{(t_m,\tau_m)})/(\tau_m \eta_m^c)$

---

FedAvg performs poorly due to stragglers. The time it takes for a client to return its updates depends on its resources and the size of the model assigned. Since the server waits for the slowest update across all the tasks before commencing the next round, the server waits much longer if a large model is assigned to a slow client. We mitigate this problem via *asynchronous* training in $\mathtt{FedAST}$, discussed next.

## 3   ALGORITHM DESCRIPTION

Next, we describe $\mathtt{FedAST}$ (Algorithm 2), our proposed Federated Asynchronous Simultaneous Training algorithm, illustrated in Figure 1 for $M = 2$ tasks. For each task $m \in [M]$, the server maintains a round index $t_m$ that is initialized to $t_m = 0$, the number of active training requests $R_m^{(t_m)}$, and buffer size $b_m^{(t_m)}$. $R_m^{(t_m)}$ and $b_m^{(t_m)}$ quantify the resources (client computation and memory) allocated to task $m$ in round $t_m$. We provide two versions of $\mathtt{FedAST}$ based on the value of *option* $\in \{S, D\}$. When *option* is $S$ (*static*), the resource allocation for each task remains the same throughout the training process (i.e., $R_m^{(t_m)} \equiv R_m$ and $b_m^{(t_m)} \equiv b_m$). With *option* $= D$ (*dynamic*), $\mathtt{FedAST}$ dynamically reallocates resources across tasks using the $\mathtt{Realloc}$ subroutine (Algorithm 3).

**Assignment of Local Training Requests to Clients and Their Execution.**   Consider task $m \in [M]$. The server begins by sending out $R_m^{(0)}$ local training requests for task $m$ to clients selected uniformly at random, along with the initial model $\mathbf{x}_m^{(0)}$ (Algorithm 2, Line 4). The number of local training requests $R_m^{(t_m)}$ is adapted over time using the $\mathtt{Realloc}$ function (Algorithm 3), enabling us to dynamically reallocate client resources across tasks. Each client processes the training request by performing $\tau_m$ local mini-batch SGD iterations (see Algorithm 1) and sends the resulting model update $\boldsymbol{\Delta}_m$ back to the server. If a client receives multiple requests, they are queued and processed in a first-come-first-served manner.[2] Therefore, the number of *active clients* (clients working on training requests) at any time might be less than the number of *active training requests* (that clients are working on or are stored in their queues).

---

[2] Processing the requests in parallel would require clients to keep all the $M$ models in local memory, which can be infeasible.

1: **Input**: Client and server learning rates $\{\eta_m^c, \eta_m^s\}_{m=1}^M$, *option* $\in \{S, D\}$, no. of local updates $\{\tau_m\}_{m=1}^M$
2: **Initialize:** $\forall m \in [M]$: $t_m \leftarrow 0$ (round index), model $\mathbf{x}_m^{(0)}$, buffer $B_m \leftarrow \emptyset$. Total no. of updates $c \leftarrow 0$
3: **for** Models $m = 1, \ldots, M$ (in parallel) **do**
4:     Randomly select $R_m^{(0)}$ clients and send LocalTrain($m$, $\tau_m$, $\mathbf{x}_m^{(0)}$, $\eta_m^c$) requests
5:     **while** $t_m < T_m$ **do**
6:         Wait until server receives an update $\mathbf{\Delta}_m$
7:         $B_m \leftarrow B_m \cup \{\mathbf{\Delta}_m\}$, $c \leftarrow c + 1$
8:         $\{(R_i^{(t_i+1)}, b_i^{(t_i+1)})\}_{i=1}^M \leftarrow$ Realloc ($option, c$)
9:         **if** $|B_m| = b_m^{(t_m)}$ **then**
10:            $\mathbf{x}_m^{(t_m+1)} \leftarrow \mathbf{x}_m^{(t_m)} - \eta_m^s \eta_m^c \tau_m \frac{1}{b_m^{(t_m)}} \sum_{\mathbf{\Delta} \in B_m} \mathbf{\Delta}$
11:            $t_m \leftarrow t_m + 1$ and $B_m \leftarrow \emptyset$
12:         **end if**
13:         Select $K_m^{(t_m)}$ random client(s) and send LocalTrain($m, \tau_m, \mathbf{x}_m^{(t_m)}, \eta_m^c$) request(s)
14:     **end while**
15: **end for**
16: **Output**: Trained models $\{\mathbf{x}_m^{(T_m)}\}_{m=1}^M$

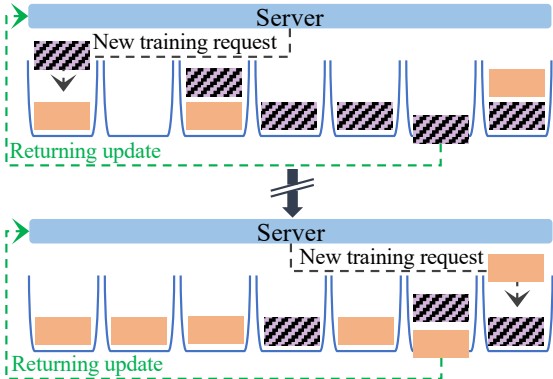

Figure 1: In our proposed algorithm FedAST, the server assigns local training requests (shown in striped and orange blocks for two simultaneous tasks), which are queued at the clients and processed in a first-come-first-served manner. Completed requests are aggregated asynchronously at the server. In the figure, snapshots of the process at two different times are seen. Adjusting the number of requests, FedAST periodically *reallocates* the resources shared across models.

**Buffered Asynchronous Aggregation at the Server.** The updates $\mathbf{\Delta}_m$ sent by the clients are aggregated at the server in an asynchronous manner as follows. To keep staleness in check, the server maintains a buffer $B_m$ for task $m$, which stores the received client updates for model $m$ (Algorithm 2, Line 7). The buffer size $b_m^{(t_m)}$ can be adapted over time (using the Realloc function). Whenever the server receives an update for task $m$, it randomly selects $K_m^{(t_m)}$ client(s) to send a new training request along with the current global model (Algorithm 2, Line 13). As we explain below, $K_m^{(t_m)} = 1$ (respectively, $K_m^{(t_m)} \in \{0, 1, 2\}$) for *option* $= S$ (*option* $= D$). When the buffer for model $m$ gets full (formally, $|B_m| = b_m^{(t_m)}$) the server aggregates the updates stored in the buffer to update the global model (Algorithm 2, Line 10).

**Dynamic Adaptation of the Number of Active Requests and Buffer Size using Realloc (Algorithm 3).** With the *static* option (*option* $= S$), the Realloc subroutine always runs its Line 6 to maintain the initial values of $R_m$ and $b_m$ throughout the whole training process. The resource allocation across tasks does not change over time. On the other hand, with the *dynamic* option, the Realloc subroutine adjusts the resource allocation during the training. The server maintains a counter $c$, tracking the total number of updates received across all $M$ tasks (Algorithm 2, Line 7). If *option* $= D$ (*dynamic*), this counter is used to periodically trigger the dynamic adaptation of the number of active training requests $R_m$ and the buffer size $b_m$ across tasks (Algorithm 2, Line 8). Intuitively, we should allocate more

clients (and consequently, more training requests $R_m$) to tasks with larger inter-client data heterogeneity. To empirically estimate this heterogeneity, the server stores the last $V$ ($V$ is a tunable parameter) updates $\mathbf{\Delta}_m$ for each task $m$ (denoted $\{\mathbf{\Delta}_{m,i}\}_{i=1}^V$) and computes

$$\hat{\sigma}_{g,m}^2 \propto \frac{1}{V} \times \sum_{i=1}^V \frac{\|\mathbf{\Delta}_{m,i} - \overline{\mathbf{\Delta}_m}\|^2}{\|\overline{\mathbf{\Delta}_m}\|^2}, \qquad (2)$$

where $\overline{\mathbf{\Delta}_m}$ is the empirical mean of the $\mathbf{\Delta}_{m,i}$'s.[3] Further, in our experiments, we empirically observe that the optimal choice of buffer size $b_m$ is proportional to the number of active requests $R_m$. See Appendix C.5 for our extensive experiments. Using (2) and these empirical observations, the optimal resource allocation emerges as the solution to the following constraints.

$$\sum_{i=1}^M R_i^{(t_i+1)} = \sum_{i=1}^M R_i^{(t_i)},$$
$$\frac{R_1^{(t_1+1)}}{\hat{\sigma}_{g,1}} = \frac{R_2^{(t_2+1)}}{\hat{\sigma}_{g,2}} = \cdots = \frac{R_M^{(t_M+1)}}{\hat{\sigma}_{g,M}}, \qquad (3)$$

where the first set of constraints maintains the total computation budget across tasks, and the second set ensures the allocation of a larger number of training requests to clients with higher heterogeneity. We elaborate on the theoretical motivation for the second set of constraints in Section 4, once we establish our convergence results. We also refer the reader to Appendix A for more details on Realloc.

**Sending out New Requests to Reach the New Resource Allocation.** To transition from one allocation $\{R_m^{(t_i)}\}_m$ to

---

[3]We normalize by $\|\overline{\mathbf{\Delta}_m}\|^2$ to account for different model sizes since larger models often have larger unnormalized variance.

**Algorithm 3** Realloc(*option,c*)

---

1: **if** *option* = $D$ and $c \mod c_{period} = 0$ **then**
2:      $\{\hat{\sigma}_{g,m}^2\}_{m=1}^M \leftarrow$ EstimateVariances()
3:      Find $\{R_m^{(t_m+1)}\}_{m=1}^M$ that solves (3)
4:      $b_m^{(t_m+1)} \leftarrow \left(b_m^{(t_m)} R_m^{(t_m+1)}\right)/R_m^{(t_m)}$ for all $m \in [M]$
5: **else**: **for all** $m \in \{i : R_i^{(t_i+1)} \text{not defined}\}$ **do**
6:      $(R_m^{(t_m+1)}, b_m^{(t_m+1)}) \leftarrow (R_m^{(t_m)}, b_m^{(t_m)})$
7: **end if**
8: **Return** $\{(R_m^{(t_m+1)}, b_m^{(t_m+1)})\}_{m=1}^M$

---

another $\{R_m^{(t_m+1)}\}_m$ in an asynchronous setting, we must adjust the number of new requests that are sent out every time the server receives a client update. The number of new requests $K_m^{(t_m)}$ sent out on receiving any update $\boldsymbol{\Delta}_m$ is always 1 in the static (*option* = $S$) case since $R_m$ remains constant throughout training. In the dynamic case (*option* = $D$), $K_m^{(t_m)}$ can be 0 (when $R_m^{(t_m+1)} < R_m^{(t_m)}$), 1 (when $R_m^{(t_m+1)} = R_m^{(t_m)}$), or 2 (when $R_m^{(t_m+1)} > R_m^{(t_m)}$). We employ this gradual transition to the desired new number of active training requests $\{R_m^{(t_m+1)}\}_m$ for each task instead of a sudden change in allocation to avoid possible longer queues at the clients during the transition phase.

## 4 CONVERGENCE ANALYSIS

In this section, we provide the convergence result for FedAST with the static *option* ($S$). Since $R_m^{(t_m)}$ and $b_m^{(t_m)}$ are constant when *option* = $S$, we drop time indices for simplicity. The convergence with dynamic allocation (*option* = $D$) can be shown with an additional assumption. We relegate this to Appendix F due to space limitations.

Next, we discuss the assumptions used in our analysis.

**Assumption 1** (Smoothness). *The loss functions are $L$-smooth, i.e., for all $i \in [N]$, for all $m \in [M]$, and for all $\mathbf{x}, \mathbf{y} \in \mathbb{R}^{d_m}$, $\|\nabla f_{m,i}(\mathbf{x}) - \nabla f_{m,i}(\mathbf{y})\| \leq L \|\mathbf{x} - \mathbf{y}\|$.*

**Assumption 2** (Bounded Variance). *The stochastic gradient at each client is an unbiased, bounded-variance estimator of the true local gradient, i.e., for all $\mathbf{x} \in \mathbb{R}^{d_m}$, $i \in [N]$, and $m \in [M]$, $\mathbb{E}[\widetilde{\nabla} f_{m,i}(\mathbf{x})] = \nabla f_{m,i}(\mathbf{x})$ and $\mathbb{E}\|\widetilde{\nabla} f_{m,i}(\mathbf{x}) - \nabla f_{m,i}(\mathbf{x})\|^2 \leq \sigma_{l,m}^2$.*

**Assumption 3** (Bounded Heterogeneity). *The local gradients are within bounded distance of the global gradient, such that for all $m \in [M]$ and $\mathbf{x} \in \mathbb{R}^{d_m}$, $\max\limits_{i \in [N]} \|\nabla f_{m,i}(\mathbf{x}) - \nabla f_m(\mathbf{x})\|^2 \leq \sigma_{g,m}^2$.*

**Assumption 4** (Bounded Staleness). *The client updates of task $m$ are received within at most $\gamma_m^{max}$ server model updates after the server sends the training request.*

These assumptions are standard in the literature. Assumptions 1-3 are commonly used in the synchronous [Wang

et al., 2020b, Jhunjhunwala et al., 2022] and asynchronous [Koloskova et al., 2022, Nguyen et al., 2022] FL analyses. Assumption 4 is used in the convergence proof to guarantee that none of the requested client updates takes an arbitrarily large time to return to the server and is also common in asynchronous FL works [Koloskova et al., 2022, Nguyen et al., 2022]. Furthermore, the maximum staleness can be enforced by dropping over-delayed updates in practice during the training.

**Theorem 1** (Convergence of FedAST). *Suppose that Assumptions 1 - 4 hold, and there are $R_m$ active local training requests corresponding to task $m \in [M]$, and the server and client learning rates, $\{\eta_m^s, \eta_m^c\}$ respectively, satisfy $\eta_m^s \leq \sqrt{\tau_m b_m}$ and $\eta_m^c \leq \min\{(6L\tau_m\sqrt{\tau_m b_m})^{-1}, (4L\tau_m\sqrt{\tau_m R_m \gamma_m^{max}})^{-1}\}$ for all tasks $m \in [M]$. Here, $b_m$ is the buffer size, and $\tau_m$ is the number of local training steps. Then, the iterates, $\{\{\mathbf{x}^{(t)}\}_{t=1}^{T_m}\}_{m=1}^M$, of Algorithm 2 satisfy:*

$$\frac{1}{T_m}\sum_{t=0}^{T_m-1}\mathbb{E}\|\nabla f_m(\mathbf{x}^{(t)})\|^2 \leq \underbrace{\mathcal{O}\left(\frac{\delta_m}{T_m \eta_m^c \eta_m^s \tau_m}\right)}_{\textit{FedAvg Error - I}}$$

$$+ \underbrace{\mathcal{O}\left(\left(\frac{L\eta_m^c \eta_m^s}{b_m} + L^2[\eta_m^c]^2\tau_m\right)\left(\sigma_{l,m}^2 + \tau_m\sigma_{g,m}^2\right)\right)}_{\textit{FedAvg Error - II}}$$

$$+ \underbrace{\mathcal{O}\left(\frac{L^2[\eta_m^s]^2[\eta_m^c]^2\tau_m R_m}{b_m^2}(\sigma_{l,m}^2 + \tau_m R_m \sigma_{g,m}^2)\right)}_{\textit{Asynchronous Aggregation Error}}, \quad (4)$$

*where $\delta_m = f_m(\mathbf{x}_m^{(0)}) - \min_{\mathbf{x}} f_m(\mathbf{x})$.*

**Proof.** See Section E in the Appendix.

**Comparison with Synchronous FL Analyses.** The *FedAvg Error - I* and *- II* terms in (4) capture the error bound for synchronous FedAvg [Jhunjhunwala et al., 2022, Theorem 1]. Since the server updates for model $m$ involve aggregating $b_m$ client updates, the buffer size $b_m$ is analogous to the number of participating clients in FedAvg. The third error term in (4) arises due to asynchronous aggregation and increases with $R_m$, the number of active local training requests. Intuitively, given the same buffer size $b_m$, increasing $R_m$ leads to higher worst-case staleness $\gamma_m^{max}$. However, as long as $L\eta_m^s\eta_m^c R_m^2\tau_m \leq b_m$, asynchrony is not the dominant source of error in (4), and we achieve the same rate of convergence as synchronous FedAvg (see Corollary 1.1).

**Comparison with Asynchronous FL Analyses.** FedBuff [Nguyen et al., 2022] considers buffered asynchronous aggregation for a single model. Still, comparing [Nguyen et al., 2022, Corollary 1] and the bound in (4) for $M = 1$, their convergence result (i) depends on stronger assumptions (bounded gradient norm and uniform arrivals of client

updates), and (ii) has worse asynchronous aggregation error. Moreover, our analysis is more general compared to [Koloskova et al., 2022] as they do not consider multiple local SGD steps and the buffer. Simultaneous asynchronous training is considered by [Chang et al., 2023], but we observe that they do not achieve convergence unless the data distribution across clients is identical (see [Chang et al., 2023, Eq. (19)]). We discuss the comparison of `FedAST` to single-model and simultaneous asynchronous federated training baselines in more detail in Appendix Section B.

**Corollary 1.1** (Asymptotic convergence after setting learning rates)**.** *Let $T_m \geq \tau_m \max \{36 b_m, 16 R_m \gamma^{max}\}$. Setting the learning rates $\eta_m^c = (\tau_m L \sqrt{T_m})^{-1}, \eta_m^s = \sqrt{\tau_m b_m}$, the bound in Theorem 1 reduces to:*

$$\frac{1}{T_m} \sum_{t=0}^{T_m-1} \mathbb{E} \|\nabla f_m(\mathbf{x}^{(t)})\|^2 \leq \mathcal{O}\left(\frac{\delta_m L}{\sqrt{b_m \tau_m T_m}}\right)$$
$$+ \mathcal{O}\left(\left(\frac{1}{\sqrt{T_m b_m \tau_m}} + \frac{1}{\tau_m T_m}\right)(\sigma_{l,m}^2 + \tau_m \sigma_{g,m}^2)\right)$$
$$+ \mathcal{O}\left(\frac{R_m}{T_m b_m}(\sigma_{l,m}^2 + \tau_m R_m \sigma_{g,m}^2)\right). \quad (5)$$

Although the given bound in Corollary 1.1 does not seem to depend on the staleness bound $\gamma^{\max}$ (Assumption 4), its effect is implicit in the number of active requests $R_m$ and buffer size $b_m$. The maximum staleness is positively correlated with $R_m$ and negatively correlated with $b_m$. In our experiments (Appendix C.5), we tune the buffer size to maintain the update staleness at a reasonable level.

Looking at the bounds in (4) or (5), increasing $R_m$ makes the bound worse because to reach the same accuracy in (5), we need to run a higher number of server updates $T_m$. However, increasing $R_m$ also shortens the duration between two successive server updates, making the algorithm faster in wall-clock time. We illustrate this effect with a wall-clock comparison to FST baselines below.

**Impact of $R_m$ on Wall-clock Time.** Suppose the arrival times of all the client updates (assuming there is no queue on the clients) are distributed as $Exp(\lambda)$. The expected time to fill the buffer corresponding to task $m$ is $b_m/(R_m \lambda)$. Therefore, in `FedAST`, the expected time to complete one round at the server is inversely proportional to $R_m$. On the other hand, the expected time to finish one round of synchronous simultaneous FedAvg training is $\frac{1}{\lambda} \sum_{k=1}^{R_1+\cdots+R_M} \frac{1}{k} \approx \frac{1}{\lambda} \log(\sum_{k=1}^M R_k)$, which increases with $R_m$. Also, the summation over simultaneously trained tasks shows an exacerbated straggler effect since all the clients wait for the slowest client across all the tasks.

**Design of `Realloc` (Algorithm 3).** Next, we theoretically justify the dynamic allocation of resources across tasks described in Section 3 (Algorithm 3, with *option = D*), which adjusts the number of active requests ($\{R_m\}_{m=1}^M$). Given the limited number of available clients (which limits the total number of active training requests), to achieve

the best possible allocation, we minimize the sum of the most dominant terms in the bounds (FedAvg Error-II in (4)) across tasks. We also use the empirical observation that the optimal choice of buffer-size $b_m$ scales linearly with $R_m$ (Appendix C.5). The resulting optimization problem is

$$\min_{\{R_m, b_m\}_{m=1}^M} \sum_{m=1}^M \frac{\eta_m^s \eta_m^c \tau_m}{R_m} \sigma_{g,m}^2 \text{ s.t. } \sum_{m=1}^M R_m = R, \quad (6)$$

where $R$ is the budget for the total number of training requests across all tasks in the system depending on the number of available clients. The `Realloc` function (Algorithm 3) solves the optimization problem (6). See Appendix A for more details.

## 5 EXPERIMENTAL RESULTS

We outline our experimental setup in Section 5.1, discuss the existing baselines in Section 5.2, and compare the baselines with `FedAST` under varied settings in Section 5.3.

### 5.1 DATASETS AND IMPLEMENTATION

We consider image classification tasks with the MNIST [Deng, 2012], Fashion-MNIST [Xiao et al., 2017] and CIFAR-10 [Krizhevsky et al., 2009] datasets, and next character prediction with the Shakespeare [Caldas et al., 2019] dataset using the same models as in previous works [Acar et al., 2021, Yu et al., 2023b, Lecun et al., 1998]. We compare the wall-clock time required by different algorithms to reach some predetermined target test accuracy levels (see Table 1). In Appendix D.5, we present experiments with other target accuracy levels to show the consistency of our results. We also validate our results with ResNet-18, a larger model, trained for the CIFAR-100 classification task in Appendix D.4. In all experiments, we conduct three Monte Carlo runs with different random seeds and report the average results.

Table 1: The datasets and models used in experiments, along with corresponding target test accuracy levels.

| Dataset | Model | Target Accuracy |
|---|---|---|
| MNIST | MLP | 93% |
| Fashion-MNIST | LeNet-5 | 82% |
| CIFAR-10 | CNN | 63% |
| Shakespeare | LSTM | 42% |

For image classification tasks, we partition the training data across clients using the Dirichlet distribution with $\alpha = 0.1$ to create inter-client data heterogeneity [Yurochkin et al., 2019]. The Shakespeare dataset is *naturally* heterogeneous as the lines of each role in the plays of Shakespeare are assigned to a different client. There are a total of 1000 clients,

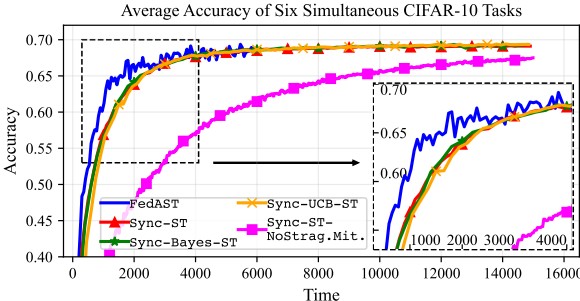

Figure 2: Mean test accuracy for compared algorithms on six identical CIFAR-10 tasks trained simultaneously. `FedAST` trains faster than synchronous methods. The synchronous method without straggler mitigation is by far the slowest.

30% of which are available to accept new training requests, independent of the past.

**Modeling Client Delays.** As suggested in [Lee et al., 2018, Dutta et al., 2021, Shi et al., 2021, Zhou et al., 2022], we use *shifted-exponential* (exponential plus constant) random variables to model the time taken by a client to complete a local training request and return the update to the server. We pick the run-time generation parameters of each task according to real measurements on NVIDIA GeForce GTX TITAN X GPUs. To simulate hardware heterogeneity across clients, we divide them into 25% slow, 50% normal-speed, and 25% fast clients [Leconte et al., 2023]. We relegate additional implementation details to the Appendix.

## 5.2 BASELINE ALGORITHMS

We explain the synchronous and asynchronous baseline methods to which we compare `FedAST`:

**Synchronous Simultaneous Training.** The following synchronous methods differ only in client selection.

1. `Sync-ST` [Bhuyan et al., 2023]: randomly partition the client set across tasks at each round;

2. `Sync-Bayes-ST` [Zhou et al., 2022]: Bayesian optimization-based assignment of clients to tasks;

3. `Sync-UCB-ST` [Bhuyan and Moharir, 2022]: client selection as a multi-armed bandit problem.

In Figure 2, we first simultaneously train six CIFAR-10 models and compare the performance of all synchronous baselines and `FedAST`. As synchronous methods perform poorly due to a severe straggler issue, we augment them with a straggler mitigation method by aggregating only the first $k$ client updates for each task and discarding the rest [Bonawitz et al., 2019] as the default option. We choose $k = 30$ by validation experiments across datasets in Appendix D.1. This extra augmentation *makes the baselines more competitive*. In Figure 2, we also add the result of `Sync-ST -NoStrag.Mit.`, which is the `Sync-ST`

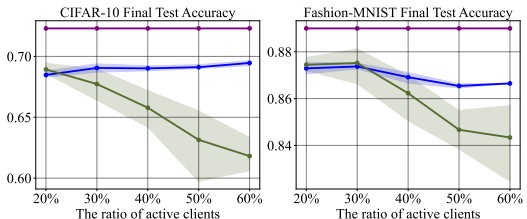

Figure 3: The mean final test accuracy values of `FedAST` (blue), `FedAST-NoBuffer` (olive green) and centralized training (violet) with varying active client ratio, when training 3 identical models. The left (right) figure is for CIFAR-10 (Fashion-MNIST) dataset. With more active clients, the importance of buffer increases due to increasing staleness.

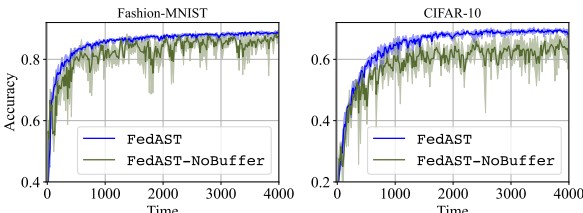

Figure 4: The mean test accuracy values of `FedAST` and `FedAST-NoBuffer`, when simultaneously training one model for CIFAR-10 and one for Fashion-MNIST. `FedAST` achieves higher and more stable accuracy levels.

without our augmented straggler mitigation. It shows that the synchronous baselines have a large straggler effect without our extra augmentation.

**Asynchronous Federated Simultaneous Training.** To our knowledge, [Chang et al., 2023] is the only other work that mainly studies asynchronous simultaneous FL. However, their client selection scheme requires the knowledge of network-wide staleness and smoothness constants, which are hard to estimate. If the tasks have similar model complexity and task difficulty, their client selection is similar to that of `FedAST` with a buffer size of 1. We thus include this *no-buffer* version of `FedAST` (we call it `FedAST-NoBuffer`) as a baseline.

## 5.3 RESULTS AND INSIGHTS

We assess the performance of `FedAST` under various scenarios. In *homogeneous-task* experiments, where multiple independent copies of the same model are trained simultaneously using the same dataset, we report the average accuracy over time. In *heterogeneous-task* experiments involving differing tasks and models, efficiently distributing resources to accelerate the completion of all tasks is the main challenge. For homogeneous tasks, we use `FedAST` with the static *option* $(S)$ and uniform client distribution across tasks. In heterogeneous-task experiments, we use dynamic allocation (*option* $= D$) to enhance resource allocation efficiency. To

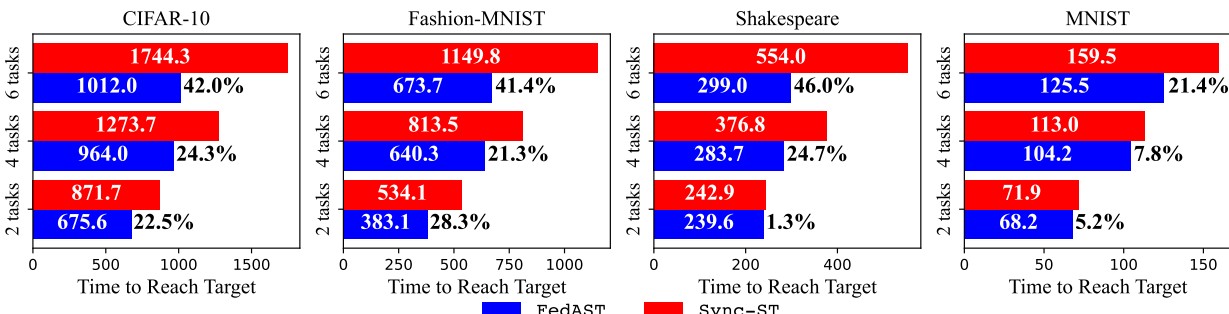

Figure 5: Mean training times of `FedAST` and `Sync-ST` to attain target accuracy levels in (Table 1) on 2/4/6 tasks with CIFAR-10, Fashion-MNIST, MNIST, and Shakespeare datasets. `FedAST` requires consistently lower wall-clock time for training compared to `Sync-ST`; the percentages represent these time gains.

show the benefits of dynamic allocation over static allocation, we also explore heterogeneous-task scenarios with *option* = $S$. Dynamic allocation reduces overall training time by up to 11.9%, with comprehensive results shown in Appendix D.6.

To quantify the time *saved* by using `FedAST` over some competing baseline, we define *time gain* as

$$\text{Gain} \triangleq \frac{T_{\texttt{Baseline}} - T_{\texttt{FedAST}}}{T_{\texttt{Baseline}}} \times 100\%,$$

where $T_{\texttt{Baseline}}$ ($T_{\texttt{FedAST}}$) is the simulated time for `Baseline` (`FedAST`) to reach the target accuracy.

**Comparison with All Synchronous FST Methods.** First, we compare the synchronous baselines discussed in Section 5.2 on the CIFAR-10 dataset (Figure 2), where we simultaneously train six identical models. We observe that synchronous methods without straggler mitigation converge very slowly. Among the straggler-mitigated synchronous variants that we implement, Figure 2 shows that `Sync-Bayes-ST`, has similar performance to `Sync-ST` because it struggles due to the large search space of the optimization problem, stemming from the exponential number of possible client schedules. Further, we do not observe any performance gains from using `Sync-UCB-ST` over `Sync-ST`. Given that `Sync-Bayes-ST` and `Sync-UCB-ST` have similar performance as `Sync-ST`, in subsequent experiments, we choose `Sync-ST` as the sole synchronous baseline.

**Need for Buffer.** As discussed earlier, incorporating the buffer mitigates the negative impact of highly stale updates. Since staleness increases with the number of active clients, asynchronous FL methods without a buffer exhibit limited scalability as the number of clients grows. To demonstrate this, in Figure 3, we conduct two experiments: 1) training three models for CIFAR-10 simultaneously, and 2) training three models for Fashion-MNIST simultaneously. We plot the final accuracy values varying the ratio of the active clients. We observe that for small

active client ratios, `FedAST` and `FedAST-NoBuffer` have comparable performance. However, with more active clients, the staleness of the updates increases, resulting in significantly worse performance of the fully asynchronous `FedAST-NoBuffer` algorithm. Then, in Figure 4, we simultaneously train two models, one each for CIFAR-10 and Fashion-MNIST. Observing the unsteady learning curves of `FedAST-NoBuffer`, we conclude that the buffer makes the system more robust to stale updates.

**Comparison with `Sync-ST`.** Next, we compare `FedAST` with the chosen synchronous method, `Sync-ST`.

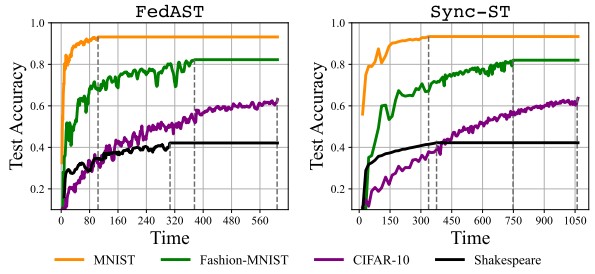

Figure 6: Training curves of a single Monte Carlo run of the *heterogeneous experiment*. Dashed vertical lines show times when tasks reach their target accuracy, with `FedAST` reaching it faster than `Sync-ST`.

1) *Homogeneous Tasks:* We conduct experiments training 2, 4, and 6 identical models for each of MNIST, Fashion-MNIST, CIFAR-10, and Shakespeare datasets. Figure 5 shows the average finish times of the algorithms, and the significant time gains of our algorithm `FedAST` over synchronous `Sync-ST` (even after incorporating straggler mitigation). We observe that the gain increases with the number of simultaneously trained tasks because `Sync-ST` is especially vulnerable to the straggler problem.

2) *Heterogeneous Tasks:* The heterogeneous experiment trains 4 models simultaneously, one each for the MNIST, Fashion-MNIST, CIFAR-10, and Shakespeare datasets. Once one model reaches its target accuracy, its training

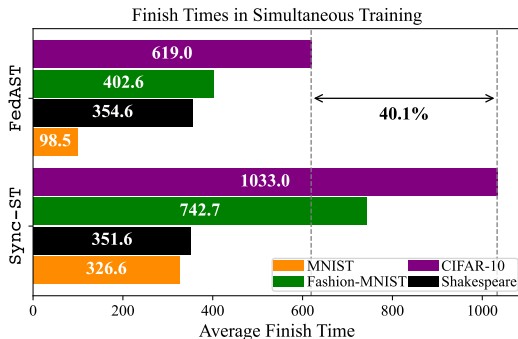

Figure 7: Mean time required to reach target accuracy and time gain of `FedAST` over `Sync-ST` in *the heterogeneous experiment*. While `FedAST` does not require manual fine-tuning, the client allocation in `Sync-ST` is tuned at 100, 84, 48, and 68 clients for MNIST, Fashion-MNIST, CIFAR-10, and Shakespeare tasks respectively. `FedAST` has notable time gain (40.1%) over `Sync-ST` to finish all tasks.

stops, and its clients are reallocated to other tasks. We use `FedAST` with *option = D* for dynamic client allocation. For the synchronous baseline `Sync-ST`, we ran 30 different client allocation schemes, including our proposed allocation scheme and uniform allocation across tasks. We report the results achieved with the best-performing scheme.

Figure 6 shows learning curves for `FedAST` and `Sync-ST` from a single Monte Carlo run. The dashed vertical lines denote the time instants when a model reaches its target accuracy, following which the clients training this model get reallocated to other tasks. Figure 7 shows the average finish times for 4 simultaneously trained models with `FedAST` and `Sync-ST`. For example, with `FedAST`, the model for MNIST dataset hits its target accuracy at 99, after which the clients training this model get reallocated to the other models. At 619, the training of the final model (for CIFAR-10) is complete. Comparing the finish times for the last model, `FedAST` provides a 40.1% time gain over `Sync-ST`.

We observe that thanks to dynamic client allocation, `FedAST` automatically detects which tasks have higher heterogeneity across clients and need a larger buffer. We notice that the Shakespeare task is allocated fewer clients because it is estimated to be less heterogeneous, which is true based on the label distribution of data samples across clients. We also repeat this experiment (Appendix D.6) using `FedAST` with static *option* (*S*) to validate the proposed dynamic client allocation strategy. Dynamic client allocation consistently has time gain up to 11.9% compared to the static version.

## 6 CONCLUSION

In this paper, we present `FedAST`, a federated learning framework to simultaneously train multiple models using buffered asynchronous aggregations. We theoretically prove the convergence of our algorithm for smooth non-convex objective functions. Experiments across multiple datasets, demonstrates the `FedAST`'s superiority over existing simultaneous FL baselines, achieving up to 46.0% reduction in training time. In future work, we plan to enhance `FedAST` by incorporating client selection based on local data distributions and computational powers of clients.

## ACKNOWLEDGEMENTS

This work was partially supported by the US National Science Foundation under grants CNS-1751075 and CNS-2106891 to CJW and NSF CCF 2045694, CNS-2112471, CPS-2111751, and ONR N00014-23-1-2149 to GJ and the Ben Cook Presidential Graduate Fellowship to BA.

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

# FedAST: Federated Asynchronous Simultaneous Training
## (Appendix)

**Baris Askin**[1]        **Pranay Sharma**[1]        **Carlee Joe-Wong**[1]        **Gauri Joshi**[1]

[1]Carnegie Mellon University, Pittsburgh, Pennsylvania, USA

## A  ADJUSTING THE NUMBER OF ACTIVE REQUESTS AND `REALLOC`

Before designing `Realloc` algorithm for dynamic client allocation *option* $(D)$, we conduct initial validation experiments with the static *option* $(S)$, wherein the allocation of active local training requests across clients and buffer sizes remain unchanged throughout the training. Note that with the static *option*, Algorithm 3 only executes its Line 6 and returns the previous round's values always. We *empirically* observe that setting the ratio of the number of active local training requests to buffer size fixed and below 37 works well. Refer to Section C.5 for the validation experiments. Then, incorporating this ratio ($R_m \approx 37 b_m$) within the convergence bound in Equation 4 of Theorem 1, we find out that the dominant term (excluding smoothness constants) becomes: $\mathcal{O}\left(\frac{\eta_m^s \eta_m^c \tau_m}{R_m}\right) \sigma_{g,m}^2$. Further, given the limited number of available clients, we cannot increase the total number of active local training requests arbitrarily without increasing the staleness. Thus, we employ $\sum_{m=1}^{M} R_m = R$ where $R$ is a constant of how many active training requests we assign in total depending on the number of available clients in the setting. Since the goal of federated simultaneous training is to minimize the objective functions of all tasks concurrently, we propose to adjust $\{R_m\}_{m=1}^{M}$ by solving,

$$\min_{\{R_m\}_{m=1}^{M}} \sum_{m=1}^{M} \frac{\eta_m^s \eta_m^c \tau_m}{R_m} \sigma_{g,m}^2 \text{ subject to } \sum_{m=1}^{M} R_m = R. \tag{7}$$

The solution of the minimization problem in (7) suggests allocating local training requests, $\{R_m\}_{m=1}^{M}$, in proportion to $\sigma_{g,m}\sqrt{\eta_m^s \eta_m^c \tau_m}$ for each model $m$. Using this approach, Algorithm 3 adjusts resource allocation. Further, as the update variance across clients may vary in time during training, we employ adaptive periodical reallocation of resources across models (Line 1 in Algorithm 3). Therefore, we use round indices to denote the changing number of active training requests and buffer sizes.

As we do not have access to true data heterogeneity levels, we need to estimate it (Line 2 in Algorithm 3). When `FedAST` is run with *option* $= D$ (dynamic client allocation option), the server keeps the latest $V$ updates of each model. This requires constant and small memory space kept in the server. To present how variance estimation (`EstimateVariances()`) works, assume that $\{\boldsymbol{\Delta}_{m,1}, \boldsymbol{\Delta}_{m,2}, \ldots, \boldsymbol{\Delta}_{m,V}\}_{m=1}^{M}$ are the sets of latest received updates of tasks $m \in [M]$ where each $\boldsymbol{\Delta}_{m,k}$ for $k \in [V]$ is the output of $k^{\text{th}}$ latest local training (Algorithm 1). As the output of any local training is the average of all calculated stochastic gradients during that local training, we use those outputs as approximations of the gradients calculated on the local data of clients. Algorithm 4 describes `EstimateVariances()`. It first calculates the mean of the latest updates for each task, $\{\overline{\boldsymbol{\Delta}_m}\}_{m=1}^{M} = \{\frac{1}{V} \times \sum_{i=1}^{V} \boldsymbol{\Delta}_{m,i}\}_{m=1}^{M}$. Then, `EstimateVariances()` returns sample variance multiplied with other terms ($\eta_m^s \eta_m^c \tau_m$) suggested by (7) and normalized by the mean update norm (to prevent large models or models with inherently large weights from dominating others), $\{\hat{\sigma}_{g,m}^2\}_{m=1}^{M} = \{\frac{\eta_m^s \eta_m^c \tau_m}{V} \times \sum_{i=1}^{V} \left\|\boldsymbol{\Delta}_{m,i} - \overline{\boldsymbol{\Delta}_m}\right\|^2 / \left\|\overline{\boldsymbol{\Delta}_m}\right\|^2\}_{m=1}^{M}$. Then, `Realloc` algorithm allocates the number of active training requests proportionally to the square root of these values (Algorithm 3, Line 3).

This approach is sensible both theoretically and intuitively. Based on our experimental observations regarding the relationship between the number of active training requests and buffer size, increasing the number of active local training requests necessitates an increase in buffer size. Moreover, a larger buffer proves beneficial in reducing the variance across updates,

---

**Algorithm 4** `EstimateVariances()`

---

**Require:** The set of latest $V$ updates $\{\boldsymbol{\Delta}_{m,1}, \boldsymbol{\Delta}_{m,2}, \ldots, \boldsymbol{\Delta}_{m,V}\}_{m=1}^{M}$, server-side learning rates $\{\eta_m^s\}_{m=1}^{M}$, client-side learning rates $\{\eta_m^c\}_{m=1}^{M}$, and the number of local SGD steps of all models $\{\tau_m\}_{m=1}^{M}$.

1: $\{\overline{\boldsymbol{\Delta}_m}\}_{m=1}^{M} \leftarrow \{\frac{1}{V} \times \sum_{i=1}^{V} \boldsymbol{\Delta}_{m,i}\}_{m=1}^{M}$                  ▷ Calculate the means of the latest updates
2: $\{\widetilde{\sigma}_{g,m}^2\}_{m=1}^{M} \leftarrow \{\frac{1}{V} \times \sum_{i=1}^{V} \left\|\boldsymbol{\Delta}_{m,i} - \overline{\boldsymbol{\Delta}_m}\right\|^2 / \left\|\overline{\boldsymbol{\Delta}_m}\right\|^2\}_{m=1}^{M}$      ▷ Calculate the normalized sample variances
3: $\{\hat{\sigma}_{g,m}^2\}_{m=1}^{M} \leftarrow \{\eta_m^c \eta_m^s \tau_m \widetilde{\sigma}_{g,m}^2\}_{m=1}^{M}$     ▷ Multiply with other constants suggested by the convergence guarantee (7)
4: **Return** $\{\hat{\sigma}_{g,m}^2\}$

---

the buffered updates are averaged during the aggregation. `Realloc` aims to allocate more clients and provide a larger buffer size for tasks with higher heterogeneity. We choose the number of stored latest updates $V = 8$ and the period of number of total updates from all clients to trigger reallocation in `Realloc` subroutine $c_{period} = 0.75 \times M \times \sum_{m=1}^{M} R_m$ in our experiments. The benefits of dynamic allocation (*option* $= D$) over static and uniform resource allocation (*option* $= S$) are demonstrated when tasks/models are heterogeneous, as shown in Figures 21-26.

# B    THEORETICAL COMPARISON OF `FedAST` WITH BASELINES

We compare `FedAST` with single-model FL methods, too. [Nguyen et al., 2022] is the most similar algorithm to `FedAST` (with single-model). However, even for the single-task case, `FedAST` differs by employing a uniform client assignment to ensure unbiased participation of clients irrespective of their hardware speeds. This allows us to relax the assumptions to prove the convergence guarantee. [Nguyen et al., 2022] relies on a strong assumption that the server receives updates from clients uniformly at random and that the norm of gradients is bounded. Moreover, compared to [Koloskova et al., 2022], our analysis is more general as `FedAST` uses multiple SGD steps in local training and a buffer. Some other recent single-model asynchronous FL works, [Zakerinia et al., 2023] and [Leconte et al., 2023], do not have straightforward and efficient simultaneous federated training extensions for multiple models.

[Chang et al., 2023] is another asynchronous simultaneous federated learning method. However, [Chang et al., 2023] indeed fails to converge to a stationary point asymptotically unless data is homogeneous, and their assumptions include Bounded Gradient Norm and Weak Convexity.

Table 2: Comparison of `FedAST`'s convergence guarantees to Nguyen et al. [2022] (single-task asynchronous buffered FL algorithm) and Chang et al. [2023] (an asynchronous FST algorithm). $T$: #global rounds, $\tau$: #local steps, $b$: buffer size.

| Algorithm | Non-standard assumptions | Convergence |
|:---:|:---:|:---:|
| Nguyen et al. [2022] | Bounded Gradient & Receiving Updates Uniformly | $\mathcal{O}\left(\sqrt{\tau/(Tb)}\right)$ [a] |
| Chang et al. [2023] | Bounded Gradient & Weak Convexity | Not converge |
| `FedAST` | — | $\mathcal{O}\left(\sqrt{\tau/(Tb)}\right)$ |

[a] Although the convergence guarantee in the published [Nguyen et al., 2022] paper seems to have a better rate, we pointed out a mistake in their proof. Here, we use the corrected version we received via private communication.

# C    EXPERIMENTAL SETUP DETAILS

In our study, we explore a simultaneous federated learning (FL) setting for multiple models. We present the details of our experiments in this section.

## C.1    SIMULATION ENVIRONMENT

We simulate the training with PyTorch on NVIDIA GeForce GTX TITAN X graphics processing units (GPUs) of our internal cluster. We build our code upon the public codes of [Wang et al., 2020a, Yu et al., 2023b].

## C.2 SETTING OVERVIEW

We consider the federated training of $M$ models simultaneously using $N$ clients. $N$ is 1000 in all experiments and $M$, specified for each experiment explicitly, varies between $2 - 6$.

## C.3 TASKS AND MODELS

We use 5 different tasks across the experiments: MNIST [Deng, 2012], Fashion-MNIST [Xiao et al., 2017], CIFAR-10 and CIFAR-100 [Krizhevsky et al., 2009] image classification tasks, and Shakespeare [Caldas et al., 2019] next character prediction task. We use a multilayer perceptron for MNIST as in [Acar et al., 2021], convolutional networks for Fashion-MNIST as in [Lecun et al., 1998] and for CIFAR-10 as in [Acar et al., 2021], ResNet-18 model for CIFAR-100 as in Acar et al. [2021], and a long short-term memory network for Shakespeare as in [Yu et al., 2023b].

## C.4 DATASETS AND DATA DISTRIBUTION

We consider the data heterogeneity across clients in FL frameworks. We download MNIST, Fashion-MNIST, and CIFAR-10/100 datasets from PyTorch built-in library methods. The train and test splits provided by the library are used without any modifications. To simulate heterogeneous data distribution across clients, we use Dirichlet distribution with $\alpha = 0.1$ following the approach suggested in Yurochkin et al. [2019]. We ensure that each client has 300 data points for MNIST, Fashion-MNIST, and CIFAR-10/100 tasks by repeating the train set if necessary. We obtain and preprocess the Shakespeare dataset as described in [Caldas et al., 2019]. This dataset has inherently heterogeneous distribution across clients as each client corresponds to a unique role from Shakespeare's plays.

## C.5 DESIGN PARAMETERS

In this section, we explain how we choose the design parameters.

**Client dataset sizes, batch sizes, and number of local steps.** While distributing CIFAR-10/100, MNIST, and Fashion-MNIST datasets across clients, each client is allocated 300 data points from each dataset. The Shakespeare dataset, however, maintains its original distribution of data points across roles, so clients have different numbers of data samples in the Shakespeare task. For CIFAR-10/100, MNIST, and Fashion-MNIST tasks, we set the batch size to 32 while we employ a batch size of 64 for the Shakespeare task. We fix the number of local steps in local training ($\tau_m$ parameter in Algorithm 1 in the main text) of clients at 27 for all tasks. This makes 3 epochs for CIFAR-10/100, MNIST, and Fashion-MNIST tasks. As the number of data points varies across clients for the Shakespeare dataset, there is no fixed number of epochs.

**Buffer size.** The buffer in FedAST is crucial for mitigating the negative impacts of highly stale updates, as extensively discussed in the main text. The staleness of updates is influenced by the number of active local training requests, denoted as $R_m$, and the buffer size, $b_m$, associated with all model $m \in [M]$. When FedAST is run with static *option* $(S)$, these numbers are kept constant during the training, but they may change (this time we denote $R_m^{(t_m)}$ and $b_m^{(t_m)}$) when we use dynamic client allocation *option* $(D)$. A higher number of simultaneous local training requests leads to a higher staleness because it increases the global model's update frequency at the server. On the other hand, buffer size is inversely related to staleness, given its opposing effect on the aggregation frequency. Based on our experimental observations, selecting the number of active training requests and the buffer size of model $m$ such that their ratio is fixed and below 37, ($R_m/b_m \lesssim 37$ or $R_m^{(t_m)}/b_m^{t_m} \lesssim 37$), works well. Selecting the buffer size of FedAST based on this observation avoids the detrimental effects of stale updates while benefiting from fast training thanks to the asynchronous algorithm. We show two experimental results in Figures 8 and 9. In Figure 8, we train one Fashion-MNIST and one CIFAR-10 models simultaneously by assigning 175 active training requests to each task and observe that buffer size of 5 strikes a balance between high final test accuracy and fast training to achieve the target accuracy for both tasks. In Figure 9, we repeat a similar experiment with MNIST and CIFAR-10 tasks by assigning 105 active training requests to each. This time, we observe that a buffer size of 3 performs the best for both tasks. These experimental results support our buffer size choice.

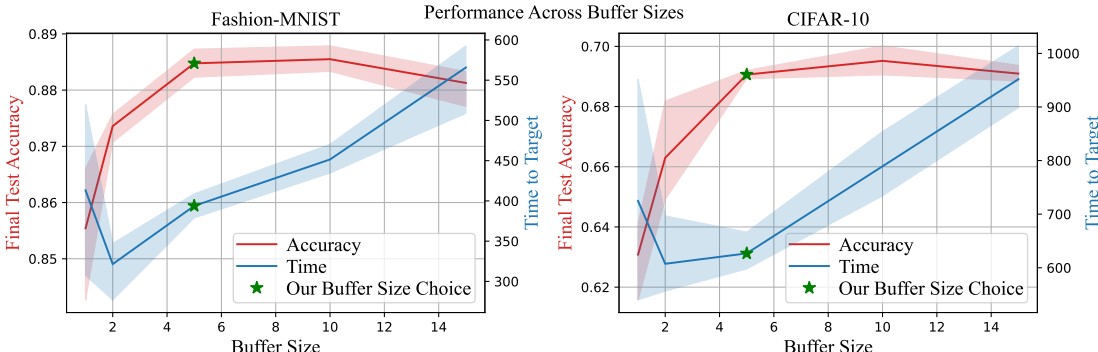

Figure 8: The final test accuracy and required time to get target accuracy (in Table 1) for simultaneous training (using `FedAST` with static *option*) of one Fashion-MNIST and one CIFAR-10 model with different buffer sizes. We assign the same number of local training requests (175) to each task.

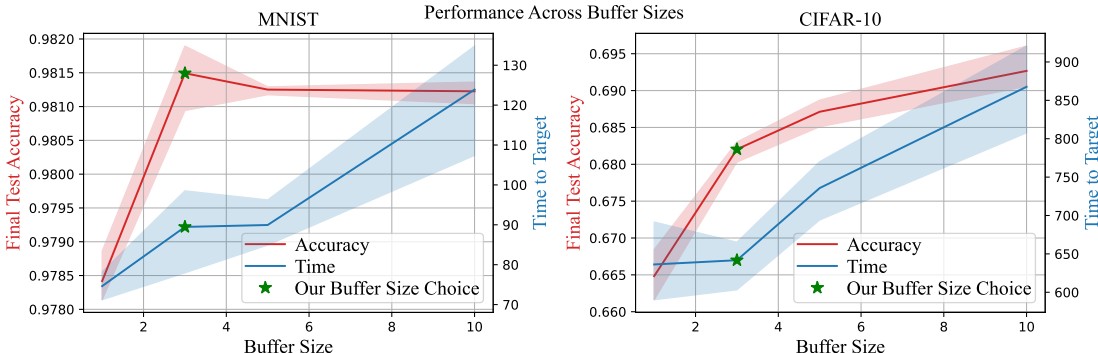

Figure 9: The final test accuracy and required time to get target accuracy (in Table 1) for simultaneous training (using `FedAST` with static *option*) of one MNIST and one CIFAR-10 model with different buffer sizes. We assign the same number of local training requests (105) to each task.

**Learning rate and weight decay.** We search for the best learning rate and weight decay hyperparameters considering the training speed and final accuracy levels. We seek client-side learning rate within the range of $[1 \times 10^{-3}, 1 \times 10]$, server-side learning rate within $[3 \times 10^{-2}, 3]$, and weight decays within $[1 \times 10^{-7}, 1 \times 10^{-2}]$. We observe that client-side learning rates of $6 \times 10^{-2}$ and 7 with weight decays of $3 \times 10^{-4}$ and $7 \times 10^{-5}$ work best respectively for Fashion-MNIST and Shakespeare tasks for all methods. For CIFAR-10 task, a client-side learning rate of $1 \times 10^{-1}$ with weight decays of $7 \times 10^{-4}$ and $3 \times 10^{-4}$ perform best for asynchronous and synchronous methods, respectively. For MNIST, we use client-side learning rates of $1 \times 10^{-1}$ and $2 \times 10^{-1}$ for asynchronous and synchronous methods, respectively, with a weight decay of $3 \times 10^{-4}$. For server-side learning rates, we observe that 1 for synchronous methods (`Sync-ST`, `Sync-Bayes-ST`, and `Sync-UCB-ST`), 0.1 for `FedAST`, and 0.038 for `FedAST-NoBuffer` perform well for all tasks.

## C.6   MODELING TRAINING TIMES, MODEL SIZES, AND CLIENT SPEED HETEROGENEITY

In our experiments, following [Lee et al., 2018, Shi et al., 2021, Zhou et al., 2022, Dutta et al., 2021], we employ the *shifted-exponential* random variables to model the duration between when the server sends a local training request to a client, and when it receives the update of the local training. The exponential component of the distribution reflects the stochastic nature of the device speeds, while the shift component accounts for unavoidable delays such as disk I/O operations.

Whenever a client $i$ performs local training for task $m$, we draw a random number from the distribution with a cumulative distribution function (CDF) of,

$$P(X \le x) = \begin{cases} 1 - \exp\{-\frac{x - \beta_{i,m}}{2\beta_{i,m}}\}, & x \ge \beta_{i,m} \\ 0, & \text{otherwise} \end{cases},$$

where $\beta_{i,m}$ depends on the speed of client $i$ and the size of the model associated with task $m$. Then, we multiply this random number by the number of local steps to calculate the simulation time between when the server requests for the local training, and when it receives the update back.

We quantify the effect of the model sizes based on the average time required to calculate one stochastic gradient for each model on the GPUs of our internal cluster. By our measurements, we set,

$$\frac{\beta_{i,\text{MNIST}}}{0.148} = \frac{\beta_{i,\text{Fashion-MNIST}}}{0.240} = \frac{\beta_{i,\text{CIFAR-10}}}{0.228} = \frac{\beta_{i,\text{Shakespeare}}}{0.555} = \frac{\beta_{i,\text{CIFAR-100}}}{2.071}, \quad \forall i \in [N].$$

In our experiments, we also take the heterogeneity in the speed of client devices into consideration. We categorize clients into three speed groups: slow ($\%25$), normal-speed ($\%50$), and fast ($\%25$). The speed rates for these categories are inversely proportional to $1.3$, $1$, and $0.7$, such that,

$$\frac{\beta_{\text{slow client},m}}{1.3} = \frac{\beta_{\text{normal-speed client},m}}{1} = \frac{\beta_{\text{fast client},m}}{0.7}, \quad \forall m \in [M].$$

# D    ADDITIONAL EXPERIMENTS

In this section, we present supplementary experiments.

## D.1    TUNING PARAMETER $k$ OF THE STRAGGLER MITIGATION TECHNIQUE USED FOR SYNCHRONOUS METHODS (ACCEPTING ONLY THE FIRST-$k$ UPDATES)

In our experiments, to mitigate the high straggler effect, the server in synchronous methods (`Sync-ST`, `Sync-Bayes-ST`, and `Sync-UCB-ST`) only aggregates the first $k$ client updates for each task and discards the rest, following [Bonawitz et al., 2019]. To tune parameter $k$, we run validation experiments with `Sync-ST` on single CIFAR-10, MNIST, and Fashion-MNIST tasks and evaluated the training performance with respect to simulated time and number of global rounds. A larger $k$ results in a longer simulated time per round since we wait for more clients. On the other hand, the variance in aggregated updates on each round becomes smaller since we average more updates. Therefore, the target accuracy is attained faster in terms of the number of global rounds. We also observed that keeping $k$ too small yields lower final accuracy. Navigating these trade-offs, we find that $k = 30$ strikes an effective balance.

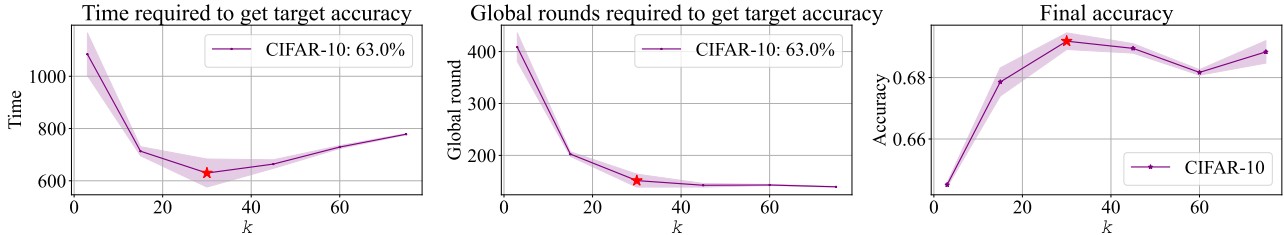

Figure 10: Performance of `Sync-ST` with varying $k$ in CIFAR-10 task. The chosen point is shown with a red star.

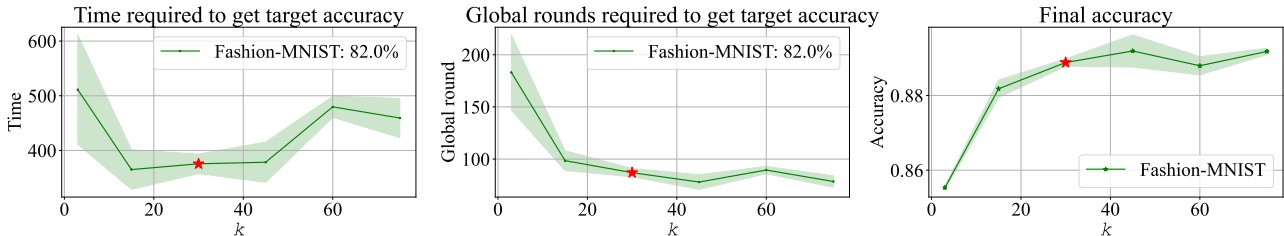

Figure 11: Performance of `Sync-ST` with varying $k$ in Fashion-MNIST task. The chosen point is shown with a red star.

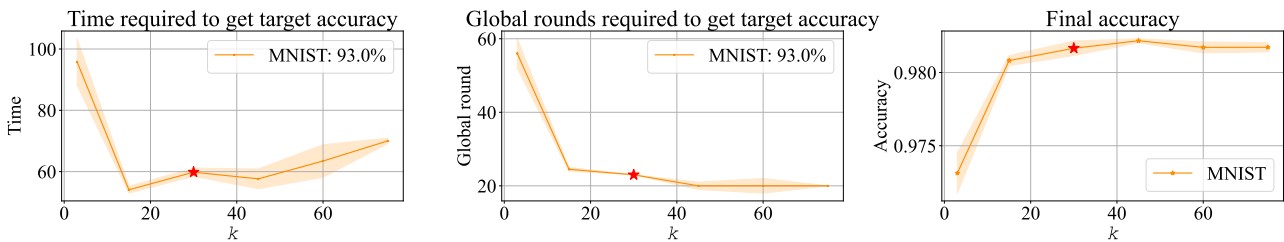

Figure 12: Performance of `Sync-ST` with varying $k$ in MNIST task. The chosen point is shown with a red star.

## D.2   TEST LOSS PLOTS OF FIGURES 3 AND 4 IN THE MAIN TEXT

We illustrate test loss plots of the experiments in Figures 3 and 4 in the main text.

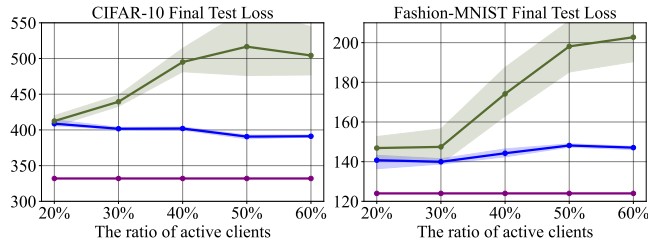

Figure 13: The mean final loss values of `FedAST` (blue), `FedAST-NoBuffer` (olive green) and centralized training (violet) with varying active client ratio, when training 3 identical models. The left figure is for CIFAR-10 dataset, while the right figure is for Fashion-MNIST dataset. With a higher number of active clients, thanks to the buffer, `FedAST` remains its performance while `FedAST-NoBuffer` gets worse.

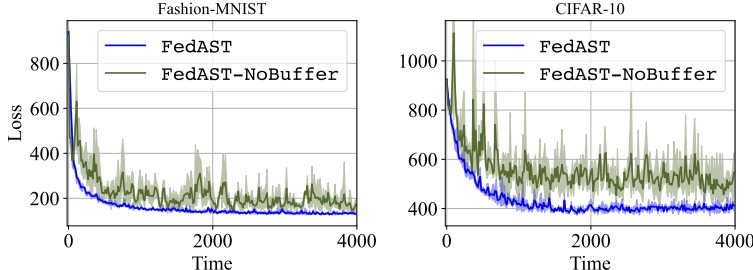

Figure 14: The mean test loss values of `FedAST` and `FedAST-NoBuffer`, when simultaneously training one model for CIFAR-10 and one for Fashion-MNIST. `FedAST` achieves lower and more stable loss levels.

## D.3   TRAINING CURVES OF HOMOGENEOUS EXPERIMENTS

In Figure 15, we provide the average training curves of the homogeneous-task experiment in Figure 5.

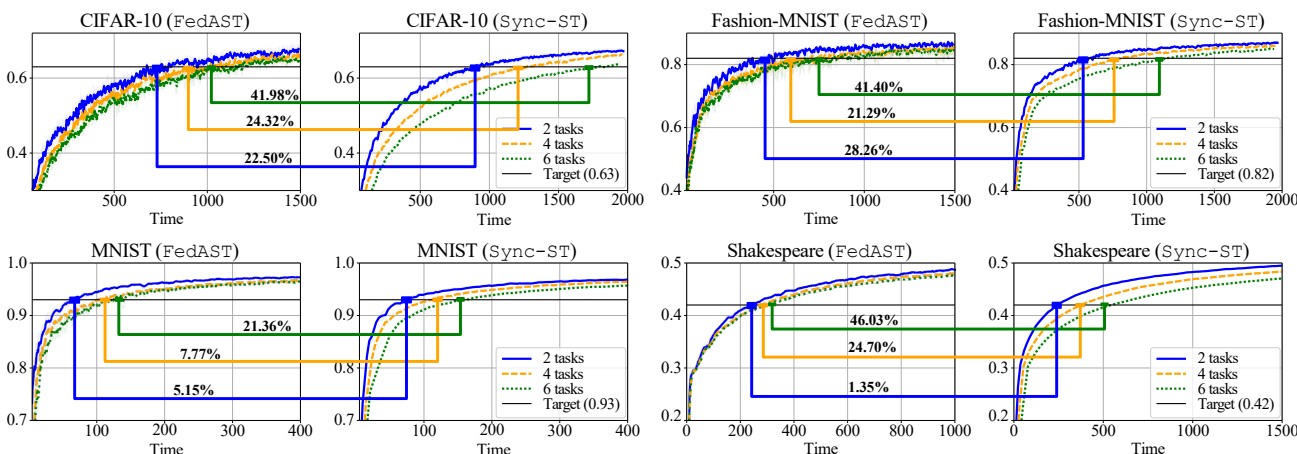

Figure 15: Training curves of FedAST and Sync-ST on 2/4/6 tasks with CIFAR-10, Fashion-MNIST, MNIST, and Shakespeare datasets. Time gains of FedAST over Sync-ST to attain target accuracy are shown on the colored horizontal lines. Horizontal black lines indicate target accuracy levels, same as the ones stated in Table 1.

### D.4 AN ADDITIONAL EXPERIMENT WITH A LARGER MODEL (RESNET-18) ON CIFAR-100

We run a homogeneous-task experiment with a larger model, ResNet-18, as implemented by Acar et al. [2021] on the CIFAR-100 dataset, a 100-class image classification dataset [Krizhevsky et al., 2009]. We use the same experimental settings as those in other experiments except for a few differences elaborated here. We use the Dirichlet distribution with $\alpha = 1$ to simulate heterogeneity following the approach suggested in Yurochkin et al. [2019]. We use a client-side learning rate ($\eta_s$) of 0.06 and the number of local SGD steps ($\tau$) of 5 for both FedAST and Sync-ST. We present the experimental results in Figure 16. Our experiments show that FedAST outperforms the synchronous baseline with a ResNet-18 model on the CIFAR-100 dataset by providing time gains of 22.0%, 40.7%, and 56.3% for 2, 4, and 6 simultaneously trained models respectively.

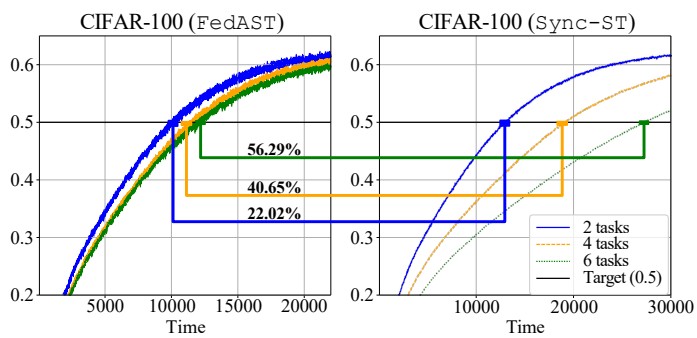

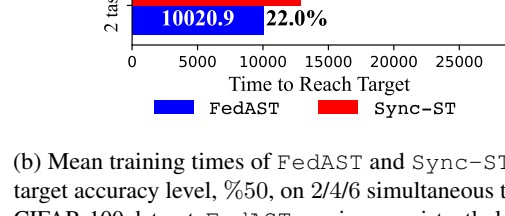

(a) Training curves of FedAST and Sync-ST on 2/4/6 simultaneous CIFAR-100 tasks. Time gains of FedAST over Sync-ST to attain target accuracy are shown on the colored horizontal lines. Horizontal black lines indicate the target accuracy level, %50.

(b) Mean training times of FedAST and Sync-ST to attain target accuracy level, %50, on 2/4/6 simultaneous tasks with CIFAR-100 dataset. FedAST requires consistently lower wall-clock time for training compared to Sync-ST; the percentages represent these time gains.

Figure 16: Experimental results on simultaneous repeated tasks with FedAST and Sync-ST on CIFAR-100.

### D.5 EXPERIMENTS WITH DIFFERENT TARGET ACCURACY LEVELS

To see how FedAST and the competitor Sync-ST work with different target accuracy, we conduct the experiment in Figure 7 with +3% higher and −10% lower target accuracy levels as presented in Table 3. We observe that proposed FedAST reduces the overall training time by 55.9% and 16.3%, respectively for higher and lower target accuracy levels.

We conclude that the advantage of `FedAST` over `Sync-ST` increases with the difficulty of the task (i.e., reaching higher accuracy).

Table 3: Different target accuracy levels used in experiments to validate the proposed methods, with lower and higher accuracy targets.

| Dataset | Lower Target Accuracy | Target Accuracy in the Main Text | Higher Target Accuracy |
|---|---|---|---|
| MNIST | 83% | 93% | 96% |
| Fashion-MNIST | 72% | 82% | 85% |
| CIFAR-10 | 53% | 63% | 66% |
| Shakespeare | 32% | 42% | 45% |

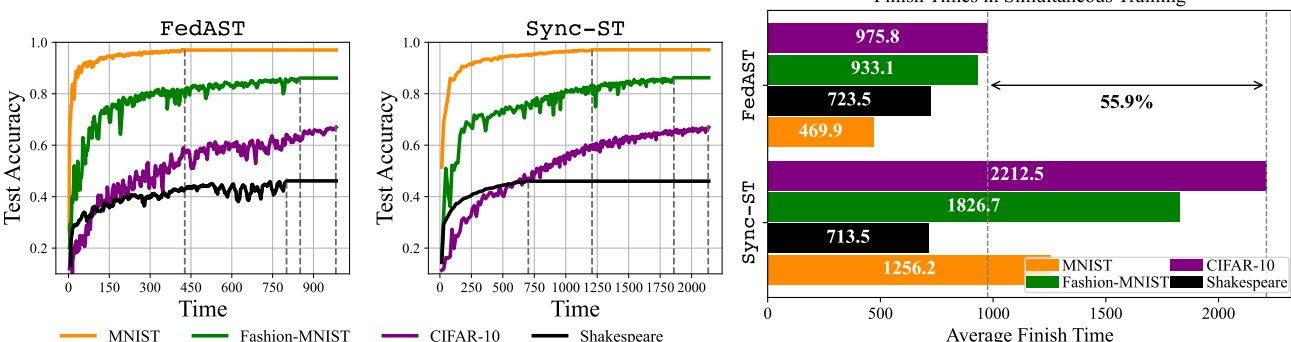

Figure 17: Training curves of a single Monte Carlo run of the *heterogeneous experiment with higher target accuracy levels* in Table 3. Dashed vertical lines show times when tasks reach their target accuracy. The setting is the same as the experiment in Figure 7.

Figure 18: Mean time required to reach target accuracy and time gain of `FedAST` over `Sync-ST` in *the heterogeneous experiment with higher target accuracy levels* in Table 3. The setting is the same as the experiment in Figure 7.

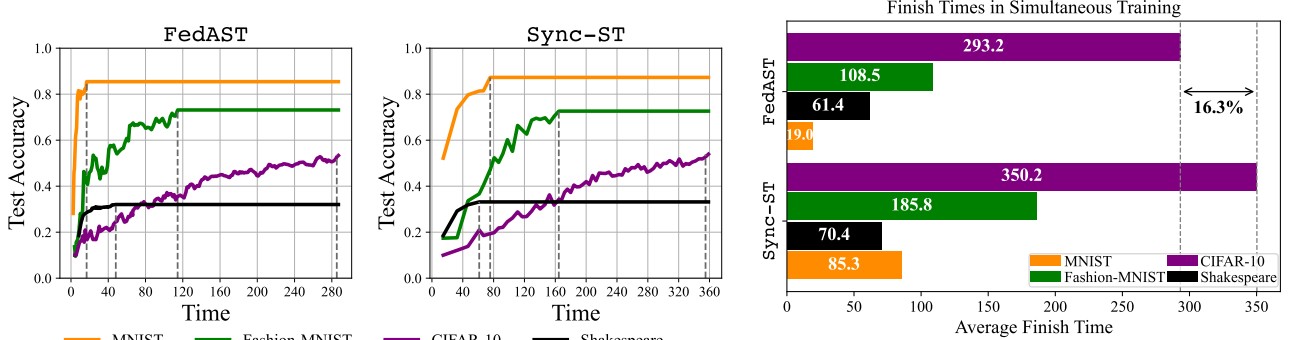

Figure 19: Training curves of a single Monte Carlo run of the *heterogeneous experiment with lower target accuracy levels* in Table 3. Dashed vertical lines show times when tasks reach their target accuracy. The setting is the same as the experiment in Figure 7.

Figure 20: Mean time required to reach target accuracy and time gain of `FedAST` over `Sync-ST` in *the heterogeneous experiment with lower target accuracy levels* in Table 3. The setting is the same as the experiment in Figure 7.

## D.6 PERFORMANCE OF `FedAST` WITHOUT STATIC RESOURCE ALLOCATION

We conduct heterogeneous-task experiments to validate the performance gain of dynamic resource allocation (`FedAST (D)`) over static option (`FedAST(S)`) with uniform allocation across tasks in heterogeneous settings. For uniform resource

allocation, we allocate the same number of active training requests to each task in `FedAST(S)`. To show the consistency of our results, we run experiments at all target accuracy levels in Table 3. We present the results in Figure 22 (higher target accuracy), Figure 24 (the target accuracy in the main text), and Figure 26 (lower target accuracy). We conclude that our dynamic client allocation based on the variance estimates of the updates reduces the total training time compared to the uniform static client allocation. The advantage of dynamic resource allocation becomes more prominent with more difficult tasks (i.e., higher target accuracy level).

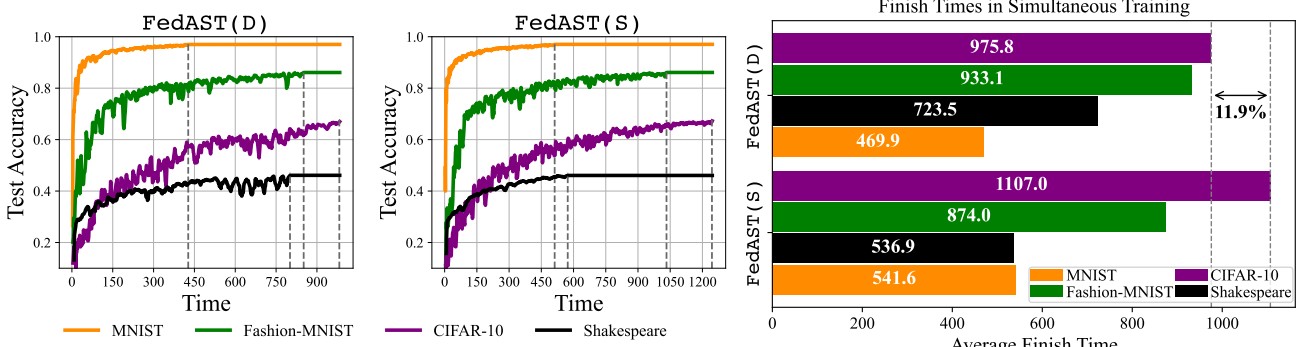

Figure 21: Training curves of a single Monte Carlo run in the experiment with dynamic resource allocation *option* (`FedAST(D)`) and static *option* with uniform resource allocation (`FedAST(S)`). The setting is the *heterogeneous experiment with higher target accuracy levels* in Table 3. Dashed vertical lines show times when tasks reach their target accuracy.

Figure 22: Mean training times required to reach target accuracy and time gain of dynamic resource allocation *option* (`FedAST(D)`) over static *option* with uniform resource allocation (`FedAST(S)`). The setting is the *heterogeneous experiment with higher target accuracy levels* in Table 3.

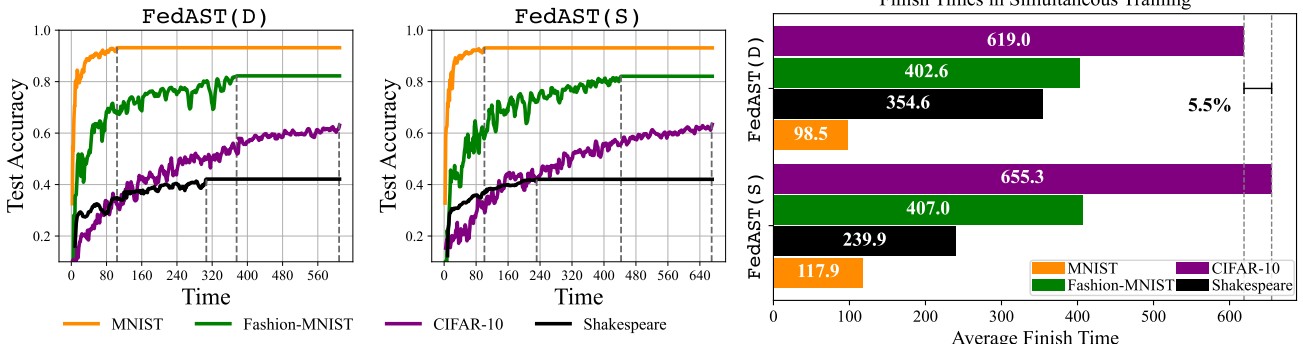

Figure 23: Training curves of a single Monte Carlo run in the experiment with dynamic resource allocation *option* (`FedAST(D)`) and static *option* with uniform resource allocation (`FedAST(S)`). The setting is the *heterogeneous experiment with the target accuracy levels used in the main text* in Table 3. Dashed vertical lines show times when tasks reach their target accuracy.

Figure 24: Mean training times required to reach target accuracy and time gain of dynamic resource allocation *option* (`FedAST(D)`) over static *option* with uniform resource allocation (`FedAST(S)`). The setting is the *heterogeneous experiment with the target accuracy levels used in the main text* in Table 3.

# E  PROOFS OF THE CONVERGENCE ANALYSIS OF `FedAST` WITH STATIC *OPTION* ($S$)

In this section, we present the proofs of the mathematical claims made in the paper. First, we define and explain the notations used in this section. After that, we introduce intermediate lemmas used in the main proof (Section E.2). Then, we present the proofs of Theorem 1 and Corollary 1.1 (Section E.3). Finally, we prove intermediate lemmas (Section E.4).

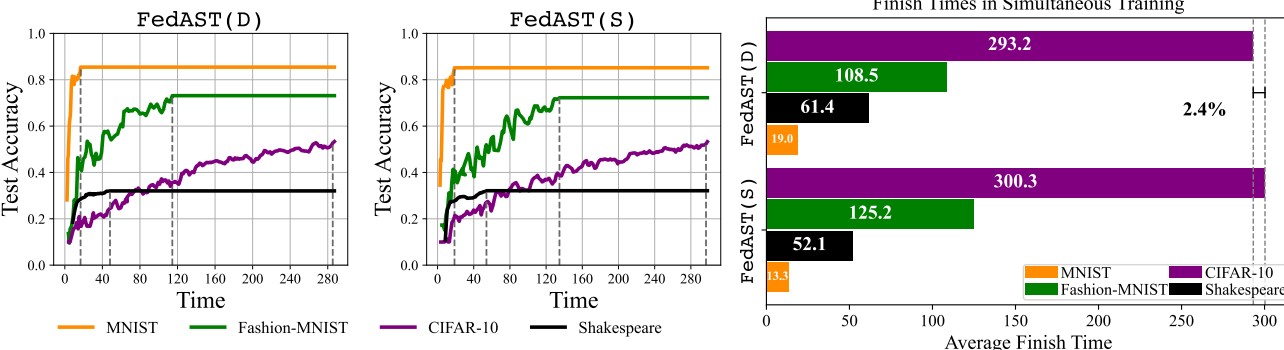

Figure 25: Training curves of a single Monte Carlo run in the experiment with dynamic resource allocation *option* (`FedAST(D)`) and static *option* with uniform resource allocation (`FedAST(S)`). The setting is the *heterogeneous experiment with lower target accuracy levels* in Table 3. Dashed vertical lines show times when tasks reach their target accuracy.

Figure 26: Mean training times required to reach target accuracy and time gain of dynamic resource allocation *option* (`FedAST(D)`) over static *option* with uniform resource allocation (`FedAST(S)`). The setting is the *heterogeneous experiment with lower target accuracy levels* in Table 3.

## E.1 NOTATIONS AND DEFINITIONS

`FedAST` enables us to divide the convergence analyses of simultaneous tasks into individual ones. We focus on the convergence analysis of a single task within a simultaneous multi-model setting and the analysis holds for all tasks trained together. For brevity, we provide the proofs for a single task of multiple models trained simultaneously. Therefore, we drop all model indices in our analysis. We also drop time indices from the number of active requests ($R$) and buffer size ($b$) terms as they remain the same during the training with the static option of `FedAST`. Table 4 summarizes all notation. Please note that the analysis presented here holds for every model $m \in [M]$ simultaneously trained within `FedAST` framework.

### E.1.1 The Update Rules of **FedAST**

We first revisit the local training and global update rules of `FedAST`. The notation may vary slightly from those in the main paper due to dropping model indices, but still accurately depicts the same algorithmic procedures, Algorithms 1 and 2 in the main text.

**Local update rule.** During local training, clients perform $\tau$ consecutive local stochastic gradient steps and return the output to the server. When a client receives the $t^{\text{th}}$ version of the global model, $\mathbf{x}^{(t)}$, it takes $\tau$ mini-batch stochastic gradient descent steps (for $k = 1, \ldots, \tau$) with following rule:

$$\mathbf{x}_i^{(t,k)} \leftarrow \mathbf{x}_i^{(t,k-1)} - \eta_c \widetilde{\nabla} f_i \left( \mathbf{x}_i^{(t,k-1)} \right), \tag{8}$$

where $\mathbf{x}_i^{(t,0)} \triangleq \mathbf{x}^{(t)}$ and $\widetilde{\nabla}$ denotes stochastic gradients. We define the average of local stochastic gradients as $\mathbf{\Delta}_i^{(t)} \triangleq \frac{1}{\tau} \sum_{k=0}^{\tau-1} \widetilde{\nabla} f_i \left( \mathbf{x}_i^{(t,k)} \right)$. Then, the client returns $\frac{\mathbf{x}^{(t)} - \mathbf{x}_i^{(t,\tau)}}{\tau \eta_c} = \frac{1}{\tau} \sum_{k=0}^{\tau-1} \widetilde{\nabla} f_i \left( \mathbf{x}_i^{(t,k)} \right) = \mathbf{\Delta}_i^{(t)}$ to the server. The server stores the updates in a buffer.

**Staleness.** The server receives the updates of local training requests asynchronously. It means that the received updates may come in a different order than local training requests sent to clients. Therefore, an aggregated update may have been calculated with an older version of the model, and this is called *staleness*. We quantify the staleness of an update in terms of the number of global rounds passed between the times when the server sends the local training request and receives the update. The staleness is random for each update, depending on client selections for all tasks and all clients' availability, computation, and communication speeds. We denote the staleness of client $i$'s update received at the server at the $t^{\text{th}}$ round as $\gamma_i^t$. Recall that Assumption 4 (Bounded Staleness) bounds this random value above at $\gamma^{\max}$.

**Global update rule.** On each global round $t$, when the buffer at the server, $\mathcal{B}^{(t)}$, is full ($|\mathcal{B}^{(t)}| = b$, where $b$ is the buffer size), the server aggregates the updates to proceed to the next global round. Here, $\mathcal{B}^{(t)}$ is the set of clients whose updates are received after $(t-1)^{\text{th}}$ and before $t^{\text{th}}$ aggregation. The aggregation rule can be written as follows:

$$\mathbf{x}^{(t+1)} \leftarrow \mathbf{x}^{(t)} - \tau\eta_s\eta_c\frac{1}{b}\sum_{i\in\mathcal{B}^{(t)}}\mathbf{\Delta}_i^{(t-\gamma_i^t)} = \mathbf{x}^{(t)} - \eta_s\frac{1}{b}\sum_{i\in\mathcal{B}^{(t)}}\left(\mathbf{x}^{(t-\gamma_i^t)} - \mathbf{x}_i^{(t-\gamma_i^t,\tau)}\right) \tag{9}$$

$$= \mathbf{x}^{(t)} - \eta_s\eta_c\frac{1}{b}\sum_{i\in\mathcal{B}^{(t)}}\sum_{k=0}^{\tau-1}\widetilde{\nabla}f_i\left(\mathbf{x}_i^{(t-\gamma_i^t,k)}\right).$$

### E.1.2 Virtual Sequence and Set Definitions

We utilize the perturbed iterate idea from Koloskova et al. [2022], Mania et al. [2017].

First, let us introduce some helpful sets and notations. Consider $\mathcal{A}^{(t)}$, which represents the set of clients chosen by the server to receive the $t^{\text{th}}$ version of the model. Recall that the server in FedAST selects the clients uniformly at random with replacement from all clients. The size of this set, $|\mathcal{A}^{(t)}|$, is always equal to the buffer size, $b$, (except initialization, $t = 0$) because $b$ new local training requests are made on each round. For instance, if $b$ is set to 3, and the server selects the 2nd, 16th, and 31st clients during the 4th aggregation round, then $\mathcal{A}^{(4)}$ is $\{2, 16, 31\}$. The server sends $\mathbf{x}^{(4)}$ to the 2nd, 16th, and 31st clients and requests local training with this model. In practical terms, $\mathcal{A}^{(t)}$ is a multiset, allowing multiple occurrences of the same client if a client is selected more than once. Throughout the proof, we consider each occurrence of the same client in multiset as a distinct update calculated on that particular client. While we acknowledge a slight abuse of notation, this does not lead to any mathematical flaw, and we believe that this significantly enhances the clarity and comprehensibility of the proof.

Now, let us define $\mathcal{C}^{(t)}$ as the set of clients that have incomplete local training requests at the time of the $t^{\text{th}}$ aggregation because of the asynchronous nature of FedAST. The size of this set, $|\mathcal{C}^{(t)}|$, is always equal to the number of active local training requests, $R$, because the server sends a new local training request for every update it receives. For instance, if $R$ is 4, and the server has sent local training requests to the 12th, 27th, 41st, and 55th clients prior to the 5th aggregation, yet these clients are still processing their updates, then $\mathcal{C}^{(5)}$ would be $\{12, 27, 41, 55\}$. Note that $\mathcal{C}^{(0)}$ is an empty set, as there are no active local training requests before the algorithm starts. It is worth noting that $\mathcal{C}^{(t)}$ is a multiset, allowing multiple occurrences of the same client if a client has more than one active local training request (recall that multiple requests are queued at the client side). Each occurrence of a client within this multiset represents a different local training calculated on that client. We again acknowledge a slight abuse of notation, but this does not lead to any mathematical flaw, and we believe that this makes the flow of proof significantly easier.

Next, we define the virtual sequence $\mathbf{z}^{(t)}$ for $t = 0, 1, \ldots, T$ as the model that receives local training updates of the global model, $\mathbf{x}^{(t)}$ for $t = 0, 1, \ldots, T$, in the correct order. Namely, unlike $\mathbf{x}^{(t)}$, $\mathbf{z}^{(t)}$ receives the local training updates in the order in which the server sends those requests. However, it is crucial to note that the local training updates are still calculated with the global model, $\mathbf{x}^{(t)}$. The update rule of the virtual sequence is:

$$\mathbf{z}^{(t+1)} \leftarrow \mathbf{z}^{(t)} - \tau\eta_s\eta_c\frac{1}{b}\sum_{i\in\mathcal{A}^{(t)}}\mathbf{\Delta}_i^{(t)} = \mathbf{z}^{(t)} - \tau\eta_s\eta_c\frac{1}{b}\sum_{i\in\mathcal{A}^{(t)}}\frac{1}{\tau}\sum_{k=0}^{\tau-1}\widetilde{\nabla}f_i\left(\mathbf{x}_i^{(t,k)}\right), \tag{10}$$

for $t = 0, 1, \ldots, T-1$ where $\mathbf{z}^{(0)} \triangleq \mathbf{x}^{(0)}$.

**Remark 1.** *Now, we state an observation using the definitions of $\mathcal{C}^{(t)}$, the virtual sequence, and the global model. When the $t^{\text{th}}$ aggregation happens at the server, the virtual sequence, $\mathbf{z}^{(t)}$, has received all the updates from all previous local training requests on rounds $0, 1, \ldots, t-1$. At the same time, the global model, $\mathbf{x}^{(t)}$, has received the same updates except for the updates of clients in $\mathcal{C}^{(t)}$. By the update rules in (9) and (10), note that each received update at the server contributes to the global model and virtual sequence equally. Therefore, we can express their difference as:*

$$\mathbf{z}^{(t)} - \mathbf{x}^{(t)} = -\tau\eta_s\eta_c\frac{1}{b}\sum_{i\in\mathcal{C}^{(t)}}\mathbf{\Delta}_i^{(t-\gamma_i^t)}. \tag{11}$$

**Remark 2.** *If we count the number of occurrences of the round index $y$ among the model versions of assigned updates in set $\mathcal{C}^{(t)}$ over all rounds $t = 0, \ldots, T-1$, we can bound this value as:*

$$\sum_{t=0}^{T-1} \sum_{i \in \mathcal{C}^{(t)}} \mathbf{1}\{t - \gamma_i^t = y\} \leq b\gamma^{max}, \quad \forall y = 0, \ldots, T-1, \tag{12}$$

*where $\mathbf{1}$ is an indicator function that returns $1$ if the statement is true, and returns $0$ otherwise. The reasoning for this observation is as follows. On each round, the server selects $b$ clients (since the server selects one new client for each received and buffered update where the buffer size is $b$) and sends them the up-to-date global model. We also know that all local training requests must be returned to the server within $\gamma^{max}$ rounds by Assumption 4 (Bounded Staleness). Therefore, over the rounds $t = 0, \ldots, T-1$, any round indices can appear at most $b\gamma^{max}$ times in the summation in the left-hand side of the inequality in (12). We will use this remark later in the proof.*

### E.1.3 Notation

We define some useful variables used in the proof and present the notation used in `FedAST` in Table 4. Also, we again want to remind the reader that we dropped all model indices in the proof as the theoretical results we present here hold for any of multiple tasks trained simultaneously, satisfying Assumptions 1 - 4.

Table 4: Summary of notations used in the mathematical analysis of `FedAST`.

| | |
|---|---|
| $f_i(\cdot)$: The loss function at client $i$ | $L$: Smoothness constant in Assumption 1 |
| $f(\cdot)$: The global loss function | $\tau$: Number of local SGD steps |
| $\mathbf{x}^{(t)}$: The global model at the $t^{\text{th}}$ round | $\eta_c$: Client-side learning rate |
| $\mathbf{x}_i^{(t,k)}$: The local model of client $i$ at the $k^{\text{th}}$ local step of the $t^{\text{th}}$ round | $\eta_s$: Server-side learning rate |
| $\mathbf{z}^{(t)}$: The virtual sequence at the $t^{\text{th}}$ round (Section E.1.2) | $b$: Buffer size |
| $\nabla, \widetilde{\nabla}$: Gradient and stochastic gradient operators | $\sigma_l^2$: Maximum local variance in Assumption 2 |
| $\mathbf{g}_i^{(t,k)} = \widetilde{\nabla} f_i\left(\mathbf{x}_i^{(t,k)}\right)$: Local stochastic gradient of client $i$ at round t and local step $k$ | $\sigma_g^2$: Maximum global variance in Assumption 3 |
| $\boldsymbol{\Delta}_i^{(t)} = \frac{1}{\tau} \sum_{k=0}^{\tau-1} \mathbf{g}_i^{(t,k)}$: The update of client $i$ at round t | $R$: Number of total active local training requests anytime |
| $\mathbf{h}_i^{(t)} = \mathbb{E}\left[\boldsymbol{\Delta}_i^{(t)}\right]$: The expected update of client $i$ at round t | $\gamma^{\max}$: Maximum staleness in Assumption 4 |
| $\widetilde{\eta}_s = \eta_s \tau$: Server learning rate multiplied by the number of local training steps | $\gamma_i^t$: The staleness of client $i$'s update at round $t$ |
| $\mathcal{A}^{(t)}$: The set of clients to which the server sends the $t^{\text{th}}$ version of the model (Section E.1.2) | $\mathcal{C}^{(t)}$: The set of clients which are requested local training, but have not returned their updates to the server yet (Section E.1.2) |

### E.2 INTERMEDIATE LEMMAS

We present intermediate lemmas used through the proof.

**Lemma 1.** For a set of $Q$ vectors, $\mathbf{u}_1, \ldots, \mathbf{u}_Q$, where $Q$ is a positive integer,

$$\left\| \sum_{q=1}^{Q} \mathbf{u}_q \right\|^2 \leq Q \sum_{q=1}^{Q} \|\mathbf{u}_q\|^2.$$

*Proof.* The lemma is a direct consequence of Jensen's inequality with a convex function $\|\cdot\|^2$ and uniform random distribution over the set of vectors $\mathbf{u}_1, \ldots, \mathbf{u}_Q$.

**Lemma 2.** Suppose that $f_i(\cdot)$ satisfies Assumption 1 (Smoothness) and Assumption 2 (Bounded Variance) for all $i \in [N]$, and assume that $\eta_c \leq \frac{1}{L\tau}$. Then the iterates of FedAST satisfy,

$$\mathbb{E}\left\|\nabla f_i\left(\mathbf{x}^{(t)}\right) - \mathbf{h}_i^{(t)}\right\|^2 \leq \frac{L^2 \eta_c^2 \tau}{2(1-D)} \sigma_l^2 + \frac{D}{1-D}\mathbb{E}\left\|\nabla f_i\left(\mathbf{x}^{(t)}\right)\right\|^2, \quad \forall i \in N,$$

where $D \triangleq L^2 \eta_c^2 \tau (\tau - 1)$.

Further, suppose Assumption 3 (Bounded Heterogeneity) holds. Then, the iterates of FedAST satisfy,

$$\frac{1}{N}\sum_{i=1}^{N}\mathbb{E}\left\|\mathbf{h}_i^{(t)} - \nabla f_i\left(\mathbf{x}^{(t)}\right)\right\|^2 \leq \frac{L^2 \eta_c^2 \tau}{2(1-D)} \sigma_l^2 + \frac{D}{1-D}\mathbb{E}\left\|\nabla f\left(\mathbf{x}^{(t)}\right)\right\|^2 + \frac{D}{1-D}\sigma_g^2.$$

**Remark 3.** *The true gradient at any client using the global model is close to the local update of that client.*

**Lemma 3.** The iterates of FedAST and defined virtual sequence satisfy,

$$T_1 \triangleq -\left\langle \nabla f\left(\mathbf{z}^{(t)}\right), \frac{1}{N}\sum_{i=1}^{N}\mathbf{h}_i^{(t)} \right\rangle \leq -\frac{1}{2}\left\|\nabla f\left(\mathbf{x}^{(t)}\right)\right\|^2 + \frac{1}{2}\left\|\nabla f\left(\mathbf{z}^{(t)}\right) - f\left(\mathbf{x}^{(t)}\right)\right\|^2 + \frac{1}{2}\left\|\sum_{i=1}^{N}\left(\mathbf{h}_i^{(t)} - \nabla f_i\left(\mathbf{x}^{(t)}\right)\right)\right\|^2.$$

**Lemma 4.** Suppose that $f_i(\cdot)$ satisfies Assumption 2 (Bounded Variance and Unbiased Stochastic Gradients) for all $i \in [N]$, then the iterates of FedAST satisfy,

$$T_2 \triangleq \mathbb{E}\left\|\frac{1}{b}\sum_{i \in \mathcal{A}^{(t)}}\boldsymbol{\Delta}_i^{(t)}\right\|^2 \leq \mathbb{E}\left\|\frac{1}{b}\sum_{i \in \mathcal{A}^{(t)}}\mathbf{h}_i^{(t)}\right\|^2 + \frac{\sigma_l^2}{\tau b}.$$

**Remark 4.** *The noisy global update due to stochastic gradients is close to the expected update calculated with full gradients. The buffer and multiple local steps are useful to reduce the variance due to local SGD steps.*

**Lemma 5.** The iterates of FedAST satisfy,

$$\mathbb{E}\left\|\frac{1}{b}\sum_{i \in \mathcal{A}^{(t)}}\left(\nabla f_i\left(\mathbf{x}^{(t)}\right) - \nabla f\left(\mathbf{x}^{(t)}\right)\right)\right\|^2 = \frac{1}{bN}\sum_{i=1}^{N}\mathbb{E}\left\|\nabla f_i\left(\mathbf{x}^{(t)}\right) - \nabla f\left(\mathbf{x}^{(t)}\right)\right\|^2.$$

Further, suppose Assumption 3 (Bounded Heterogeneity) holds. Then, the iterates of FedAST also satisfy,

$$T_3 \triangleq \mathbb{E}\left\|\frac{1}{b}\sum_{i \in \mathcal{A}^{(t)}}\mathbf{h}_i^{(t)}\right\|^2 \leq \frac{3}{N}\sum_{i=1}^{N}\mathbb{E}\left\|\mathbf{h}_i^{(t)} - \nabla f_i\left(\mathbf{x}^{(t)}\right)\right\|^2 + \frac{3\sigma_g^2}{b} + 3\mathbb{E}\left\|\nabla f\left(\mathbf{x}^{(t)}\right)\right\|^2.$$

**Remark 5.** *FedAST benefits the global variance reduction thanks to the buffer.*

**Lemma 6.** The virtual sequence and the iterates of FedAST satisfy,

$$\frac{1}{T}\sum_{t=0}^{T-1}\mathbb{E}\left\|\nabla f\left(\mathbf{z}^{(t)}\right) - \nabla f\left(\mathbf{x}^{(t)}\right)\right\|^2 \leq \left(1 + \frac{3RL^2\eta_c^2\tau^2}{2(1-D)}\right)\frac{L^2\widetilde{\eta}_s^2\eta_c^2 R}{b^2\tau}\sigma_l^2$$

$$+ \frac{1+D}{1-D}\frac{3L^2\widetilde{\eta}_s^2\eta_c^2 R^2}{b^2}\sigma_g^2 + \frac{3L^2\widetilde{\eta}_s^2\eta_c^2 R\gamma^{\max}}{b}\frac{1+D}{1-D}\frac{1}{T}\sum_{t=0}^{T-1}\mathbb{E}\left\|\nabla f\left(\mathbf{x}^{(t)}\right)\right\|^2.$$

**Remark 6.** *As discussed in Remark 1, although the virtual sequence and global model get updates in a different order, they receive the same updates. Therefore, we can bound their difference.*

## E.3  PROOFS OF MAIN STATEMENTS

We present and prove Theorem 1 and Corollary 1.1 here.

### E.3.1  Theorem 1 (Convergence bound)

First we restate the theorem:

**Theorem 1.** *(Convergence bound): Suppose Assumptions 1 - 4 hold, there are $R$ active local training requests, and the server and client learning rates, $\eta_s, \eta_c$ respectively, satisfy $\eta_s \leq \sqrt{\tau b}$ and $\eta_c \leq \min\left\{\frac{1}{6L\tau\sqrt{\tau b}}, \frac{1}{4L\tau\sqrt{\tau R\gamma^{max}}}\right\}$, where $b$ is the buffer size, and $\tau$ is the number of local training steps. Then, the iterations of Algorithm 2 (FedAST) satisfy:*

$$
\frac{1}{T}\sum_{t=0}^{T-1}\mathbb{E}\left\|\nabla f\left(\mathbf{x}^{(t)}\right)\right\|^2 \leq \mathcal{O}\left(\frac{f\left(\mathbf{x}^{(0)}\right) - \min_{\mathbf{x}} f\left(\mathbf{x}\right)}{T\eta_s\eta_c\tau}\right) + \mathcal{O}\left(\left(\frac{L\eta_s\eta_c}{b} + L^2\eta_c^2\tau + \frac{L^2\eta_s^2\eta_c^2\tau R}{b^2}\right)\sigma_l^2\right)
$$

$$
+ \mathcal{O}\left(\left(\frac{L\eta_s\eta_c\tau}{b} + L^2\eta_c^2\tau\left(\tau - 1\right) + \frac{L^2\eta_s^2\eta_c^2\tau^2 R^2}{b^2}\right)\sigma_g^2\right).
$$

*Proof.* Using the update rule of the virtual sequence (10) and Assumption (*Asm.*) 1 (Smoothness), and taking the conditional expectation with respect to $\mathbf{z}^{(t)}$, we have,

$$
\mathbb{E}\left[f\left(\mathbf{z}^{(t+1)}\right)\right] \leq f\left(\mathbf{z}^{(t)}\right) + \left\langle\nabla f\left(\mathbf{z}^{(t)}\right), \mathbb{E}\left[\mathbf{z}^{(t+1)} - \mathbf{z}^{(t)}\right]\right\rangle + \frac{L}{2}\mathbb{E}\left\|\mathbf{z}^{(t+1)} - \mathbf{z}^{(t)}\right\|^2
$$

$$
= f\left(\mathbf{z}^{(t)}\right) + \left\langle\nabla f\left(\mathbf{z}^{(t)}\right), \mathbb{E}\left[-\widetilde{\eta}_s\eta_c\frac{1}{b}\sum_{i\in\mathcal{A}^{(t)}}\boldsymbol{\Delta}_i^{(t)}\right]\right\rangle + \frac{L}{2}\mathbb{E}\left\|\widetilde{\eta}_s\eta_c\frac{1}{b}\sum_{i\in\mathcal{A}^{(t)}}\boldsymbol{\Delta}_i^{(t)}\right\|^2
$$

$$
\overset{Asm.\ 2}{=} f\left(\mathbf{z}^{(t)}\right) - \widetilde{\eta}_s\eta_c\frac{1}{b}\left\langle\nabla f\left(\mathbf{z}^{(t)}\right), \mathbb{E}\left[\sum_{i\in\mathcal{A}^{(t)}}\mathbf{h}_i^{(t)}\right]\right\rangle + \frac{L}{2}\widetilde{\eta}_s^2\eta_c^2\mathbb{E}\left\|\frac{1}{b}\sum_{i\in\mathcal{A}^{(t)}}\boldsymbol{\Delta}_i^{(t)}\right\|^2
$$

$$
\overset{\substack{Uniform\\client\\selection}}{=} f\left(\mathbf{z}^{(t)}\right) + \widetilde{\eta}_s\eta_c\mathbb{E}\left[\underbrace{-\left\langle\nabla f\left(\mathbf{z}^{(t)}\right), \frac{1}{N}\sum_{i=1}^{N}\mathbf{h}_i^{(t)}\right\rangle}_{\triangleq T_1}\right] + \frac{L}{2}\widetilde{\eta}_s^2\eta_c^2\underbrace{\mathbb{E}\left\|\frac{1}{b}\sum_{i\in\mathcal{A}^{(t)}}\boldsymbol{\Delta}_i^{(t)}\right\|^2}_{\triangleq T_2}.
$$

Using Lemmas 3 and 4, we can bound $T_1$ and $T_2$. Then, dividing both sides by $\widetilde{\eta}_s\eta_c$:

$$
\frac{\mathbb{E}\left[f\left(\mathbf{z}^{(t+1)}\right)\right] - f\left(\mathbf{z}^{(t)}\right)}{\widetilde{\eta}_s\eta_c} \leq -\frac{1}{2}\mathbb{E}\left\|\nabla f\left(\mathbf{x}^{(t)}\right)\right\|^2 + \frac{1}{2}\mathbb{E}\left\|\nabla f\left(\mathbf{z}^{(t)}\right) - \nabla f\left(\mathbf{x}^{(t)}\right)\right\|^2
$$

$$
+ \frac{1}{2N}\sum_{i=1}^{N}\left(\mathbb{E}\left\|\mathbf{h}_i^{(t)} - \nabla f_i\left(\mathbf{x}^{(t)}\right)\right\|^2\right) + \frac{L\widetilde{\eta}_s\eta_c}{2}\underbrace{\mathbb{E}\left\|\frac{1}{b}\sum_{i\in\mathcal{A}^{(t)}}\mathbf{h}_i^{(t)}\right\|^2}_{\triangleq T_3} + \frac{L\widetilde{\eta}_s\eta_c}{2}\frac{\sigma_l^2}{\tau b}.
$$

Using Lemma 5, we can bound $T_3$:

$$
\frac{\mathbb{E}\left[f\left(\mathbf{z}^{(t+1)}\right)\right] - f\left(\mathbf{z}^{(t)}\right)}{\widetilde{\eta}_s\eta_c}
$$

$$
\leq -\frac{1}{2}\mathbb{E}\left\|\nabla f\left(\mathbf{x}^{(t)}\right)\right\|^2 + \frac{1}{2}\mathbb{E}\left\|\nabla f\left(\mathbf{z}^{(t)}\right) - \nabla f\left(\mathbf{x}^{(t)}\right)\right\|^2 + \frac{1}{2N}\sum_{i=1}^{N}\left(\mathbb{E}\left\|\mathbf{h}_i^{(t)} - \nabla f_i\left(\mathbf{x}^{(t)}\right)\right\|^2\right)
$$

$$
+ L\widetilde{\eta}_s\eta_c\left(\frac{3}{2N}\sum_{i=1}^{N}\mathbb{E}\left\|\mathbf{h}_i^{(t)} - \nabla f_i\left(\mathbf{x}^{(t)}\right)\right\|^2 + \frac{3\sigma_g^2}{2b} + \frac{3}{2}\mathbb{E}\left\|\nabla f\left(\mathbf{x}^{(t)}\right)\right\|^2 + \frac{\sigma_l^2}{2\tau b}\right)
$$

$$
= \left( -\frac{1}{2} + \frac{3L\widetilde{\eta}_s\eta_c}{2} \right) \mathbb{E} \left\| \nabla f\left(\mathbf{x}^{(t)}\right) \right\|^2 + \frac{1}{2}\mathbb{E}\left\| \nabla f\left(\mathbf{z}^{(t)}\right) - \nabla f\left(\mathbf{x}^{(t)}\right) \right\|^2 + L\widetilde{\eta}_s\eta_c \left( \frac{3\sigma_g^2}{2b} + \frac{\sigma_l^2}{2\tau b} \right)
$$

$$
+ \left( \frac{3L\widetilde{\eta}_s\eta_c}{2} + \frac{1}{2} \right) \frac{1}{N} \sum_{i=1}^{N} \mathbb{E} \left\| \mathbf{h}_i^{(t)} - \nabla f_i\left(\mathbf{x}^{(t)}\right) \right\|^2
$$

$$
\overset{Lemma\ 2}{\leq} \left( -\frac{1}{2} + \frac{3L\widetilde{\eta}_s\eta_c}{2} \right) \mathbb{E} \left\| \nabla f\left(\mathbf{x}^{(t)}\right) \right\|^2 + \frac{1}{2}\mathbb{E}\left\| \nabla f\left(\mathbf{z}^{(t)}\right) - \nabla f\left(\mathbf{x}^{(t)}\right) \right\|^2 + L\widetilde{\eta}_s\eta_c \left( \frac{3\sigma_g^2}{2b} + \frac{\sigma_l^2}{2\tau b} \right)
$$

$$
+ \left( \frac{3L\widetilde{\eta}_s\eta_c}{2} + \frac{1}{2} \right) \left( \frac{L^2\eta_c^2\tau}{2(1-D)}\sigma_l^2 + \frac{D}{1-D}\mathbb{E}\left\| \nabla f\left(\mathbf{x}^{(t)}\right) \right\|^2 + \frac{D}{1-D}\sigma_g^2 \right)
$$

$$
= \left( -\frac{1}{2} + \frac{3L\widetilde{\eta}_s\eta_c}{2} + \frac{D}{2(1-D)} + \frac{3L\widetilde{\eta}_s\eta_c D}{2(1-D)} \right) \mathbb{E}\left\| \nabla f\left(\mathbf{x}^{(t)}\right) \right\|^2 + \frac{1}{2}\mathbb{E}\left\| \nabla f\left(\mathbf{z}^{(t)}\right) - \nabla f\left(\mathbf{x}^{(t)}\right) \right\|^2
$$

$$
+ \left( \frac{L\widetilde{\eta}_s\eta_c}{2\tau b} + \frac{3L^3\eta_c^3\widetilde{\eta}_s\tau}{4(1-D)} + \frac{L^2\eta_c^2\tau}{4(1-D)} \right)\sigma_l^2 + \left( \frac{3L\widetilde{\eta}_s\eta_c}{2b} + \frac{3L\widetilde{\eta}_s\eta_c D}{2(1-D)} + \frac{D}{2(1-D)} \right)\sigma_g^2,
$$

where $D \triangleq L^2\eta_c^2\tau(\tau-1)$. Using the tower property of conditional expectation, telescoping the inequality over the round indices $t = 0, 1, \ldots, T-1$, and using Lemma 6, we get,

$$
\frac{1}{T}\sum_{t=0}^{T-1}\left( \frac{1}{2} - \frac{3L\widetilde{\eta}_s\eta_c}{2} - \frac{D}{2(1-D)} - \frac{3L\widetilde{\eta}_s\eta_c D}{2(1-D)} \right)\mathbb{E}\left\| \nabla f\left(\mathbf{x}^{(t)}\right) \right\|^2 \leq \frac{1}{2T}\sum_{t=0}^{T-1}\mathbb{E}\left\| \nabla f\left(\mathbf{z}^{(t)}\right) - \nabla f\left(\mathbf{x}^{(t)}\right) \right\|^2
$$

$$
+ \frac{f\left(\mathbf{z}^{(0)}\right) - \mathbb{E}\left[f\left(\mathbf{z}^{(T)}\right)\right]}{T\widetilde{\eta}_s\eta_c} + \left( \frac{L\widetilde{\eta}_s\eta_c}{2\tau b} + \frac{3L^3\eta_c^3\widetilde{\eta}_s\tau}{4(1-D)} + \frac{L^2\eta_c^2\tau}{4(1-D)} \right)\sigma_l^2 + \left( \frac{3L\widetilde{\eta}_s\eta_c}{2b} + \frac{3L\widetilde{\eta}_s\eta_c D}{2(1-D)} + \frac{D}{2(1-D)} \right)\sigma_g^2
$$

$$
\overset{Lemma\ 6}{\leq} \left( 1 + \frac{3RL^2\eta_c^2\tau^2}{2(1-D)} \right)\frac{L^2\widetilde{\eta}_s^2\eta_c^2 R}{2b^2\tau}\sigma_l^2 + \frac{1+D}{1-D}\frac{3L^2\widetilde{\eta}_s^2\eta_c^2 R^2}{2b^2}\sigma_g^2 + \frac{3L^2\widetilde{\eta}_s^2\eta_c^2 R\gamma^{\max}}{2b}\frac{1+D}{1-D}\frac{1}{T}\sum_{t=0}^{T-1}\mathbb{E}\left\| \nabla f\left(\mathbf{x}^{(t)}\right) \right\|^2
$$

$$
+ \frac{f\left(\mathbf{z}^{(0)}\right) - \mathbb{E}\left[f\left(\mathbf{z}^{(T)}\right)\right]}{T\widetilde{\eta}_s\eta_c} + \left( \frac{L\widetilde{\eta}_s\eta_c}{2\tau b} + \frac{3L^3\eta_c^3\widetilde{\eta}_s\tau}{4(1-D)} + \frac{L^2\eta_c^2\tau}{4(1-D)} \right)\sigma_l^2 + \left( \frac{3L\widetilde{\eta}_s\eta_c}{2b} + \frac{3L\widetilde{\eta}_s\eta_c D}{2(1-D)} + \frac{D}{2(1-D)} \right)\sigma_g^2.
$$

Suppose the learning rates satisfy $\eta_s \leq \sqrt{\tau b}$ (which also makes $\widetilde{\eta}_s \leq \tau\sqrt{\tau b}$) and $\eta_c \leq \min\left\{ \frac{1}{6L\tau\sqrt{\tau b}}, \frac{1}{4L\tau\sqrt{\tau R\gamma^{\max}}} \right\}$, the following inequality holds:

$$
\frac{1}{2} - \frac{3L\widetilde{\eta}_s\eta_c}{2} - \frac{D}{2(1-D)} - \frac{3L\widetilde{\eta}_s\eta_c D}{2(1-D)} - \frac{3L^2\widetilde{\eta}_s^2\eta_c^2 R\gamma^{\max}}{2b}\frac{1+D}{1-D} \geq \frac{1}{11}. \tag{13}
$$

Also, notice that $\mathbf{z}^{(0)}$ is equal to $\mathbf{x}^{(0)}$ by definitions (Section E.1.2) of these sequences and $\min_{\mathbf{x}} f(\mathbf{x}) \leq f\left(\mathbf{z}^{(T)}\right)$.

$$
\frac{1}{T}\sum_{t=0}^{T-1}\mathbb{E}\left\| \nabla f\left(\mathbf{x}^{(t)}\right) \right\|^2 \leq 11\frac{f\left(\mathbf{x}^{(0)}\right) - \min_{\mathbf{x}} f(\mathbf{x})}{T\widetilde{\eta}_s\eta_c} \tag{Using (13)}
$$

$$
+ 11\left( \frac{L\widetilde{\eta}_s\eta_c}{2\tau b} + \frac{3L^3\eta_c^3\widetilde{\eta}_s\tau}{4(1-D)} + \frac{L^2\eta_c^2\tau}{4(1-D)} + \left( 1 + \frac{3RL^2\eta_c^2\tau^2}{2(1-D)} \right)\frac{L^2\widetilde{\eta}_s^2\eta_c^2 R}{2b^2\tau} \right)\sigma_l^2
$$

$$
+ 11\left( \frac{3L\widetilde{\eta}_s\eta_c}{2b} + \frac{3L\widetilde{\eta}_s\eta_c D}{2(1-D)} + \frac{D}{2(1-D)} + \frac{1+D}{1-D}\frac{3L^2\widetilde{\eta}_s^2\eta_c^2 R^2}{2b^2} \right)\sigma_g^2.
$$

Define $\delta \triangleq f\left(\mathbf{x}^{(0)}\right) - \min_{\mathbf{x}} f(\mathbf{x})$. After reducing high-order terms using the assumptions, $\eta_s \leq \sqrt{\tau b}$ (which also makes $\widetilde{\eta}_s \leq \tau\sqrt{\tau b}$) and $\eta_c \leq \min\left\{ \frac{1}{6L\tau\sqrt{\tau b}}, \frac{1}{4L\tau\sqrt{\tau R\gamma^{\max}}} \right\}$, and incorporating the constants into the $\mathcal{O}(\cdot)$ notation, we have:

$$
\frac{1}{T}\sum_{t=0}^{T-1}\mathbb{E}\left\| \nabla f\left(\mathbf{x}^{(t)}\right) \right\|^2 \leq \mathcal{O}\left( \frac{\delta}{T\eta_s\eta_c\tau} \right) + \mathcal{O}\left( \left( \frac{L\eta_s\eta_c}{b} + L^2\eta_c^2\tau + \frac{L^2\eta_s^2\eta_c^2\tau R}{b^2} \right)\sigma_l^2 \right)
$$

$$
+ \mathcal{O}\left( \left( \frac{L\eta_s\eta_c\tau}{b} + L^2\eta_c^2\tau(\tau-1) + \frac{L^2\eta_s^2\eta_c^2\tau^2 R^2}{b^2} \right)\sigma_g^2 \right).
$$

This concludes the proof.

### E.3.2 Proof of Corollary 1 (Convergence Rate)

First, notice that learning rates, $\eta_s = \sqrt{\tau b}$ and $\eta_c = \min\left\{\frac{1}{\tau L\sqrt{T}}, \frac{1}{6L\tau\sqrt{\tau b}}, \frac{1}{4L\tau\sqrt{\tau R\gamma^{\max}}}\right\}$ satisfy the assumptions ($\eta_s \leq \sqrt{\tau b}$ and $\eta_c \leq \min\left\{\frac{1}{6L\tau\sqrt{\tau b}}, \frac{1}{4L\tau\sqrt{\tau R\gamma^{\max}}}\right\}$) used through the proof.

When $T \geq \max\{36b\tau, 16\tau R\gamma^{\max}\}$; set learning rates $\eta_s = \sqrt{\tau b}$ and $\eta_c = \frac{1}{\tau L\sqrt{T}}$. Then, the bound in Theorem 1 reduces to:

$$\frac{1}{T}\sum_{t=0}^{T-1}\mathbb{E}\left\|\nabla f\left(\mathbf{x}^{(t)}\right)\right\|^2 \leq \mathcal{O}\left(\frac{L}{\sqrt{Tb\tau}}\right)\delta + \mathcal{O}\left(\frac{1}{\sqrt{Tb\tau}} + \frac{1}{\tau T} + \frac{R}{Tb}\right)\sigma_l^2 + \mathcal{O}\left(\sqrt{\frac{\tau}{Tb}} + \frac{1}{T} + \frac{\tau R^2}{Tb}\right)\sigma_g^2.$$

### E.4 PROOFS OF INTERMEDIATE LEMMAS

*Proof of Lemma 2.* We borrow the proof technique from [Wang et al., 2020b, C.5].

$$\mathbb{E}\left\|\nabla f_i\left(\mathbf{x}^{(t)}\right) - \mathbf{h}_i^{(t)}\right\|^2 = \mathbb{E}\left\|\nabla f_i\left(\mathbf{x}^{(t)}\right) - \frac{1}{\tau}\sum_{k=0}^{\tau-1}\nabla f_i\left(\mathbf{x}_i^{(t,k)}\right)\right\|^2$$

$$\stackrel{\text{Lemma 1}}{\leq} \frac{1}{\tau}\sum_{k=1}^{\tau-1}\mathbb{E}\left\|\nabla f_i\left(\mathbf{x}^{(t)}\right) - \nabla f_i\left(\mathbf{x}_i^{(t,k)}\right)\right\|^2$$

$$\stackrel{\text{Asm. 1}}{\leq} \frac{L^2}{\tau}\underbrace{\sum_{k=1}^{\tau-1}\mathbb{E}\left\|\mathbf{x}^{(t)} - \mathbf{x}_i^{(t,k)}\right\|^2}_{\triangleq T_{recursive}} = \frac{L^2\eta_c^2}{\tau}\sum_{k=1}^{\tau-1}\mathbb{E}\left\|\sum_{v=0}^{k-1}\mathbf{g}_i^{(t,v)}\right\|^2 \tag{14}$$

$$\stackrel{\substack{\text{Lemma 2 in}\\ \text{[Wang et al., 2020b]}}}{=} \frac{L^2\eta_c^2}{\tau}\sum_{k=1}^{\tau-1}\left(\sum_{v=0}^{k-1}\mathbb{E}\left\|\mathbf{g}_i^{(t,v)} - \nabla f_i\left(\mathbf{x}_i^{(t,v)}\right)\right\|^2 + \mathbb{E}\left\|\sum_{v=0}^{k-1}\nabla f_i\left(\mathbf{x}_i^{(t,v)}\right)\right\|^2\right) \quad \text{(Using Assumption 2)}$$

$$\stackrel{\text{Lemma 1}}{\leq} \frac{L^2\eta_c^2}{\tau}\sum_{k=1}^{\tau-1}\sum_{v=0}^{k-1}\left(\mathbb{E}\left\|\mathbf{g}_i^{(t,v)} - \nabla f_i\left(\mathbf{x}_i^{(t,v)}\right)\right\|^2 + k\mathbb{E}\left\|\nabla f_i\left(\mathbf{x}_i^{(t,v)}\right)\right\|^2\right)$$

$$\stackrel{\text{Asm. 2}}{\leq} \frac{L^2\eta_c^2}{\tau}\sum_{k=1}^{\tau-1}\left(k\sigma_l^2 + k\sum_{v=0}^{k-1}\mathbb{E}\left\|\nabla f_i\left(\mathbf{x}_i^{(t,v)}\right)\right\|^2\right)$$

$$\leq \frac{L^2\eta_c^2}{\tau}\left(\frac{(\tau-1)\tau}{2}\sigma_l^2 + \frac{(\tau-1)\tau}{2}\sum_{k=0}^{\tau-2}\mathbb{E}\left\|\nabla f_i\left(\mathbf{x}_i^{(t,k)}\right)\right\|^2\right)$$

$$\leq \eta_c^2 L^2\frac{\tau-1}{2}\left(\sigma_l^2 + \sum_{k=0}^{\tau-2}\mathbb{E}\left\|\nabla f_i\left(\mathbf{x}_i^{(t,k)}\right)\right\|^2\right)$$

$$\leq \eta_c^2 L^2\frac{\tau-1}{2}\left(\sigma_l^2 + \sum_{k=0}^{\tau-2}\left(2\mathbb{E}\left\|\nabla f_i\left(\mathbf{x}_i^{(t,k)}\right) - \nabla f_i\left(\mathbf{x}^{(t)}\right)\right\|^2 + 2\mathbb{E}\left\|\nabla f_i\left(\mathbf{x}^{(t)}\right)\right\|^2\right)\right)$$

$$\stackrel{\text{Asm. 1}}{\leq} \eta_c^2 L^2\frac{\tau-1}{2}\left(\sigma_l^2 + \sum_{k=0}^{\tau-2}\left(2L^2\mathbb{E}\left\|\mathbf{x}_i^{(t,k)} - \mathbf{x}^{(t)}\right\|^2 + 2\mathbb{E}\left\|\nabla f_i\left(\mathbf{x}^{(t)}\right)\right\|^2\right)\right)$$

$$\leq \eta_c^2 L^2\frac{\tau-1}{2}\left(\sigma_l^2 + \sum_{k=1}^{\tau-1}\left(2L^2\mathbb{E}\left\|\mathbf{x}_i^{(t,k)} - \mathbf{x}^{(t)}\right\|^2 + 2\mathbb{E}\left\|\nabla f_i\left(\mathbf{x}^{(t)}\right)\right\|^2\right)\right)$$

$$\leq \eta_c^2 L^2\frac{\tau-1}{2}\left(\sigma_l^2 + 2L^2 T_{recursive} + 2\tau\mathbb{E}\left\|\nabla f_i\left(\mathbf{x}^{(t)}\right)\right\|^2\right). \tag{15}$$

Using the recursive appearances of $T_{recursive}$ in (14) and (15):

$$\frac{T_{recursive}}{\tau} = \frac{1}{\tau}\sum_{k=1}^{\tau-1}\mathbb{E}\left\|\mathbf{x}_i^{(t,k)} - \mathbf{x}^{(t)}\right\|^2 \leq \eta_c^2\frac{\tau-1}{2}\sigma_l^2 + \eta_c^2\tau(\tau-1)\mathbb{E}\left\|f_i\left(\mathbf{x}^{(t)}\right)\right\|^2 + \eta_c^2 L^2(\tau-1)T_{recursive}.$$

Arranging the terms, defining $D \triangleq L^2 \eta_c^2 \tau (\tau - 1)$, and assuming $\eta_c \leq \frac{1}{L\tau}$ which makes $D \leq 1$,

$$\mathbb{E} \left\| \nabla f_i \left( \mathbf{x}^{(t)} \right) - \mathbf{h}_i^{(t)} \right\|^2 \leq \frac{L^2 T_{recursive}}{\tau} \leq \frac{L^2 \eta_c^2 (\tau - 1) \sigma_l^2 / 2 + L^2 \eta_c^2 \tau (\tau - 1) \mathbb{E} \left\| \nabla f_i \left( \mathbf{x}^{(t)} \right) \right\|^2}{1 - L^2 \eta_c^2 \tau (\tau - 1)}$$

$$\leq \frac{L^2 \eta_c^2 \tau}{2(1 - D)} \sigma_l^2 + \frac{D}{1 - D} \mathbb{E} \left\| f_i \left( \mathbf{x}^{(t)} \right) \right\|^2, \quad \forall i \in N.$$

This proves the first part of Lemma 2. Now, averaging it across clients:

$$\frac{1}{N} \sum_{i=1}^{N} \mathbb{E} \left\| \mathbf{h}_i^{(t)} - \nabla f_i \left( \mathbf{x}^{(t)} \right) \right\|^2 \leq \frac{1}{N} \sum_{i=1}^{N} \left( \frac{L^2 \eta_c^2 \tau}{2(1 - D)} \sigma_l^2 + \frac{D}{1 - D} \mathbb{E} \left\| f_i \left( \mathbf{x}^{(t)} \right) \right\|^2 \right)$$

$$= \frac{L^2 \eta_c^2 \tau}{2(1 - D)} \sigma_l^2 + \frac{D}{1 - D} \frac{1}{N} \sum_{i=1}^{N} \mathbb{E} \left\| \nabla f_i \left( \mathbf{x}^{(t)} \right) - \nabla f \left( \mathbf{x}^{(t)} \right) + \nabla f \left( \mathbf{x}^{(t)} \right) \right\|^2$$

$$= \frac{L^2 \eta_c^2 \tau}{2(1 - D)} \sigma_l^2 + \frac{D}{1 - D} \frac{1}{N} \sum_{i=1}^{N} \mathbb{E} \left\| \nabla f_i \left( \mathbf{x}^{(t)} \right) - \nabla f \left( \mathbf{x}^{(t)} \right) \right\|^2$$

$$+ \frac{D}{1 - D} \mathbb{E} \left\| \nabla f \left( \mathbf{x}^{(t)} \right) \right\|^2 + \frac{D}{1 - D} \frac{2}{N} \sum_{i=1}^{N} \left\langle \nabla f_i \left( \mathbf{x}^{(t)} \right) - \nabla f \left( \mathbf{x}^{(t)} \right), \nabla f \left( \mathbf{x}^{(t)} \right) \right\rangle$$

$$\overset{Asm. 3}{\leq} \frac{L^2 \eta_c^2 \tau}{2(1 - D)} \sigma_l^2 + \frac{D}{1 - D} \mathbb{E} \left\| \nabla f \left( \mathbf{x}^{(t)} \right) \right\|^2 + \frac{D}{1 - D} \sigma_g^2. \qquad \text{(Since } \frac{1}{N} \sum_{i=1}^{N} \nabla f_i \left( \mathbf{x}^{(t)} \right) = \nabla f \left( \mathbf{x}^{(t)} \right) \text{)}$$

This concludes the proof of Lemma 2.

*Proof of Lemma 3.*

$$T_1 \triangleq - \left\langle \nabla f \left( \mathbf{z}^{(t)} \right), \frac{1}{N} \sum_{i=1}^{N} \mathbf{h}_i^{(t)} \right\rangle = - \left\langle \nabla f \left( \mathbf{z}^{(t)} \right), \frac{1}{N} \sum_{i=1}^{N} \left( \mathbf{h}_i^{(t)} - \nabla f_i \left( \mathbf{x}^{(t)} \right) + \nabla f \left( \mathbf{x}^{(t)} \right) \right) \right\rangle$$

$$= - \left\langle \nabla f \left( \mathbf{z}^{(t)} \right), \nabla f \left( \mathbf{x}^{(t)} \right) \right\rangle - \left\langle \nabla f \left( \mathbf{z}^{(t)} \right), \frac{1}{N} \sum_{i=1}^{N} \left( \mathbf{h}_i^{(t)} - \nabla f_i \left( \mathbf{x}^{(t)} \right) \right) \right\rangle$$

$$= -\frac{1}{2} \left\| \nabla f \left( \mathbf{z}^{(t)} \right) \right\|^2 - \frac{1}{2} \left\| \nabla f \left( \mathbf{x}^{(t)} \right) \right\|^2 + \frac{1}{2} \left\| \nabla f \left( \mathbf{z}^{(t)} \right) - \nabla f \left( \mathbf{x}^{(t)} \right) \right\|^2 - \frac{1}{2} \left\| \nabla f \left( \mathbf{z}^{(t)} \right) \right\|^2$$

$$- \frac{1}{2} \left\| \frac{1}{N} \sum_{i=1}^{N} \left( \mathbf{h}_i^{(t)} - \nabla f_i \left( \mathbf{x}^{(t)} \right) \right) \right\|^2 + \frac{1}{2} \left\| \nabla f \left( \mathbf{z}^{(t)} \right) - \frac{1}{N} \sum_{i=1}^{N} \left( \mathbf{h}_i^{(t)} - \nabla f_i \left( \mathbf{x}^{(t)} \right) \right) \right\|^2$$

$$\overset{Lemma 1}{\leq} - \left\| \nabla f \left( \mathbf{z}^{(t)} \right) \right\|^2 - \frac{1}{2} \left\| \nabla f \left( \mathbf{x}^{(t)} \right) \right\|^2 + \frac{1}{2} \left\| \nabla f \left( \mathbf{z}^{(t)} \right) - \nabla f \left( \mathbf{x}^{(t)} \right) \right\|^2$$

$$+ \frac{1}{2} \left\| \frac{1}{N} \sum_{i=1}^{N} \left( \mathbf{h}_i^{(t)} - \nabla f_i \left( \mathbf{x}^{(t)} \right) \right) \right\|^2 + \left\| \nabla f \left( \mathbf{z}^{(t)} \right) \right\|^2$$

$$= -\frac{1}{2} \left\| \nabla f \left( \mathbf{x}^{(t)} \right) \right\|^2 + \frac{1}{2} \left\| \nabla f \left( \mathbf{z}^{(t)} \right) - \nabla f \left( \mathbf{x}^{(t)} \right) \right\|^2 + \frac{1}{2} \left\| \frac{1}{N} \sum_{i=1}^{N} \left( \mathbf{h}_i^{(t)} - \nabla f_i \left( \mathbf{x}^{(t)} \right) \right) \right\|^2.$$

*Proof of Lemma 4.*

$$T_2 \triangleq \mathbb{E} \left\| \frac{1}{b} \sum_{i \in \mathcal{A}^{(t)}} \mathbf{\Delta}_i^{(t)} \right\|^2 = \mathbb{E} \left\| \frac{1}{b} \sum_{i \in \mathcal{A}^{(t)}} \mathbf{h}_i^{(t)} + \frac{1}{b} \sum_{i \in \mathcal{A}^{(t)}} \left( \mathbf{\Delta}_i^{(t)} - \mathbf{h}_i^{(t)} \right) \right\|^2$$

$$= \mathbb{E} \left\| \frac{1}{b} \sum_{i \in \mathcal{A}^{(t)}} \mathbf{h}_i^{(t)} + \frac{1}{b} \sum_{i \in \mathcal{A}^{(t)}} \left( \frac{1}{\tau} \sum_{k=0}^{\tau-1} \left( \mathbf{g}_i^{(t,k)} - \nabla f_i \left( \mathbf{x}_i^{(t,k)} \right) \right) \right) \right\|^2$$

$$= \mathbb{E}\left\|\frac{1}{b}\sum_{i\in\mathcal{A}^{(t)}}\mathbf{h}_i^{(t)}\right\|^2 + \mathbb{E}\left\|\frac{1}{b}\sum_{i\in\mathcal{A}^{(t)}}\left(\frac{1}{\tau}\sum_{k=0}^{\tau-1}\left(\mathbf{g}_i^{(t,k)} - \nabla f_i\left(\mathbf{x}_i^{(t,k)}\right)\right)\right)\right\|^2 \qquad \text{(Using Assumption 2)}$$

$$\overset{\substack{\textit{Lemma 2 in}\\ \textit{[Wang et al., 2020b]}}}{=} \mathbb{E}\left\|\frac{1}{b}\sum_{i\in\mathcal{A}^{(t)}}\mathbf{h}_i^{(t)}\right\|^2 + \frac{1}{bN}\sum_{i=1}^{N}\frac{1}{\tau^2}\sum_{k=0}^{\tau-1}\mathbb{E}\left\|\left(\mathbf{g}_i^{(t,k)} - \nabla f_i\left(\mathbf{x}_i^{(t,k)}\right)\right)\right\|^2$$

$$\leq \mathbb{E}\left\|\frac{1}{b}\sum_{i\in\mathcal{A}^{(t)}}\mathbf{h}_i^{(t)}\right\|^2 + \frac{\sigma_l^2}{\tau b}.$$

*Proof of Lemma 5.*

$$\mathbb{E}\left\|\frac{1}{b}\sum_{i\in\mathcal{A}^{(t)}}\left(\nabla f_i\left(\mathbf{x}^{(t)}\right) - \nabla f\left(\mathbf{x}^{(t)}\right)\right)\right\|^2$$

$$= \frac{1}{b^2}\mathbb{E}\left[\sum_{i\in\mathcal{A}^{(t)}}\left\|\nabla f_i\left(\mathbf{x}^{(t)}\right) - \nabla f\left(\mathbf{x}^{(t)}\right)\right\|^2 + \sum_{\substack{i\text{ and }r\text{ are}\\ \text{two different}\\ \text{items in }\mathcal{A}^{(t)}}}\left\langle\nabla f_i\left(\mathbf{x}^{(t)}\right) - \nabla f\left(\mathbf{x}^{(t)}\right), \nabla f_r\left(\mathbf{x}^{(t)}\right) - \nabla f\left(\mathbf{x}^{(t)}\right)\right\rangle\right]$$

$$\overset{(a)}{=} \frac{1}{bN}\sum_{i=1}^{N}\mathbb{E}\left\|\nabla f_i\left(\mathbf{x}^{(t)}\right) - \nabla f\left(\mathbf{x}^{(t)}\right)\right\|^2 + \mathbb{E}\left[\frac{1}{N^2}\sum_{i=1}^{N}\sum_{r=1}^{N}\left\langle\nabla f_i\left(\mathbf{x}^{(t)}\right) - \nabla f\left(\mathbf{x}^{(t)}\right), \nabla f_r\left(\mathbf{x}^{(t)}\right) - \nabla f\left(\mathbf{x}^{(t)}\right)\right\rangle\right]$$

$$\overset{(b)}{=} \frac{1}{bN}\sum_{i=1}^{N}\mathbb{E}\left\|\nabla f_i\left(\mathbf{x}^{(t)}\right) - \nabla f\left(\mathbf{x}^{(t)}\right)\right\|^2, \tag{16}$$

where *(a)* follows that the clients in $\mathcal{A}^{(t)}$ are selected uniformly at random with replacement among all clients (see Section E.1.2), and *(b)* follows that $\sum_{i=1}^{N}\nabla f_i\left(\mathbf{x}^{(t)}\right) = Nf\left(\mathbf{x}^{(t)}\right)$. This proves the first part of Lemma 5.

$$T_3 \triangleq \mathbb{E}\left\|\frac{1}{b}\sum_{i\in\mathcal{A}^{(t)}}\mathbf{h}_i^{(t)}\right\|^2 = \mathbb{E}\left\|\frac{1}{b}\sum_{i\in\mathcal{A}^{(t)}}\left(\mathbf{h}_i^{(t)} - \nabla f_i\left(\mathbf{x}^{(t)}\right) + \nabla f_i\left(\mathbf{x}^{(t)}\right) - \nabla f\left(\mathbf{x}^{(t)}\right)\right) + \nabla f\left(\mathbf{x}^{(t)}\right)\right\|^2$$

$$\overset{\textit{Lemma 1}}{\leq} 3\mathbb{E}\left\|\frac{1}{b}\sum_{i\in\mathcal{A}^{(t)}}\left(\mathbf{h}_i^{(t)} - \nabla f_i\left(\mathbf{x}^{(t)}\right)\right)\right\|^2 + 3\mathbb{E}\left\|\frac{1}{b}\sum_{i\in\mathcal{A}^{(t)}}\left(\nabla f_i\left(\mathbf{x}^{(t)}\right) - \nabla f\left(\mathbf{x}^{(t)}\right)\right)\right\|^2 + 3\mathbb{E}\left\|\nabla f\left(\mathbf{x}^{(t)}\right)\right\|^2$$

$$\overset{\substack{\textit{Using (16)}\\ \textit{and}\\ \textit{Lemma 1}}}{\leq} \frac{3}{N}\sum_{i=1}^{N}\mathbb{E}\left\|\mathbf{h}_i^{(t)} - \nabla f_i\left(\mathbf{x}^{(t)}\right)\right\|^2 + \frac{3}{bN}\sum_{i=1}^{N}\mathbb{E}\left\|\nabla f_i\left(\mathbf{x}^{(t)}\right) - \nabla f\left(\mathbf{x}^{(t)}\right)\right\|^2 + 3\mathbb{E}\left\|\nabla f\left(\mathbf{x}^{(t)}\right)\right\|^2$$

$$\overset{\textit{Asm. 3}}{\leq} \frac{3}{N}\sum_{i=1}^{N}\mathbb{E}\left\|\mathbf{h}_i^{(t)} - \nabla f_i\left(\mathbf{x}^{(t)}\right)\right\|^2 + \frac{3\sigma_g^2}{b} + 3\mathbb{E}\left\|\nabla f\left(\mathbf{x}^{(t)}\right)\right\|^2.$$

*Proof of Lemma 6.* We start by using Assumption 1 (Smoothness) and Remark 1.

$$\mathbb{E}\left\|\nabla f\left(\mathbf{z}^{(t)}\right) - \nabla f\left(\mathbf{x}^{(t)}\right)\right\|^2 \leq L^2\mathbb{E}\left\|\mathbf{z}^{(t)} - \mathbf{x}^{(t)}\right\|^2 = L^2\mathbb{E}\left\|\widetilde{\eta}_s\eta_c\frac{1}{b}\sum_{i\in\mathcal{C}^{(t)}}\mathbf{\Delta}_i^{(t-\gamma_i^t)}\right\|^2$$

$$= L^2\mathbb{E}\left\|\frac{\widetilde{\eta}_s\eta_c}{b}\sum_{i\in\mathcal{C}^{(t)}}\left(\mathbf{\Delta}_i^{(t-\gamma_i^t)} - \mathbf{h}_i^{(t-\gamma_i^t)} + \mathbf{h}_i^{(t-\gamma_i^t)}\right)\right\|^2$$

$$\overset{\textit{Asm. 2}}{=} L^2\widetilde{\eta}_s^2\eta_c^2\mathbb{E}\left\|\frac{1}{b}\sum_{i\in\mathcal{C}^{(t)}}\left(\mathbf{\Delta}_i^{(t-\gamma_i^t)} - \mathbf{h}_i^{(t-\gamma_i^t)}\right)\right\|^2 + L^2\widetilde{\eta}_s^2\eta_c^2\mathbb{E}\left\|\frac{1}{b}\sum_{i\in\mathcal{C}^{(t)}}\mathbf{h}_i^{(t-\gamma_i^t)}\right\|^2$$

$$= L^2\widetilde{\eta}_s^2\eta_c^2\mathbb{E}\left\|\frac{1}{b}\sum_{i\in\mathcal{C}^{(t)}}\frac{1}{\tau}\sum_{k=0}^{\tau-1}\left(\mathbf{g}_i^{(t-\gamma_i^t,k)}-\nabla f_i\left(\mathbf{x}_i^{(t-\gamma_i^t,k)}\right)\right)\right\|^2 + L^2\widetilde{\eta}_s^2\eta_c^2\mathbb{E}\left\|\frac{1}{b}\sum_{i\in\mathcal{C}^{(t)}}\mathbf{h}_i^{(t-\gamma_i^t)}\right\|^2$$

$$\stackrel{Asm.\ 2}{\leq}\frac{L^2\widetilde{\eta}_s^2\eta_c^2 R}{b^2\tau}\sigma_l^2 + \frac{L^2\widetilde{\eta}_s^2\eta_c^2 R}{b^2}\mathbb{E}\left[\sum_{i\in\mathcal{C}^{(t)}}\left\|\mathbf{h}_i^{(t-\gamma_i^t)}\right\|^2\right]$$

$$\leq \frac{L^2\widetilde{\eta}_s^2\eta_c^2 R}{b^2\tau}\sigma_l^2 + \frac{L^2\widetilde{\eta}_s^2\eta_c^2 R}{b^2}\mathbb{E}\left[\sum_{i\in\mathcal{C}^{(t)}}\left\|\mathbf{h}_i^{(t-\gamma_i^t)}-\nabla f_i\left(\mathbf{x}^{(t-\gamma_i^t)}\right)+\nabla f_i\left(\mathbf{x}^{(t-\gamma_i^t)}\right)-\nabla f\left(\mathbf{x}^{(t-\gamma_i^t)}\right)+\nabla f\left(\mathbf{x}^{(t-\gamma_i^t)}\right)\right\|^2\right]$$

$$\stackrel{Lemma\ 1}{\leq}\frac{L^2\widetilde{\eta}_s^2\eta_c^2 R}{b^2\tau}\sigma_l^2$$

$$+\frac{3L^2\widetilde{\eta}_s^2\eta_c^2 R}{b^2}\mathbb{E}\left[\sum_{i\in\mathcal{C}^{(t)}}\left(\left\|\nabla f\left(\mathbf{x}^{(t-\gamma_i^t)}\right)\right\|^2+\left\|\mathbf{h}_i^{(t-\gamma_i^t)}-\nabla f_i\left(\mathbf{x}^{(t-\gamma_i^t)}\right)\right\|^2+\left\|\nabla f\left(\mathbf{x}^{(t-\gamma_i^t)}\right)-\nabla f_i\left(\mathbf{x}^{(t-\gamma_i^t)}\right)\right\|^2\right)\right]$$

$$\leq \frac{L^2\widetilde{\eta}_s^2\eta_c^2 R}{b^2\tau}\sigma_l^2 + \frac{3L^2\widetilde{\eta}_s^2\eta_c^2 R^2}{b^2}\sigma_g^2 + \frac{3L^2\widetilde{\eta}_s^2\eta_c^2 R}{b^2}\mathbb{E}\left[\sum_{i\in\mathcal{C}^{(t)}}\left(\left\|\nabla f\left(\mathbf{x}^{(t-\gamma_i^t)}\right)\right\|^2+\left\|\mathbf{h}_i^{(t-\gamma_i^t)}-\nabla f_i\left(\mathbf{x}^{(t-\gamma_i^t)}\right)\right\|^2\right)\right].$$

Telescoping the inequality over $t=0,\ldots,T-1$:

$$\frac{1}{T}\sum_{t=0}^{T-1}\mathbb{E}\left\|\nabla f\left(\mathbf{z}^{(t)}\right)-\nabla f\left(\mathbf{x}^{(t)}\right)\right\|^2 \leq \frac{L^2\widetilde{\eta}_s^2\eta_c^2 R}{b^2\tau}\sigma_l^2 + \frac{3L^2\widetilde{\eta}_s^2\eta_c^2 R^2}{b^2}\sigma_g^2$$

$$+\frac{3L^2\widetilde{\eta}_s^2\eta_c^2 R}{b^2}\frac{1}{T}\sum_{t=0}^{T-1}\mathbb{E}\left[\sum_{i\in\mathcal{C}^{(t)}}\left(\left\|\nabla f\left(\mathbf{x}^{(t-\gamma_i^t)}\right)\right\|^2+\left\|\mathbf{h}_i^{(t-\gamma_i^t)}-\nabla f_i\left(\mathbf{x}^{(t-\gamma_i^t)}\right)\right\|^2\right)\right]$$

$$\stackrel{Remark\ 2}{\leq}\frac{L^2\widetilde{\eta}_s^2\eta_c^2 R}{b^2\tau}\sigma_l^2 + \frac{3L^2\widetilde{\eta}_s^2\eta_c^2 R^2}{b^2}\sigma_g^2 + \frac{3L^2\widetilde{\eta}_s^2\eta_c^2 R\gamma^{\max}}{b}\frac{1}{T}\sum_{t=0}^{T-1}\mathbb{E}\left\|\nabla f\left(\mathbf{x}^{(t)}\right)\right\|^2$$

$$+\frac{3L^2\widetilde{\eta}_s^2\eta_c^2 R}{b^2}\frac{1}{T}\sum_{t=0}^{T-1}\mathbb{E}\left[\sum_{i\in\mathcal{C}^{(t)}}\left\|\mathbf{h}_i^{(t-\gamma_i^t)}-\nabla f_i\left(\mathbf{x}^{(t-\gamma_i^t)}\right)\right\|^2\right]$$

$$\stackrel{Lemma\ 2}{\leq}\frac{L^2\widetilde{\eta}_s^2\eta_c^2 R}{b^2\tau}\sigma_l^2 + \frac{3L^2\widetilde{\eta}_s^2\eta_c^2 R^2}{b^2}\sigma_g^2 + \frac{3L^2\widetilde{\eta}_s^2\eta_c^2 R\gamma^{\max}}{b}\frac{1}{T}\sum_{t=0}^{T-1}\mathbb{E}\left\|\nabla f\left(\mathbf{x}^{(t)}\right)\right\|^2$$

$$+\frac{3L^2\widetilde{\eta}_s^2\eta_c^2 R}{b^2}\frac{1}{T}\sum_{t=0}^{T-1}\mathbb{E}\left[\sum_{i\in\mathcal{C}^{(t)}}\left(\frac{L^2\eta_c^2\tau}{2(1-D)}\sigma_l^2+\frac{D}{1-D}\left\|\nabla f_i\left(\mathbf{x}^{(t-\gamma_i^t)}\right)\right\|^2\right)\right]$$

$$\leq \left(1+\frac{3RL^2\eta_c^2\tau^2}{2(1-D)}\right)\frac{L^2\widetilde{\eta}_s^2\eta_c^2 R}{b^2\tau}\sigma_l^2 + \frac{3L^2\widetilde{\eta}_s^2\eta_c^2 R^2}{b^2}\sigma_g^2 + \frac{3L^2\widetilde{\eta}_s^2\eta_c^2 R\gamma^{\max}}{b}\frac{1}{T}\sum_{t=0}^{T-1}\mathbb{E}\left\|\nabla f\left(\mathbf{x}^{(t)}\right)\right\|^2$$

$$+\frac{3L^2\widetilde{\eta}_s^2\eta_c^2 R}{b^2}\frac{1}{T}\sum_{t=0}^{T-1}\mathbb{E}\left[\sum_{i\in\mathcal{C}^{(t)}}\frac{D}{1-D}\left\|\nabla f_i\left(\mathbf{x}^{(t-\gamma_i^t)}\right)\right\|^2\right]$$

$$\stackrel{Lemma\ 1}{\leq}\left(1+\frac{3RL^2\eta_c^2\tau^2}{2(1-D)}\right)\frac{L^2\widetilde{\eta}_s^2\eta_c^2 R}{b^2\tau}\sigma_l^2 + \frac{3L^2\widetilde{\eta}_s^2\eta_c^2 R^2}{b^2}\sigma_g^2 + \frac{3L^2\widetilde{\eta}_s^2\eta_c^2 R\gamma^{\max}}{b}\frac{1}{T}\sum_{t=0}^{T-1}\mathbb{E}\left\|\nabla f\left(\mathbf{x}^{(t)}\right)\right\|^2$$

$$+\frac{3L^2\widetilde{\eta}_s^2\eta_c^2 R}{b^2}\frac{1}{T}\sum_{t=0}^{T-1}\mathbb{E}\left[\sum_{i\in\mathcal{C}^{(t)}}\left(\frac{2D}{1-D}\left\|\nabla f_i\left(\mathbf{x}^{(t-\gamma_i^t)}\right)-\nabla f\left(\mathbf{x}^{(t-\gamma_i^t)}\right)\right\|^2+\frac{2D}{1-D}\left\|\nabla f\left(\mathbf{x}^{(t-\gamma_i^t)}\right)\right\|^2\right)\right]$$

$$\stackrel{Asm.\ 3}{\leq}\left(1+\frac{3RL^2\eta_c^2\tau^2}{2(1-D)}\right)\frac{L^2\widetilde{\eta}_s^2\eta_c^2 R}{b^2\tau}\sigma_l^2 + \frac{1+D}{1-D}\frac{3L^2\widetilde{\eta}_s^2\eta_c^2 R^2}{b^2}\sigma_g^2 + \frac{3L^2\widetilde{\eta}_s^2\eta_c^2 R\gamma^{\max}}{b}\frac{1}{T}\sum_{t=0}^{T-1}\mathbb{E}\left\|\nabla f\left(\mathbf{x}^{(t)}\right)\right\|^2$$

$$+ \frac{3L^2 \widetilde{\eta}_s^2 \eta_c^2 R}{b^2} \frac{1}{T} \sum_{t=0}^{T-1} \mathbb{E} \left[ \sum_{i \in \mathcal{C}^{(t)}} \frac{2D}{1-D} \left\| \nabla f \left( \mathbf{x}^{(t-\gamma_i^t)} \right) \right\|^2 \right]$$

$$\overset{\text{Remark 2}}{\leq} \left( 1 + \frac{3RL^2 \eta_c^2 \tau^2}{2(1-D)} \right) \frac{L^2 \widetilde{\eta}_s^2 \eta_c^2 R}{b^2 \tau} \sigma_l^2 + \frac{1+D}{1-D} \frac{3L^2 \widetilde{\eta}_s^2 \eta_c^2 R^2}{b^2} \sigma_g^2 + \frac{3L^2 \widetilde{\eta}_s^2 \eta_c^2 R \gamma^{\max}}{b} \frac{1}{T} \sum_{t=0}^{T-1} \mathbb{E} \left\| \nabla f \left( \mathbf{x}^{(t)} \right) \right\|^2$$

$$+ \frac{3L^2 \widetilde{\eta}_s^2 \eta_c^2 R \gamma^{\max}}{b} \frac{2D}{1-D} \frac{1}{T} \sum_{t=0}^{T-1} \mathbb{E} \left\| \nabla f \left( \mathbf{x}^{(t)} \right) \right\|^2$$

$$= \left( 1 + \frac{3RL^2 \eta_c^2 \tau^2}{2(1-D)} \right) \frac{L^2 \widetilde{\eta}_s^2 \eta_c^2 R}{b^2 \tau} \sigma_l^2 + \frac{1+D}{1-D} \frac{3L^2 \widetilde{\eta}_s^2 \eta_c^2 R^2}{b^2} \sigma_g^2 + \frac{3L^2 \widetilde{\eta}_s^2 \eta_c^2 R \gamma^{\max}}{b} \frac{1+D}{1-D} \frac{1}{T} \sum_{t=0}^{T-1} \mathbb{E} \left\| \nabla f \left( \mathbf{x}^{(t)} \right) \right\|^2 .$$

# F   CONVERGENCE OF `FedAST` WITH DYNAMIC CLIENT ALLOCATION (*OPTION* $= D$)

With a similar approach to the proof of static client allocation, we can show the convergence of the `FedAST` with dynamic client allocation (*option* $= D$), too. Adopting all of the previously used notation, we also need some new definitions to analyze this version of the algorithm, as the number of active training requests and buffer size can change dynamically during the training.

**Notation for changing buffer size and number of active training requests.**   Let us define $b^{(t)}$ and $R^{(t)}$ as the buffer size and the number of active local training requests of the model. Further, define $b_{min}$ and $b_{max}$ the minimum and maximum value that the buffer size can take. Similarly, define $R_{min}$ and $R_{max}$ as the minimum and maximum number of active training requests. Moreover, we define $\rho_b \triangleq b_{max}/b_{min}$ as the measure of skewness in buffer size.

**Global update rule and virtual sequence definition.**   Although the local update rule remains the same, the global update rule slightly changes for dynamic client allocation due to changing buffer size:

$$\mathbf{x}^{(t+1)} \leftarrow \mathbf{x}^{(t)} - \tau \eta_s \eta_c \frac{1}{b^{(t)}} \sum_{i \in \mathcal{B}^{(t)}} \mathbf{\Delta}_i^{(t-\gamma_i^t)} = \mathbf{x}^{(t)} - \eta_s \frac{1}{b^{(t)}} \sum_{i \in \mathcal{B}^{(t)}} \left( \mathbf{x}^{(t-\gamma_i^t)} - \mathbf{x}_i^{(t-\gamma_i^t, \tau)} \right) \tag{17}$$

$$= \mathbf{x}^{(t)} - \eta_s \eta_c \frac{1}{b^{(t)}} \sum_{i \in \mathcal{B}^{(t)}} \sum_{k=0}^{\tau-1} \widetilde{\nabla} f_i \left( \mathbf{x}_i^{(t-\gamma_i^t, k)} \right),$$

where $|\mathcal{B}^{(t)}| = b^{(t)}$. Note that (17) is almost identical to (9), except the varying buffer-size $b^{(t)}$.

Next, we define $r_i(t)$ as the index of the global round when a local training request sent to client $i$ in round $t$ returns to the server. Basically, it is the current round index $t$, added to the future value of staleness that the requested update will have. We need to define a new virtual sequence $\mathbf{y}^{(t)}$, which is different from the $\mathbf{z}^{(t)}$ defined earlier.

$$\mathbf{y}^{(t+1)} \leftarrow \mathbf{y}^{(t)} - \tau \eta_s \eta_c \sum_{i \in \mathcal{A}^{(t)}} \frac{1}{b^{r_i(t)}} \mathbf{\Delta}_i^{(t)} = \mathbf{y}^{(t)} - \tau \eta_s \eta_c \sum_{i \in \mathcal{A}^{(t)}} \frac{1}{b^{r_i(t)}} \frac{1}{\tau} \sum_{k=0}^{\tau-1} \widetilde{\nabla} f_i \left( \mathbf{x}_i^{(t,k)} \right), \tag{18}$$

for $t = 0, 1, \ldots, T-1$ where $\mathbf{y}^{(0)} \triangleq \mathbf{x}^{(0)}$. Here, $\mathcal{A}^{(t)}$ is defined similarly as it was in Section E.1.2. Note that the probability of being in $\mathcal{A}^{(t)}$ is equal across clients due to uniform client selection. However, this time, the size of this set does not have to be equal to the buffer size at round $t$. Due to the new client selection rule (Line 13 in Algorithm 2), the server may assign 0, 1, or 2 clients for each received update. Therefore, we know that $0 < |\mathcal{A}^{(t)}| \leq 2b^{(t)}$.

Here, we need a simplifying assumption for the purpose of this proof:

**Assumption 5** ($b^{r_i(t)}$ Values)**.**   *We assume that any $b^{r_i(t)}$ value is known at the time when a local training request is sent to client $i$ at round $t$, and these values are independent of any future information including the received updates. We further assume that $b^{r_i(t)}$ values are equal (denote $b^{r(t)}$) for all clients in $\mathcal{A}^{(t)}$.*

**Remark 7.**   *When we keep the period of dynamic client allocation long enough, we observe that most of the assigned local training requests at one round fall in the same window before the next dynamic client allocation happens (Line 8 in*

*Algorithm 2). Hence, based on our empirical observations, what assumption implies holds for most of the local training requests. Further, **this assumption can be avoided** by taking an average of the updates during aggregation weighted inversely with the number of local training requests sent at the same global round. In other words, one may have avoided this assumption by weighting an update from client $i$ with $1/|\mathcal{A}^{(t-\gamma_i^t)}|$ instead of taking average over buffer during aggregation at round $t$. However, we did not see any practical benefit of this type of weighting in our experiments, and this strange weighting would be just for theoretical purposes. Therefore, we keep the current version.*

We first state the theorem showing the convergence of `FedAST` with dynamic client allocation option.

**Theorem 2.** (*Convergence of* `FedAST` *with option* $= D$): *Suppose Assumptions 1 - 5 hold, and the learning rates satisfy* $\eta_s \leq \rho_b^{-3/2}\sqrt{\tau b}$ *and* $\eta_c \leq \min\left\{\frac{\rho_b^{-3/2}}{24L\tau\sqrt{\tau b}}, \frac{\rho_b^{-3/2}}{16L\tau\sqrt{\tau R\gamma^{max}}}\right\}$. *Then, the iterations of Algorithm 1* (`FedAST`) *with option* $= D$ *satisfy:*

$$\frac{1}{T}\sum_{t=0}^{T-1}\mathbb{E}\left\|\nabla f\left(\mathbf{x}^{(t)}\right)\right\|^2 \leq \mathcal{O}\left(\frac{f\left(\mathbf{x}^{(0)}\right) - \min_{\mathbf{x}} f\left(\mathbf{x}\right)}{T\eta_s\eta_c\tau}\right) + \mathcal{O}\left(\left(\frac{L\eta_s\eta_c\rho_b^3}{b_{min}} + L^2\eta_c^2\rho_b^2\tau + \frac{L^2\eta_s^2\eta_c^2\tau R_{max}\rho_b^2}{b_{min}^2}\right)\sigma_l^2\right)$$
$$+ \mathcal{O}\left(\left(\frac{L\eta_s\eta_c\tau\rho_b^3}{b_{min}} + L^2\eta_c^2\tau\left(\tau-1\right)\rho_b^2 + \frac{L^2\eta_s^2\eta_c^2\tau^2 R_{max}^2\rho_b^2}{b_{min}^2}\right)\sigma_g^2\right).$$

**Proof.**

We will need one extra lemma corresponding to Lemma 6.

**Lemma 7.** The new virtual sequence $\left(\mathbf{y}^{(t)}\right)$ and the iterates of `FedAST` satisfy,

$$\frac{1}{T}\sum_{t=0}^{T-1}\mathbb{E}\left\|\nabla f\left(\mathbf{y}^{(t)}\right) - \nabla f\left(\mathbf{x}^{(t)}\right)\right\|^2 \leq \left(1 + \frac{3R_{max}L^2\eta_c^2\tau^2}{2\left(1-D\right)}\right)\frac{L^2\widetilde{\eta}_s^2\eta_c^2 R_{max}}{b_{min}^2\tau}\sigma_l^2$$
$$+ \frac{1+D}{1-D}\frac{3L^2\widetilde{\eta}_s^2\eta_c^2 R_{max}^2}{b_{min}^2}\sigma_g^2 + \frac{6L^2\widetilde{\eta}_s^2\eta_c^2 R_{max}\rho_b\gamma^{max}}{b_{min}}\frac{1+D}{1-D}\frac{1}{T}\sum_{t=0}^{T-1}\mathbb{E}\left\|\nabla f\left(\mathbf{x}^{(t)}\right)\right\|^2.$$

Now, using the update rule of the virtual sequence (18) and Assumption 1 (Smoothness), and taking the conditional expectation with respect to $\mathbf{y}^{(t)}$, we have,

$$\mathbb{E}\left[f\left(\mathbf{y}^{(t+1)}\right)\right] \leq f\left(\mathbf{y}^{(t)}\right) + \left\langle\nabla f\left(\mathbf{y}^{(t)}\right), \mathbb{E}\left[\mathbf{y}^{(t+1)} - \mathbf{y}^{(t)}\right]\right\rangle + \frac{L}{2}\mathbb{E}\left\|\mathbf{y}^{(t+1)} - \mathbf{y}^{(t)}\right\|^2$$

$$= f\left(\mathbf{y}^{(t)}\right) + \left\langle\nabla f\left(\mathbf{y}^{(t)}\right), \mathbb{E}\left[-\widetilde{\eta}_s\eta_c\sum_{i\in\mathcal{A}^{(t)}}\frac{1}{b^{r_i(t)}}\boldsymbol{\Delta}_i^{(t)}\right]\right\rangle + \frac{L}{2}\mathbb{E}\left\|\widetilde{\eta}_s\eta_c\sum_{i\in\mathcal{A}^{(t)}}\frac{1}{b^{r_i(t)}}\boldsymbol{\Delta}_i^{(t)}\right\|^2$$

$$\leq f\left(\mathbf{y}^{(t)}\right) - \widetilde{\eta}_s\eta_c\frac{|\mathcal{A}^{(t)}|}{b^{r(t)}}\mathbb{E}\left[\left\langle\nabla f\left(\mathbf{y}^{(t)}\right), \frac{1}{|\mathcal{A}^{(t)}|}\sum_{i\in\mathcal{A}^{(t)}}\mathbf{h}_i^{(t)}\right\rangle\right] \qquad \text{(Using Assumption 5)}$$

$$+ \frac{L|\mathcal{A}^{(t)}|^2}{2(b^{r(t)})^2}\mathbb{E}\left\|\widetilde{\eta}_s\eta_c\frac{1}{|\mathcal{A}^{(t)}|}\sum_{i\in\mathcal{A}^{(t)}}\boldsymbol{\Delta}_i^{(t)}\right\|^2 \qquad (|\mathcal{A}^{(t)}| \text{ is not random with conditional expectation})$$

$$= f\left(\mathbf{y}^{(t)}\right) + \widetilde{\eta}_s\eta_c\frac{|\mathcal{A}^{(t)}|}{b^{r(t)}}\mathbb{E}\left[\underbrace{-\left\langle\nabla f\left(\mathbf{y}^{(t)}\right), \frac{1}{N}\sum_{i=1}^{N}\mathbf{h}_i^{(t)}\right\rangle}_{\triangleq T_1}\right] + 2\rho_b^2 L\widetilde{\eta}_s^2\eta_c^2\mathbb{E}\left\|\frac{1}{|\mathcal{A}^{(t)}|}\sum_{i\in\mathcal{A}^{(t)}}\boldsymbol{\Delta}_i^{(t)}\right\|^2. \qquad (\frac{|\mathcal{A}^{(t)}|}{b^{r(t)}} \leq 2\rho_b)$$

Next, using Lemma 3 (with $\mathbf{y}^{(t)}$ sequence) and Lemma 4 (with $|\mathcal{A}^{(t)}|$), using $1/\rho_b \leq |\mathcal{A}^{(t)}|/b^{r(t)} \leq 2\rho_b$, and dividing both sides by $\widetilde{\eta}_s\eta_c$ we obtain,

$$\frac{\mathbb{E}\left[f\left(\mathbf{y}^{(t+1)}\right)\right] - f\left(\mathbf{y}^{(t)}\right)}{\widetilde{\eta}_s\eta_c} \leq -\frac{1}{2\rho_b}\mathbb{E}\left\|\nabla f\left(\mathbf{x}^{(t)}\right)\right\|^2 + \rho_b\mathbb{E}\left\|\nabla f\left(\mathbf{y}^{(t)}\right) - \nabla f\left(\mathbf{x}^{(t)}\right)\right\|^2$$

$$+ \frac{\rho_b}{N} \sum_{i=1}^{N} \mathbb{E} \left\| \mathbf{h}_i^{(t)} - \nabla f_i \left( \mathbf{x}^{(t)} \right) \right\|^2 + 2\rho_b^2 L \widetilde{\eta}_s \eta_c \mathbb{E} \left\| \frac{1}{|\mathcal{A}^{(t)}|} \sum_{i \in \mathcal{A}^{(t)}} \mathbf{h}_i^{(t)} \right\|^2 + 2\rho_b^2 L \widetilde{\eta}_s \eta_c \frac{\sigma_l^2}{\tau b_{min}}.$$

Using Lemma 5, we get,

$$\frac{\mathbb{E} \left[ f \left( \mathbf{y}^{(t+1)} \right) \right] - f \left( \mathbf{y}^{(t)} \right)}{\widetilde{\eta}_s \eta_c}$$

$$\leq -\frac{1}{2\rho_b} \mathbb{E} \left\| \nabla f \left( \mathbf{x}^{(t)} \right) \right\|^2 + \rho_b \mathbb{E} \left\| \nabla f \left( \mathbf{y}^{(t)} \right) - \nabla f \left( \mathbf{x}^{(t)} \right) \right\|^2 + \frac{\rho_b}{N} \sum_{i=1}^{N} \mathbb{E} \left\| \mathbf{h}_i^{(t)} - \nabla f_i \left( \mathbf{x}^{(t)} \right) \right\|^2$$

$$+ \rho_b^2 L \widetilde{\eta}_s \eta_c \left( \frac{6}{N} \sum_{i=1}^{N} \mathbb{E} \left\| \mathbf{h}_i^{(t)} - \nabla f_i \left( \mathbf{x}^{(t)} \right) \right\|^2 + \frac{6\sigma_g^2}{b_{min}} + 6\mathbb{E} \left\| \nabla f \left( \mathbf{x}^{(t)} \right) \right\|^2 + \frac{2\sigma_l^2}{\tau b_{min}} \right)$$

$$= \left( -\frac{1}{2\rho_b} + 6\rho_b^2 L \widetilde{\eta}_s \eta_c \right) \mathbb{E} \left\| \nabla f \left( \mathbf{x}^{(t)} \right) \right\|^2 + \rho_b \mathbb{E} \left\| \nabla f \left( \mathbf{y}^{(t)} \right) - \nabla f \left( \mathbf{x}^{(t)} \right) \right\|^2 + \rho_b^2 L \widetilde{\eta}_s \eta_c \left( \frac{6\sigma_g^2}{b_{min}} + \frac{2\sigma_l^2}{\tau b_{min}} \right)$$

$$+ \left( 6\rho_b^2 L \widetilde{\eta}_s \eta_c + \rho_b \right) \frac{1}{N} \sum_{i=1}^{N} \mathbb{E} \left\| \mathbf{h}_i^{(t)} - \nabla f_i \left( \mathbf{x}^{(t)} \right) \right\|^2$$

$$\leq \left( -\frac{1}{2\rho_b} + 6\rho_b^2 L \widetilde{\eta}_s \eta_c \right) \mathbb{E} \left\| \nabla f \left( \mathbf{x}^{(t)} \right) \right\|^2 + \rho_b \mathbb{E} \left\| \nabla f \left( \mathbf{y}^{(t)} \right) - \nabla f \left( \mathbf{x}^{(t)} \right) \right\|^2 + \rho_b^2 L \widetilde{\eta}_s \eta_c \left( \frac{6\sigma_g^2}{b_{min}} + \frac{2\sigma_l^2}{\tau b_{min}} \right)$$

$$+ \left( 6\rho_b^2 L \widetilde{\eta}_s \eta_c + \rho_b \right) \left( \frac{L^2 \eta_c^2 \tau}{2 (1 - D)} \sigma_l^2 + \frac{D}{1 - D} \mathbb{E} \left\| \nabla f \left( \mathbf{x}^{(t)} \right) \right\|^2 + \frac{D}{1 - D} \sigma_g^2 \right) \qquad \text{(Using Lemma 2)}$$

$$= \left( -\frac{1}{2\rho_b} + 6\rho_b^2 L \widetilde{\eta}_s \eta_c + \frac{\rho_b D}{(1 - D)} + \frac{6\rho_b^2 L \widetilde{\eta}_s \eta_c D}{(1 - D)} \right) \mathbb{E} \left\| \nabla f \left( \mathbf{x}^{(t)} \right) \right\|^2 + \rho_b \mathbb{E} \left\| \nabla f \left( \mathbf{y}^{(t)} \right) - \nabla f \left( \mathbf{x}^{(t)} \right) \right\|^2$$

$$+ \left( \frac{2\rho_b^2 L \widetilde{\eta}_s \eta_c}{\tau b_{min}} + \frac{3\rho_b^2 L^3 \eta_c^3 \widetilde{\eta}_s \tau}{(1 - D)} + \frac{\rho_b L^2 \eta_c^2 \tau}{2 (1 - D)} \right) \sigma_l^2 + \left( \frac{6\rho_b^2 L \widetilde{\eta}_s \eta_c}{b_{min}} + \frac{6\rho_b^2 L \widetilde{\eta}_s \eta_c D}{(1 - D)} + \frac{\rho_b D}{(1 - D)} \right) \sigma_g^2,$$

where $D \triangleq L^2 \eta_c^2 \tau (\tau - 1)$. Using tower property of conditional expectation, telescoping the inequality over the round indices $t = 0, 1, \ldots, T - 1$, and using Lemma 7, we get,

$$\frac{1}{T} \sum_{t=0}^{T-1} \left( \frac{1}{2\rho_b} - 6\rho_b^2 L \widetilde{\eta}_s \eta_c - \frac{\rho_b D}{(1 - D)} - \frac{6\rho_b^2 L \widetilde{\eta}_s \eta_c D}{(1 - D)} \right) \mathbb{E} \left\| \nabla f \left( \mathbf{x}^{(t)} \right) \right\|^2 \leq \frac{\rho_b}{T} \sum_{t=0}^{T-1} \mathbb{E} \left\| \nabla f \left( \mathbf{y}^{(t)} \right) - \nabla f \left( \mathbf{x}^{(t)} \right) \right\|^2$$

$$+ \frac{f \left( \mathbf{y}^{(0)} \right) - \mathbb{E} \left[ f \left( \mathbf{y}^{(T)} \right) \right]}{T \widetilde{\eta}_s \eta_c} + \left( \frac{2\rho_b^2 L \widetilde{\eta}_s \eta_c}{\tau b_{min}} + \frac{3\rho_b^2 L^3 \eta_c^3 \widetilde{\eta}_s \tau}{(1 - D)} + \frac{\rho_b L^2 \eta_c^2 \tau}{2 (1 - D)} \right) \sigma_l^2 + \left( \frac{6\rho_b^2 L \widetilde{\eta}_s \eta_c}{b_{min}} + \frac{6\rho_b^2 L \widetilde{\eta}_s \eta_c D}{(1 - D)} + \frac{\rho_b D}{(1 - D)} \right) \sigma_g^2$$

$$\leq \left( 1 + \frac{3R_{max} L^2 \eta_c^2 \tau^2}{2 (1 - D)} \right) \frac{L^2 \widetilde{\eta}_s^2 \eta_c^2 R_{max} \rho_b}{b_{min}^2 \tau} \sigma_l^2 + \frac{1 + D}{1 - D} \frac{3L^2 \widetilde{\eta}_s^2 \eta_c^2 R_{max}^2 \rho_b}{b_{min}^2} \sigma_g^2 + \frac{6L^2 \widetilde{\eta}_s^2 \eta_c^2 R_{max} \rho_b^2 \gamma^{\max}}{b_{min}} \frac{1 + D}{1 - D} \frac{1}{T} \sum_{t=0}^{T-1} \mathbb{E} \left\| \nabla f \left( \mathbf{x}^{(t)} \right) \right\|^2$$

$$+ \frac{f \left( \mathbf{y}^{(0)} \right) - \mathbb{E} \left[ f \left( \mathbf{y}^{(T)} \right) \right]}{T \widetilde{\eta}_s \eta_c} + \left( \frac{2\rho_b^2 L \widetilde{\eta}_s \eta_c}{\tau b_{min}} + \frac{3\rho_b^2 L^3 \eta_c^3 \widetilde{\eta}_s \tau}{(1 - D)} + \frac{\rho_b L^2 \eta_c^2 \tau}{2 (1 - D)} \right) \sigma_l^2 + \left( \frac{6\rho_b^2 L \widetilde{\eta}_s \eta_c}{b_{min}} + \frac{6\rho_b^2 L \widetilde{\eta}_s \eta_c D}{(1 - D)} + \frac{\rho_b D}{(1 - D)} \right) \sigma_g^2.$$

Suppose the learning rates satisfy $\eta_s \leq \rho_b^{-3/2} \sqrt{\tau b}$ (which also makes $\widetilde{\eta}_s \leq \rho_b^{-3/2} \tau \sqrt{\tau b}$) and $\eta_c \leq \min \left\{ \frac{\rho_b^{-3/2}}{24 L \tau \sqrt{\tau b}}, \frac{\rho_b^{-3/2}}{16 L \tau \sqrt{\tau R \gamma^{\max}}} \right\}$, the following inequality holds:

$$\frac{1}{2} - 6\rho_b^3 L \widetilde{\eta}_s \eta_c - \frac{\rho_b^2 D}{(1 - D)} - \frac{6\rho_b^3 L \widetilde{\eta}_s \eta_c D}{(1 - D)} - \frac{6L^2 \widetilde{\eta}_s^2 \eta_c^2 R_{max} \rho_b^3 \gamma^{\max}}{b_{min}} \frac{1 + D}{1 - D} \geq \frac{1}{11}. \qquad (19)$$

Also, notice that $\mathbf{y}^{(0)}$ is equal to $\mathbf{x}^{(0)}$ by definitions (Section F) of these sequences and $\min_{\mathbf{x}} f (\mathbf{x}) \leq f \left( \mathbf{y}^{(T)} \right)$.

$$\frac{1}{T} \sum_{t=0}^{T-1} \mathbb{E} \left\| \nabla f \left( \mathbf{x}^{(t)} \right) \right\|^2 \leq 11 \frac{f \left( \mathbf{x}^{(0)} \right) - \min_{\mathbf{x}} f (\mathbf{x})}{T \widetilde{\eta}_s \eta_c} \rho_b \qquad \text{(Using (19))}$$

$$+ 11 \left( \frac{2\rho_b^3 L \widetilde{\eta}_s \eta_c}{\tau b_{min}} + \frac{3\rho_b^3 L^3 \eta_c^3 \widetilde{\eta}_s \tau}{(1-D)} + \frac{\rho_b^2 L^2 \eta_c^2 \tau}{2(1-D)} + \left( 1 + \frac{3R_{max} L^2 \eta_c^2 \tau^2}{2(1-D)} \right) \frac{L^2 \widetilde{\eta}_s^2 \eta_c^2 R_{max} \rho_b^2}{b_{min}^2 \tau} \right) \sigma_l^2$$

$$+ 11 \left( \frac{6\rho_b^3 L \widetilde{\eta}_s \eta_c}{b_{min}} + \frac{6\rho_b^3 L \widetilde{\eta}_s \eta_c D}{(1-D)} + \frac{\rho_b^2 D}{(1-D)} + \frac{1+D}{1-D} \frac{3L^2 \widetilde{\eta}_s^2 \eta_c^2 R_{max}^2 \rho_b^2}{b_{min}^2} \right) \sigma_g^2.$$

Define $\delta \triangleq f\left(\mathbf{x}^{(0)}\right) - \min_{\mathbf{x}} f\left(\mathbf{x}\right)$. After reducing high-order terms using the assumptions, $\eta_s \leq \rho_b^{-3/2} \sqrt{\tau b}$ (which also makes $\widetilde{\eta}_s \leq \rho_b^{-3/2} \tau \sqrt{\tau b}$) and $\eta_c \leq \min \left\{ \frac{\rho_b^{-3/2}}{24 L \tau \sqrt{\tau b}}, \frac{\rho_b^{-3/2}}{16 L \tau \sqrt{\tau R \gamma^{\max}}} \right\}$, and incorporating the constants into the $\mathcal{O}(\cdot)$ notation, we have:

$$\frac{1}{T} \sum_{t=0}^{T-1} \mathbb{E} \left\| \nabla f\left(\mathbf{x}^{(t)}\right) \right\|^2 \leq \mathcal{O}\left( \frac{\delta \rho_b}{T \eta_s \eta_c \tau} \right) + \mathcal{O}\left( \left( \frac{L \eta_s \eta_c \rho_b^3}{b_{min}} + L^2 \eta_c^2 \rho_b^2 \tau + \frac{L^2 \eta_s^2 \eta_c^2 \tau R_{max} \rho_b^2}{b_{min}^2} \right) \sigma_l^2 \right)$$

$$+ \mathcal{O}\left( \left( \frac{L \eta_s \eta_c \tau \rho_b^3}{b_{min}} + L^2 \eta_c^2 \tau (\tau-1) \rho_b^2 + \frac{L^2 \eta_s^2 \eta_c^2 \tau^2 R_{max}^2 \rho_b^2}{b_{min}^2} \right) \sigma_g^2 \right).$$

This concludes the proof.

**Proof of Lemma 7:** We start by using Assumption 1 (Smoothness) and observing that Remark 1 still holds with $\mathbf{y}^{(t)}$ for the dynamic client allocation option.

$$\mathbb{E} \left\| \nabla f\left(\mathbf{y}^{(t)}\right) - \nabla f\left(\mathbf{x}^{(t)}\right) \right\|^2 \leq L^2 \mathbb{E} \left\| \mathbf{y}^{(t)} - \mathbf{x}^{(t)} \right\|^2 = L^2 \mathbb{E} \left\| \widetilde{\eta}_s \eta_c \sum_{i \in \mathcal{C}^{(t)}} \frac{1}{b^{r(t-\gamma_i^t)}} \boldsymbol{\Delta}_i^{(t-\gamma_i^t)} \right\|^2$$

$$= L^2 \mathbb{E} \left\| \widetilde{\eta}_s \eta_c \sum_{i \in \mathcal{C}^{(t)}} \frac{1}{b^{r(t-\gamma_i^t)}} \left( \boldsymbol{\Delta}_i^{(t-\gamma_i^t)} - \mathbf{h}_i^{(t-\gamma_i^t)} + \mathbf{h}_i^{(t-\gamma_i^t)} \right) \right\|^2$$

$$= L^2 \widetilde{\eta}_s^2 \eta_c^2 \mathbb{E} \left\| \sum_{i \in \mathcal{C}^{(t)}} \frac{1}{b^{r(t-\gamma_i^t)}} \left( \boldsymbol{\Delta}_i^{(t-\gamma_i^t)} - \mathbf{h}_i^{(t-\gamma_i^t)} \right) \right\|^2 + L^2 \widetilde{\eta}_s^2 \eta_c^2 \mathbb{E} \left\| \sum_{i \in \mathcal{C}^{(t)}} \frac{1}{b^{r(t-\gamma_i^t)}} \mathbf{h}_i^{(t-\gamma_i^t)} \right\|^2$$

$$= L^2 \widetilde{\eta}_s^2 \eta_c^2 \mathbb{E} \left\| \sum_{i \in \mathcal{C}^{(t)}} \frac{1}{b^{r(t-\gamma_i^t)}} \frac{1}{\tau} \sum_{k=0}^{\tau-1} \left( \mathbf{g}_i^{(t-\gamma_i^t, k)} - \nabla f_i\left(\mathbf{x}_i^{(t-\gamma_i^t, k)}\right) \right) \right\|^2 + L^2 \widetilde{\eta}_s^2 \eta_c^2 \mathbb{E} \left\| \sum_{i \in \mathcal{C}^{(t)}} \frac{1}{b^{r(t-\gamma_i^t)}} \mathbf{h}_i^{(t-\gamma_i^t)} \right\|^2$$

$$\text{(Using Assumption 2)}$$

$$\leq \frac{L^2 \widetilde{\eta}_s^2 \eta_c^2 R_{max}}{b_{min}^2 \tau} \sigma_l^2 + \frac{L^2 \widetilde{\eta}_s^2 \eta_c^2 R_{max}}{b_{min}^2} \mathbb{E} \left[ \sum_{i \in \mathcal{C}^{(t)}} \left\| \mathbf{h}_i^{(t-\gamma_i^t)} \right\|^2 \right] \quad \text{(Using } \left\| \sum_{i=0}^n x_i \right\|^2 \leq n \sum_{i=0}^n \|x_i\|^2 \text{ and } |\mathcal{C}^{(t)}| \leq R_{max} \text{)}$$

$$\leq \frac{L^2 \widetilde{\eta}_s^2 \eta_c^2 R_{max}}{b_{min}^2 \tau} \sigma_l^2 + \frac{L^2 \widetilde{\eta}_s^2 \eta_c^2 R_{max}}{b_{min}^2} \mathbb{E} \left[ \sum_{i \in \mathcal{C}^{(t)}} \left\| \mathbf{h}_i^{(t-\gamma_i^t)} - \nabla f_i\left(\mathbf{x}^{(t-\gamma_i^t)}\right) + \nabla f_i\left(\mathbf{x}^{(t-\gamma_i^t)}\right) - \nabla f\left(\mathbf{x}^{(t-\gamma_i^t)}\right) + \nabla f\left(\mathbf{x}^{(t-\gamma_i^t)}\right) \right\|^2 \right]$$

$$\leq \frac{L^2 \widetilde{\eta}_s^2 \eta_c^2 R_{max}}{b_{min}^2 \tau} \sigma_l^2$$

$$+ \frac{3L^2 \widetilde{\eta}_s^2 \eta_c^2 R_{max}}{b_{min}^2} \mathbb{E} \left[ \sum_{i \in \mathcal{C}^{(t)}} \left( \left\| \nabla f\left(\mathbf{x}^{(t-\gamma_i^t)}\right) \right\|^2 + \left\| \mathbf{h}_i^{(t-\gamma_i^t)} - \nabla f_i\left(\mathbf{x}^{(t-\gamma_i^t)}\right) \right\|^2 + \left\| \nabla f\left(\mathbf{x}^{(t-\gamma_i^t)}\right) - \nabla f_i\left(\mathbf{x}^{(t-\gamma_i^t)}\right) \right\|^2 \right) \right]$$

$$\leq \frac{L^2 \widetilde{\eta}_s^2 \eta_c^2 R_{max}}{b_{min}^2 \tau} \sigma_l^2 + \frac{3L^2 \widetilde{\eta}_s^2 \eta_c^2 R_{max}^2}{b_{min}^2} \sigma_g^2 + \frac{3L^2 \widetilde{\eta}_s^2 \eta_c^2 R_{max}}{b_{min}^2} \mathbb{E} \left[ \sum_{i \in \mathcal{C}^{(t)}} \left( \left\| \nabla f\left(\mathbf{x}^{(t-\gamma_i^t)}\right) \right\|^2 + \left\| \mathbf{h}_i^{(t-\gamma_i^t)} - \nabla f_i\left(\mathbf{x}^{(t-\gamma_i^t)}\right) \right\|^2 \right) \right].$$

Telescoping the inequality over $t = 0, \ldots, T-1$:

$$\frac{1}{T} \sum_{t=0}^{T-1} \mathbb{E} \left\| \nabla f\left(\mathbf{y}^{(t)}\right) - \nabla f\left(\mathbf{x}^{(t)}\right) \right\|^2 \leq \frac{L^2 \widetilde{\eta}_s^2 \eta_c^2 R_{max}}{b_{min}^2 \tau} \sigma_l^2 + \frac{3L^2 \widetilde{\eta}_s^2 \eta_c^2 R_{max}^2}{b_{min}^2} \sigma_g^2$$

$$+ \frac{3L^2\widetilde{\eta}_s^2\eta_c^2 R_{max}}{b_{min}^2} \frac{1}{T} \sum_{t=0}^{T-1} \mathbb{E}\left[\sum_{i\in\mathcal{C}^{(t)}} \left(\left\|\nabla f\left(\mathbf{x}^{(t-\gamma_i^t)}\right)\right\|^2 + \left\|\mathbf{h}_i^{(t-\gamma_i^t)} - \nabla f_i\left(\mathbf{x}^{(t-\gamma_i^t)}\right)\right\|^2\right)\right]$$

$$\leq \frac{L^2\widetilde{\eta}_s^2\eta_c^2 R_{max}}{b_{min}^2\tau}\sigma_l^2 + \frac{3L^2\widetilde{\eta}_s^2\eta_c^2 R_{max}^2}{b_{min}^2}\sigma_g^2 + \frac{6L^2\widetilde{\eta}_s^2\eta_c^2 R_{max}\rho_b\gamma^{\max}}{b_{min}} \frac{1}{T}\sum_{t=0}^{T-1}\mathbb{E}\left\|\nabla f\left(\mathbf{x}^{(t)}\right)\right\|^2$$

(Using Remark 2, however, this time,)

$$+ \frac{3L^2\widetilde{\eta}_s^2\eta_c^2 R_{max}}{b_{min}^2} \frac{1}{T} \sum_{t=0}^{T-1} \mathbb{E}\left[\sum_{i\in\mathcal{C}^{(t)}} \left\|\mathbf{h}_i^{(t-\gamma_i^t)} - \nabla f_i\left(\mathbf{x}^{(t-\gamma_i^t)}\right)\right\|^2\right] \qquad \text{(the maximum appearance can be } 2\gamma^{\max}b_{max})$$

$$\leq \frac{L^2\widetilde{\eta}_s^2\eta_c^2 R_{max}}{b_{min}^2\tau}\sigma_l^2 + \frac{3L^2\widetilde{\eta}_s^2\eta_c^2 R_{max}^2}{b_{min}^2}\sigma_g^2 + \frac{6L^2\widetilde{\eta}_s^2\eta_c^2 R_{max}\rho_b\gamma^{\max}}{b_{min}} \frac{1}{T}\sum_{t=0}^{T-1}\mathbb{E}\left\|\nabla f\left(\mathbf{x}^{(t)}\right)\right\|^2$$

$$+ \frac{3L^2\widetilde{\eta}_s^2\eta_c^2 R_{max}}{b_{min}^2} \frac{1}{T} \sum_{t=0}^{T-1} \mathbb{E}\left[\sum_{i\in\mathcal{C}^{(t)}} \left(\frac{L^2\eta_c^2\tau}{2(1-D)}\sigma_l^2 + \frac{D}{1-D}\left\|\nabla f_i\left(\mathbf{x}^{(t-\gamma_i^t)}\right)\right\|^2\right)\right] \qquad \text{(Using Lemma 2)}$$

$$\leq \left(1 + \frac{3R_{max}L^2\eta_c^2\tau^2}{2(1-D)}\right)\frac{L^2\widetilde{\eta}_s^2\eta_c^2 R_{max}}{b_{min}^2\tau}\sigma_l^2 + \frac{3L^2\widetilde{\eta}_s^2\eta_c^2 R_{max}^2}{b_{min}^2}\sigma_g^2 + \frac{6L^2\widetilde{\eta}_s^2\eta_c^2 R_{max}\rho_b\gamma^{\max}}{b_{min}} \frac{1}{T}\sum_{t=0}^{T-1}\mathbb{E}\left\|\nabla f\left(\mathbf{x}^{(t)}\right)\right\|^2$$

$$+ \frac{3L^2\widetilde{\eta}_s^2\eta_c^2 R_{max}}{b_{min}^2} \frac{1}{T} \sum_{t=0}^{T-1} \mathbb{E}\left[\sum_{i\in\mathcal{C}^{(t)}} \frac{D}{1-D}\left\|\nabla f_i\left(\mathbf{x}^{(t-\gamma_i^t)}\right)\right\|^2\right]$$

$$\leq \left(1 + \frac{3R_{max}L^2\eta_c^2\tau^2}{2(1-D)}\right)\frac{L^2\widetilde{\eta}_s^2\eta_c^2 R_{max}}{b_{min}^2\tau}\sigma_l^2 + \frac{3L^2\widetilde{\eta}_s^2\eta_c^2 R_{max}^2}{b_{min}^2}\sigma_g^2 + \frac{6L^2\widetilde{\eta}_s^2\eta_c^2 R_{max}\rho_b\gamma^{\max}}{b_{min}} \frac{1}{T}\sum_{t=0}^{T-1}\mathbb{E}\left\|\nabla f\left(\mathbf{x}^{(t)}\right)\right\|^2$$

$$+ \frac{3L^2\widetilde{\eta}_s^2\eta_c^2 R_{max}}{b_{min}^2} \frac{1}{T} \sum_{t=0}^{T-1} \mathbb{E}\left[\sum_{i\in\mathcal{C}^{(t)}} \left(\frac{2D}{1-D}\left\|\nabla f_i\left(\mathbf{x}^{(t-\gamma_i^t)}\right) - \nabla f\left(\mathbf{x}^{(t-\gamma_i^t)}\right)\right\|^2 + \frac{2D}{1-D}\left\|\nabla f\left(\mathbf{x}^{(t-\gamma_i^t)}\right)\right\|^2\right)\right]$$

$$\leq \left(1 + \frac{3R_{max}L^2\eta_c^2\tau^2}{2(1-D)}\right)\frac{L^2\widetilde{\eta}_s^2\eta_c^2 R_{max}}{b_{min}^2\tau}\sigma_l^2 + \frac{1+D}{1-D}\frac{3L^2\widetilde{\eta}_s^2\eta_c^2 R_{max}^2}{b_{min}^2}\sigma_g^2 + \frac{6L^2\widetilde{\eta}_s^2\eta_c^2 R_{max}\rho_b\gamma^{\max}}{b_{min}} \frac{1}{T}\sum_{t=0}^{T-1}\mathbb{E}\left\|\nabla f\left(\mathbf{x}^{(t)}\right)\right\|^2$$

$$+ \frac{3L^2\widetilde{\eta}_s^2\eta_c^2 R_{max}}{b_{min}^2} \frac{1}{T} \sum_{t=0}^{T-1} \mathbb{E}\left[\sum_{i\in\mathcal{C}^{(t)}} \frac{2D}{1-D}\left\|\nabla f\left(\mathbf{x}^{(t-\gamma_i^t)}\right)\right\|^2\right]$$

$$\leq \left(1 + \frac{3R_{max}L^2\eta_c^2\tau^2}{2(1-D)}\right)\frac{L^2\widetilde{\eta}_s^2\eta_c^2 R_{max}}{b_{min}^2\tau}\sigma_l^2 + \frac{1+D}{1-D}\frac{3L^2\widetilde{\eta}_s^2\eta_c^2 R_{max}^2}{b_{min}^2}\sigma_g^2 + \frac{6L^2\widetilde{\eta}_s^2\eta_c^2 R_{max}\rho_b\gamma^{\max}}{b_{min}} \frac{1}{T}\sum_{t=0}^{T-1}\mathbb{E}\left\|\nabla f\left(\mathbf{x}^{(t)}\right)\right\|^2$$

$$+ \frac{6L^2\widetilde{\eta}_s^2\eta_c^2 R_{max}\rho_b\gamma^{\max}}{b_{min}} \frac{2D}{1-D} \frac{1}{T} \sum_{t=0}^{T-1} \mathbb{E}\left\|\nabla f\left(\mathbf{x}^{(t)}\right)\right\|^2 \qquad \text{(Using Remark 2)}$$

$$= \left(1 + \frac{3R_{max}L^2\eta_c^2\tau^2}{2(1-D)}\right)\frac{L^2\widetilde{\eta}_s^2\eta_c^2 R_{max}}{b_{min}^2\tau}\sigma_l^2 + \frac{1+D}{1-D}\frac{3L^2\widetilde{\eta}_s^2\eta_c^2 R_{max}^2}{b_{min}^2}\sigma_g^2 + \frac{6L^2\widetilde{\eta}_s^2\eta_c^2 R_{max}\rho_b\gamma^{\max}}{b_{min}} \frac{1+D}{1-D} \frac{1}{T}\sum_{t=0}^{T-1}\mathbb{E}\left\|\nabla f\left(\mathbf{x}^{(t)}\right)\right\|^2.$$