# OpenReview forum: "FedAST: Federated Asynchronous Simultaneous Training"
_auai.org/UAI/2024/Conference — UAI 2024 poster_

### Official Review · Reviewer_UvrE · 2024-03-17

**Q2-1 Originality-Novelty:** 3
**Q2-2 Correctness-Technical Quality:** 3
**Q2-5 Clarity Of Writing:** 4

**Q10 Ethical Concerns:**

No.

**Q1 Summary And Contributions:**

This paper proposes a novel algorithm, named FedAST, to solve the staleness issues under the setting of multiple tasks federated learning. FedAST consists of two key components: a buffered asynchronous training strategy and an adaptive resource allocation method. Both experimental results and theoretical guarantees demonstrate the superiority over previous studies.

**Q2-3 Extent To Which Claims Are Supported By Evidence:**

3: Good: the main claims are supported by convincing evidence (in the form of adequate experimental evaluation, proofs, (pseudo-)code, references, assumptions).

**Q2-4 Reproducibility:**

3: Good: key resources (e.g. proofs, code, data) are available and key details (e.g. proofs, experimental setup) are sufficiently well-described for competent researchers to confidently reproduce the main results.

**Q3 Main Strengths:**

- The problem investigated in this paper is practical, and thus the motivation is sufficient.

- This paper is well-structured and clearly written.

- Extensive experiments are conducted to verify the effectiveness of the proposed method. Moreover, additional parameter-tuning studies are also shown in the appendix

**Q4 Main Weakness:**

- What is the size of the models used in table 1? And, could you replace the backbone with others, e.g., ResNet-18, to see whether the same results can be obtained as that shown in Figure 2.

- In Figure 2, what is the 'Sync-ST-NoStrag.Mit.'? There should be a brief introduction.

**Q5 Detailed Comments To The Authors:**

Please answer the above questions in Weakness.

**Q9 Complying With Reviewing Instructions:**

Yes

---

> ### Author Rebuttal · Authors · 2024-04-04
>
> We thank the reviewer for the positive and constructive comments.
>
> (1) The number trainable parameters of the models used in the experiments are MNIST: ~200 k, Fashion-MNIST: ~45 k, CIFAR-10: ~800k, Shakespeare: ~55 k.
> As the reviewer suggested, we additionally did an experiment using ResNet-18 (11.2 M parameters) backbone on CIFAR-100 dataset. We follow the implementation in [1] and adapt it to our simultaneous training setting. We run **four** ResNet-18 models simultaneously. Our method ($\texttt{FedAST}$) achieves the target accuracy level ($40\\%$) at $t\approx7100$ simulation time while the synchronous baseline ($\texttt{Sync-ST}$) achieves it at $t\approx11900$. This results in $40\\%$ time gain of the proposed method over the baseline. We will include this result and more experimental results with ResNet-18 in the paper as additional experiments, too!
>
>
> (2) FedAST mitigates the straggler issue using asynchronous updates. There are also other methods in the literature proposed for the straggler mitigation in synchronous FL, such as subsampling the clients in every round [2] or only accepting the fastest returned updates and discarding others [3]. In all of our experiments, we enhance the performance of competing synchronous baselines by adding extra straggler mitigation (only accepting the fastest updates every round), as explained in Section 5.2. In Figure 2, 'Sync-ST-NoStrag.Mit.' is the synchronous baseline without extra straggler mitigation that we add to improve the performance of the competing baseline. We will clarify this in the paper.
>
> In Figure 2, we show the poor performance of 'Sync-ST-NoStrag.Mit.'. Then, we use more difficult baselines by adding extra straggler mitigation in all other experiments as a default option. We also provide the validation experiments of this extra straggler mitigation technique for the competing baseline in Appendix D.1.
>
> We hope that our response has clarified your concerns and that you will consider increasing your score. We would be more than happy to answer any further questions you may have.
>
> [1] Jhunjhunwala, Divyansh, Shiqiang Wang, and Gauri Joshi. "FedExP: Speeding Up Federated Averaging via Extrapolation." The Eleventh International Conference on Learning Representations. 2022.
>
> [2] Luo, Bing, et al. "Tackling system and statistical heterogeneity for federated learning with adaptive client sampling." IEEE INFOCOM 2022-IEEE conference on computer communications. IEEE, 2022.
>
> [3] Bonawitz, Keith, et al. "Towards federated learning at scale: System design." Proceedings of machine learning and systems 1 (2019): 374-388.

---

### Official Review · Reviewer_Jkkp · 2024-03-25

**Q2-1 Originality-Novelty:** 3
**Q2-2 Correctness-Technical Quality:** 3
**Q2-5 Clarity Of Writing:** 4

**Q1 Summary And Contributions:**

The paper is motivated by the need to improve the efficiency and effectiveness of federated learning (FL) systems, especially in handling the straggler effect and stale updates in simultaneous model training. Its key contribution is the development of FedAST, a federated learning framework that employs buffered asynchronous aggregations to simultaneously train multiple models. FedAST significantly reduces training time, demonstrates robustness against stale updates, and outperforms existing synchronous FL methods, offering a scalable and efficient solution for simultaneous training in heterogeneous and dynamic environments.

**Q2-3 Extent To Which Claims Are Supported By Evidence:**

3: Good: the main claims are supported by convincing evidence (in the form of adequate experimental evaluation, proofs, (pseudo-)code, references, assumptions).

**Q2-4 Reproducibility:**

3: Good: key resources (e.g. proofs, code, data) are available and key details (e.g. proofs, experimental setup) are sufficiently well-described for competent researchers to confidently reproduce the main results.

**Q3 Main Strengths:**

1. Addresses the straggler problem in synchronous federated learning by using buffered asynchronous aggregation.
2. Dynamically reallocates client resources based on task progress, optimizing resource use across heterogeneous tasks.
3. Provides formal theoretical convergence guarantees for smooth non-convex objective functions.
4. Demonstrates superior empirical performance, with up to 46% reduction in training time across multiple real-world datasets.

**Q4 Main Weakness:**

1. Complexity in managing dynamic resource allocation across heterogeneous tasks.
2. Reliance on accurate estimations of heterogeneity and staleness for effective resource allocation.
3. Potential scalability issues in large federated networks due to the overhead of continuous resource management.
4. Limited exploration of other challenges associated with asynchronous federated learning, such as network latency and communication noise.
5. Baseline comparison with existing SOTA Async FL methods

**Q5 Detailed Comments To The Authors:**

1. There are few minor typos in this paper. I recommend the authors to proof-read this paper again (Grammarly is a suggestion).
2. Can you explain why Algorithm 2 Line 10 has client learning rate also in the equation?
3. I am struggling to understand what does R_m mean really? Can you explain a bit more (a sentence or so). I am assuming it’s a binary value for each client requesting them if they need to train or not. But its not clear in page 3
4. Can you comment on the work done in the following literature?
	a. Adaptive asynchronous federated learning in resource-constrained edge computing
	b. ASFL: Adaptive Semi-asynchronous Federated Learning for Balancing Model Accuracy and Total Latency in Mobile Edge Networks
5. Can you comment on the selection of buffer size? How is it chosen (optimally) and what are the effects of varying buffer sizes on the system's performance and accuracy? The varying buffer size might come from the resourced constrained EDGE devices as well right.

6. The literature review is missing reference of few important topics like Decentralized Federated learning. There are several works  on DFL with and without noisy channel + server/client selection + dynamic model parameter updates (including local updates, regularization etc.). Suggesting to refer/cite some of below:
	a. Robust federated learning with noisy communication
	b. On the Convergence of Decentralized Federated Learning Under Imperfect Information Sharing

7. How does FedAST's straggler mitigation strategy compare in performance with existing straggler mitigation techniques in federated learning? Can you do this comparision?

8. The comparison are primarily with synchronous models in literature. Can you perform comparative analysis with existing asynchronous models?
9. Scalability seems to be a concern on this method too. Can you explain what are the scalability limits , how does its performance change with an increasing number of clients and tasks?
How does the Last but one step become "<=". Is it due to Young's inequality with alpha = 1? You can use overset (a),(b) etc and mention (a) follows Jensen's/Young's/L-smoothness etc. it makes it easier to review the proof/lemmas

**Q9 Complying With Reviewing Instructions:**

Yes

---

> ### Author Rebuttal · Authors · 2024-04-04
>
> We thank the reviewer for the positive and constructive comments.
>
> Q2-4 Reproducibility & Q7 Score
>
> We are afraid the reviewer might have missed the supplementary materials, including the zip file of our code, a ReadMe file, and a demo experiment notebook. All other experimental details can be found in the main text and in Appendix C. We hope this addresses the reviewer’s concerns and prompts a reconsideration of the score.
>
> Q4
>
> (1) Despite the apparent complexity of our dynamic resource allocation method, its implementation is quite straightforward. It requires extra server-side memory to store the latest $V$ ($V$ is $8$ in our experiments) updates for each model. Then, the server estimates data heterogeneities by simple vector calculations (Eq. (2)) and finds the next resource allocation by solving Eq. (3), proportional to the estimated heterogeneity levels. All these operations, happening periodically and skipped in most rounds, are at the server, typically not compute/memory-limited.
>
> (2)&(3) We recognize the dependency of our approach on accurate heterogeneity estimates. However, the estimate quality increases with more stored updates, $V$, and the central servers usually have large memories. In cases where the server is resource-constrained, running FedAST without dynamic client allocation (static option, $option=S$) may be more practical. We also experimentally show the effectiveness of the proposed resource allocation in Appendix Section D.5.
>
> (4) Using asynchronous aggregation can, in fact, alleviate challenges due to network latency and communication noise, which will severely hamper the performance of synchronous algorithms. Using an asynchronous algorithm can help reduce the bandwidth requirement as the updates are not sent out or collected concurrently.  We will include a discussion on these in the Related Work section.
>
> (5) There is only one prior work [5] on asynchronous *simultaneous* FL, as discussed in Section 5.2. As [5] can be seen as a bufferless static version of our algorithm, we did ablation experiments (Figures 2 & 3 for buffer, and Appendix D.5 for dynamic resource allocation ablations). Setting M=1 in FedAST reduces it to special cases with and without buffer considered in asynchronous single-model works [1] and [2]. Therefore, we expect a similar single-model performance to [1]. We also provide a theoretical comparison in Appendix B. Recent asynchronous algorithms [3, 4] can be competitive to [1], yet they lack the straightforward extension to simultaneous federated learning. We kindly note that the setting and contributions in our work are different than the single-model asynchronous FL.
>
> Q5
>
> (2) In Alg. 1 Line 5 returns the accumulated raw updates divided by the learning rate. So, the multiplication in Line 10 of Alg. 2 simply re-applies the learning rate. This approach eases the proof notation without any practical difference.
>
> (3) The variables $R_m$ denote the numbers of active training requests for all *task $m\in[M]$*. As we have limited available clients, $R_m$ distribution across tasks balances how much resources are allocated to each task.
>
> (4) Thanks for bringing these related works to our attention. Both (a.) and (b.) contribute to asynchronous *single-model* FL rather than simultaneous training of multiple models. We will discuss them in Related Work.
>
> (5) The buffer size is chosen through validation experiments depending on the number of training requests, as detailed in Appendix C.5. We find a balance between less stale updates (large buffer) and faster global rounds (small buffer).
>
> (6) We will include the discussions on decentralized FL and communication noise in Related Work section using the reference papers.
>
> (7) Please see our Answer 2 to the last reviewer (UvrE) due to space limitation.
>
> (8) Please see our answer to Q4 (5) above.
>
> (9) FedAST demonstrates scalability advantages over baselines. Figure 5 shows FedAST's increased time gain with more simultaneous tasks. Figure 3 shows that FedAST maintains its performance even with an increasing number of available clients thanks to the buffer.
>
> We will improve the writing of our proofs with your suggestions. Our proof relaxes impractical assumptions used in [1] and provides more general analysis than [2] by considering the buffer and multiple local steps. For the specific query on the proof step, we are ready to provide a detailed explanation if you can clarify which line on which page you mentioned.
>
> We hope that our response has clarified your concerns and that you will consider increasing your score.
>
> [1]Federated learning with buffered asynchronous aggregation
>
> [2]Sharper convergence guarantees for asynchronous SGD for distributed and federated learning
>
> [3]Favas: Federated averaging with asynchronous clients
>
> [4]QuAFL: Federated Averaging Made Asynchronous and Communication-Efficient
>
> [5]Asynchronous multi-model federated learning over wireless networks: Theory, modeling, and optimization

---

### Official Review · Reviewer_DovW · 2024-03-26

**Q2-1 Originality-Novelty:** 2
**Q2-2 Correctness-Technical Quality:** 3
**Q2-5 Clarity Of Writing:** 2

**Q1 Summary And Contributions:**

The paper proposes a method to tackle the asynchronous federated multiple models training problem. The problem setup assumes that there are many models that need to be trainned in a federated learning approach. Due to heterogeneity of clients, there might be stragglers in the training process. The paper proposes a federated learning method that allows for asynchronous model aggregation with the help of model update buffers. The authors develope optimization convergence guarantee for their proposed method. They also demonstrated the advantages of their proposed method by implementing several numerical experiments.

**Q2-3 Extent To Which Claims Are Supported By Evidence:**

3: Good: the main claims are supported by convincing evidence (in the form of adequate experimental evaluation, proofs, (pseudo-)code, references, assumptions).

**Q2-4 Reproducibility:**

3: Good: key resources (e.g. proofs, code, data) are available and key details (e.g. proofs, experimental setup) are sufficiently well-described for competent researchers to confidently reproduce the main results.

**Q3 Main Strengths:**

The main challenges that are addressed by the paper are two-fold: (i) there are multiple models that need to be trained simulataneously; (ii) due to heterogeneity among clients and tasks, there are stragglers.

The proposed method utilizes model update buffers to allow for asyncronous model aggregation. The number of active training requests and the buffer size can be adjusted dynamically. While these ideas have their origin in previous literature, the authors combine them in a non-trivial way.

Besides, the authors provide theoretical analysis on the optimization convergence rate of the proposed method. I appreciate the theoretical attempt to justify their proposed method, though I have technical questions (see below).

The authors implemented several numerical experiments, compared with several baseline methods on several datasets. These experiments help empirically justify the proposed method.

**Q4 Main Weakness:**

1. I have doubts on the pratical interest of training multiple models simultaneously in a federated fashion. I think the authors may consider providing more examples to justify why this problem matters.

2. Assumption 4 can be a strong assumption. Although I did not check the proof carefully, my guess is that Assumption 4 helps circumvent the technical challenges of showing convergence of asynchronous aggregation. While I understand why this assumption may be technically important, in some cases, $\gamma^{\max}_m$ can be very large or even infinite.

3. I am a little bit confused about how $R_m$ plays a role in Algorithm 2. Based on my understanding, it seems that $R_m$ only affects the number of clients selected in the first round, and it does not affect the client selection in the future rounds. But I do not think this makes sense. Is this a typo? Maybe the authors should improve the writing of Algorithm 2.

**Q5 Detailed Comments To The Authors:**

1. Provide more examples and justification of the pratical interest of the studied problem.

2. Try to justify Assumption 4 more carefully. I noticed that the authors made some attempts to justify it when assuming the waiting time of each client follow some distribution. It is worth making it more rigorous. For example, by showing that Assumption 4 can be satisifed with high probability when the waiting time of each client follows a certain stochatisc model.

3. Try to improve the writing on Algorithm 2 and potentially on other technical descriptions.

**Q9 Complying With Reviewing Instructions:**

Yes

---

> ### Author Rebuttal · Authors · 2024-04-04
>
> We thank the reviewer for the positive and constructive comments.
>
> (Q4&Q5-1) Simultaneous training (ST), where multiple ML models are trained together in a federated setting, can be used in many applications that require multiple ML models. For instance, our smartphones utilize multiple ML models, including a next-word predictor keyboard and vision models for predicting images possibly liked [1]. A chat application may require speech recognition and response text generator models concurrently. Another example is a simple form of neural architecture search: we may train many different architectures in parallel to figure out which one works the best on a particular dataset. Recent experimental works further validate ST's effectiveness; for instance, [2] proposes ST in smart car networks for multiple tasks, such as pothole detection and driving maneuver prediction simultaneously. [3] suggests the ST for air quality index forecasting. We will enrich the paper's discussion with more examples!
>
> (Q4&Q5-2) Assumption 4 (Maximum Staleness) is used in the convergence proof to guarantee that none of the assigned updates takes an arbitrarily large time to return to the server. Although this might not hold for heavy-tailed delay distributions, it holds with a high probability for distributions such as uniform, exponential, and shifted exponential, which are frequently used in modeling delays in ML works [4, 5, 6]. Therefore, this assumption is frequently used in theoretical works in the literature on distributed machine learning and asynchronous federated learning [5, 7, 8].  Furthermore, maximum staleness can be easily enforced algorithmically by simply dropping overly delayed updates [8].
>
> (Q4&Q5-3) The variable $R_m$ denotes the number of active training requests for task $m$ in the system. In the first round of training, for each model $m$, the server sends out $R_m$ local training requests to $R_m$ clients. Whenever a client returns an update of task $m$ back to the server, the server sends out a new training request for task $m$. So, the number of active training requests for any task $m$ in the system is kept at $R_m$. Without dynamic reallocation ($option=S$), $R_m$ values do not change throughout the training. However, with dynamic reallocation ($option=D$), $R_m$ values are redistributed periodically based on the estimated data heterogeneity levels of tasks. This corresponds to the reallocation of resources as the server basically adjusts how many training requests are allocated for each task $m$. During reallocation, the server maintains the total number of active training requests ($\sum_mR_m$) fixed since this depends on how many available clients the system has, which plays a role as a constraint on available resources.
>
> We hope that our response has clarified your concerns and that you will consider increasing your score. We would be more than happy to answer any further questions you may have.
>
> [1] Communication-efficient learning of deep networks from decentralized data
>
> [2] FLOW: A Scalable Multi-Model Federated Learning Framework on the Wheels
>
> [3] Insights into Multi-Model Federated Learning: An Advanced Approach for Air Quality Index Forecasting
>
> [4] Slow and stale gradients can win the race: Error-runtime trade-offs in distributed SGD
>
> [5] Federated learning with buffered asynchronous aggregation
>
> [6] Fedpaq: A communication-efficient federated learning method with periodic averaging and quantization
>
> [7] More effective distributed ml via a stale synchronous parallel parameter server
>
> [8] Sharper convergence guarantees for asynchronous SGD for distributed and federated learning

---

### Official Review · Reviewer_Uwys · 2024-04-01

**Q2-1 Originality-Novelty:** 3
**Q2-2 Correctness-Technical Quality:** 3
**Q2-5 Clarity Of Writing:** 3

**Q1 Summary And Contributions:**

In this paper authors focus on multi-task multi-model asynchronous federated learning (each model for a single task). They proposed, FedAST, a buffered asynchronous federated simultaneous training algorithm that overcomes bottlenecks from slow models and adaptively allocates client resources across heterogeneous tasks. They provide theoretical convergence guarantees for FedAST for smooth non-convex objective functions. Extensive experiments over multiple real-world datasets demonstrate that their proposed method outperforms existing simultaneous FL approaches, achieving up to 46.0% reduction in time to train multiple tasks to completion.

**Q2-3 Extent To Which Claims Are Supported By Evidence:**

2: Fair: the main claims are somewhat supported by evidence (but the experimental evaluation may be weak, or does not match entirely with the claims, important baselines may be missing, proofs contain important ideas but lack rigor, algorithmic details are only discussed superficially, references are imprecise, assumptions are not sufficiently motivated or explicated, etc.).

**Q2-4 Reproducibility:**

3: Good: key resources (e.g. proofs, code, data) are available and key details (e.g. proofs, experimental setup) are sufficiently well-described for competent researchers to confidently reproduce the main results.

**Q3 Main Strengths:**

(1) The targeted task scenario is very interesting and related to real life applications, where indeed multiple models are usually involved in a single application. For instance, in the real life online e-commerce application, multiple models are involved in modeling user preferences (ads, product, and etc)

(2) The core idea of the paper to dynamically allocate the resource based on the data heterogeneity of each task is very novel and solid. Using real time parameter update heterogeneity to dynamically adjust the number of active clients and buffer size (essentially the batch size) to reduce model drifting and forgetting caused by heterogeneity is very elegant.

(3) The experiment results indeed support the authors' claim of significant acceleration. Also authors provide detailed theoretical analysis for their method.

**Q4 Main Weakness:**

(1) The main concern I have is that it is unclear in the paper how the proposed method dynamically allocates the new requests across devices (like based on the heterogeneous computation power of the devices). I believe in section 3, authors mentioned the new training requests are sent to uniformly sampled devices. This seems not optimal as (1) the faster devices should handle more requests as it will decrease the overall training time (2) the allocation algorithm should also consider the data distribution or the model parameter drifting on each device. I believe more discussion on this can also help to illustrate how the current method can further alleviate the staleness problem in asynchronous federated learning.

(2) Another weakness is that the experiment datasets are too simple to really support the solidity of the proposed method. In most of the federated learning papers, at least CIFAR100, tiny-Imagent and similar datasets are widely used.

**Q5 Detailed Comments To The Authors:**

Please the weakness section

**Q9 Complying With Reviewing Instructions:**

Yes

---

> ### Author Rebuttal · Authors · 2024-04-04
>
> We thank the reviewer for the positive and constructive comments.
>
> (1) Our proposed dynamic resource allocation method balances how many clients and resources should be allocated to each task. However, it does not affect which specific clients are selected. Instead, the server employs a uniform-at-random client selection strategy to ensure fair and unbiased training across all clients for all tasks. Our theoretical guarantees also utilize this uniform sampling assumption. Clients have comparable computing and communication capabilities, but there is inherent randomness in task finish times. Employing an asynchronous training mechanism in effect will average out randomness in their task finish times by eliminating the need for clients to wait for other clients to finish. Considering the heterogeneous local data distributions or computational capabilities, as the reviewer suggests, could be beneficial. We will discuss these as future directions that can be pursued orthogonally and combined with our proposed algorithm.
>
> (2) We thank the reviewer for the suggestion! To enrich our study and address the reviewer's concern, we have added the experiments using ResNet-18 (a larger model than currently used CNNs) on CIFAR-100 dataset. We follow the implementation in [1] and adapt to our simultaneous training setting. We run **four** ResNet-18 models simultaneously. Our method ($\texttt{FedAST}$) achieves the target accuracy level ($40\\%$) at $t\approx7100$ simulation time while the synchronous baseline ($\texttt{Sync-ST}$) achieves it at $t\approx11900$. This results in $40\\%$ time gain of the proposed method over the baseline. We will include this result and more experimental results with CIFAR-100 in the paper as additional experiments, too!
>
>
> We hope that our response has clarified your concerns and that you will consider increasing your score. We would be more than happy to answer any further questions you may have.
>
>
> [1] Jhunjhunwala, Divyansh, Shiqiang Wang, and Gauri Joshi. "FedExP: Speeding Up Federated Averaging via Extrapolation." The Eleventh International Conference on Learning Representations. 2022.

---

### Meta-Review · Area_Chair_8jWN · 2024-04-16

This paper introduces "FedAST," an innovative approach to asynchronous federated
learning that dynamically allocates client resources to simultaneously train
multiple models. The proposed method aims to address the issues of slow models
and system bottlenecks in multi-task, multi-model federated learning
environments. The authors provide both theoretical convergence guarantees and
empirical results using real-world datasets to support their claims.

Pros of the Paper:
+ Introduces an innovative asynchronous federated learning algorithm, FedAST, which dynamically allocates resources to handle multiple models and tasks simultaneously.
+ Offers theoretical convergence guarantees for non-convex objectives, adding robustness to the method's scientific foundation.
+ Demonstrates empirical success with significant reductions in training time across various real-world datasets.
+ Provides a detailed theoretical analysis and extensive experimental validation that supports the claims made.

Cons of the Paper:
- Some practical aspects of the dynamic resource allocation method are not fully explored, particularly the optimal allocation of new training requests across devices.
- Uses relatively simple experimental datasets, which may not convincingly demonstrate the method's effectiveness on more complex or realistic tasks.
- Limited comparison with existing state-of-the-art asynchronous federated learning methods, which could contextualize its performance and innovative aspects better.
- Some parts of the algorithm and its assumptions are insufficiently justified or explained, which could impact the clarity and reproducibility of the research.

Given the paper's contributions to tackling real-world problems in federated
learning, I recommend accepting this paper. Minor revisions can be incorporated
when preparing the camera ready version. The authors should focus on enhancing
the clarity of their writing, extending their comparisons with both synchronous
and asynchronous methods, and incorporating the suggested broader applications
and theoretical justifications into their final manuscript.